# Enhanced responsivity and detectivity of fast WSe$_2$ phototransistor using electrostatically tunable in-plane lateral p-n homojunction

Sayantan Ghosh[1], Abin Varghese [1,2,3], Kartikey Thakar [1], Sushovan Dhara[1] & Saurabh Lodha [1✉]

Layered transition metal dichalcogenides have shown tremendous potential for photo-detection due to their non-zero direct bandgaps, high light absorption coefficients and carrier mobilities, and ability to form atomically sharp and defect-free heterointerfaces. A critical and fundamental bottleneck in the realization of high performance detectors is their trap-dependent photoresponse that trades off responsivity with speed. This work demonstrates a facile method of attenuating this trade-off by nearly 2x through integration of a lateral, in-plane, electrostatically tunable p-n homojunction with a conventional WSe$_2$ phototransistor. The tunable p-n junction allows modulation of the photocarrier population and width of the conducting channel independently from the phototransistor. Increased illumination current with the lateral p-n junction helps achieve responsivity enhancement upto 2.4x at nearly the same switching speed (14–16 µs) over a wide range of laser power (300 pW–33 nW). The added benefit of reduced dark current enhances specific detectivity ($D^*$) by nearly 25x to yield a maximum measured flicker noise-limited $D^*$ of $1.1 \times 10^{12}$ Jones. High responsivity of 170 A/W at 300 pW laser power along with the ability to detect sub-1 pW laser switching are demonstrated.

[1] Department of Electrical Engineering, IIT Bombay, Mumbai, India. [2] Department of Materials Science and Engineering, Monash University, Clayton, VIC, Australia. [3] IITB-Monash Research Academy, IIT Bombay, Mumbai, India. ✉email: slodha@ee.iitb.ac.in

Photodetectors play a significant role in modern society due to diverse applications across the electromagnetic spectrum, such as in short- and long-range communication, visible and infrared camera sensors, and biomedical imaging. Commercially available high responsivity, high detectivity, and fast photodetectors are based on silicon and group III–V semiconductors. Recently, two-dimensional layered van der Waals (vdW) materials, especially transition metal dichalcogenides (TMDs), have emerged as promising optoelectronic candidates due to their (i) large optical absorption, (ii) broad spectral response ranging from deep ultraviolet to infrared, and (iii) excellent tensile strength for flexible device applications. Among the TMDs, $WSe_2$ has been studied in great detail owing to its (i) sizeable bandgap (~1.2 eV for few-layer and bulk), (ii) relatively high carrier mobility (hole mobility of ~118 cm$^2$/Vs)[1], (iii) high absorption coefficient (> 10$^5$ cm$^{-1}$ at 532 nm)[2], and (iv) ambient air stability[3] unlike other popular vdW materials such as black phosphorus[4,5] and InSe[6,7].

There exists a strong trade-off between the two key performance metrics of TMD photodetectors, viz. responsivity ($R$) and speed[8]. $R$ is directly proportional to the photoconductive gain ($G$) of a phototransistor, where $G$ depends on the channel mobility and photogenerated carrier lifetime. Most studies have observed that high carrier lifetime leads to large gain, and consequently enhanced photoresponsivity[8,9]. On the other hand, the speed of a phototransistor decreases with increasing carrier lifetimes. For example, Lopez-Sanchez et al. have reported a monolayer (ML) $MoS_2$ phototransistor with a high responsivity of 880 A/W but a response time > 4 s limits its speed[10]. Similarly, a $WSe_2$-based phototransistor has been reported with ultrafast rise and fall switching times < 8 μs but with a responsivity of 0.6 A/W[11]. Hence, ways of attenuating the trade-off between responsivity and speed can truly enable high-performance photodetectors that simultaneously exhibit fast switching and high responsivity. Furthermore, photodetector's specific detectivity ($D^*$) quantifies its ability to detect low incident laser power such that a large $D^*$ value is desirable for highly sensitive photodetection. Typically, $D^*$ is limited by the dark current of the detector, and hence, a low dark current is essential to achieve high $D^*$[12,13].

In this work, we demonstrate a phototransistor architecture wherein a conventional $WSe_2$ phototransistor has been integrated with an in-plane, lateral, p–n diode in the direction transverse to the source-drain direction. The p–n homojunction diode is electrostatically tunable through gate electrodes. Chemical doping has been employed to enhance the photoresponsivity on layered semiconductors till date[14,15]. This work relies on electrostatic doping due to its ease of tunability, reversibility, and area selectivity[16–18]. We show substantial attenuation (2x) of the responsivity-speed trade-off through electrostatic modulation of the lateral p–n junction using comprehensive steady-state and temporal photoresponse measurements under a 532 nm laser. As a result, the lateral p–n diode action enables responsivity enhancement by 1.1x –2.4x at nearly the same switching speed (14–16 μs) over a wide range of laser power (300 pW–33 nW). High responsivity (94 A/W), and speed (14 μs) are demonstrated for 1 nW incident power, reaching peak values of 170 A/W at 300 pW. Along with this, a flicker noise-limited maximum measured $D^*$ value of $1.1 \times 10^{12}$ Jones, enhanced by 4.2x –25x due to dark current reduction by the lateral p–n junction, makes this one of the fastest high-responsivity and high-detectivity $WSe_2$ phototransistors till date.

## Results

### Phototransistor architecture and fabrication.
Three interdigitated bottom metal gates (Fig. 1a) were fabricated adjacent to each other on a 285 nm $SiO_2$/Si substrate using e-beam lithography and metal sputtering. The two side gates (SGs) that sandwich the

middle gate (MG) were electrically shorted by connecting them with an additional metal line during fabrication. The bottom gates were metallized using a Cr (2 nm)/Au (30 nm) stack. Next, an hBN flake was transferred selectively on top of the three bottom gates, followed by a thin $WSe_2$ flake on top of the hBN. Here hBN serves as the bottom gate dielectric and $WSe_2$ as the channel material. The flake transfers were followed by e-beam patterning of source/drain (S/D) contacts on $WSe_2$ at both ends of the MG, perpendicular to the SG–MG–SG direction and overlapping with the MG. Cr (2 nm)/Pt (30 nm)/Au (80 nm) stack was deposited by sputtering to form S/D contacts. Finally, the as-fabricated device was annealed in ambient at 150 °C for 1 h. The anneal helps in Pt d-orbital hybridization with $WSe_2$, thereby lowering the S/D contact resistance for improved p-type conduction[15]. This completes the formation of a field-effect transistor (FET) with a $WSe_2$ conduction channel over the MG along with adjacent (side) $WSe_2$ regions over the SGs. The fabrication process flow has been described in detail in Supplementary Fig. 1. Although mechanically exfoliated flakes from bulk crystals have been employed in this work, the use of CVD-grown large area flakes/layers of hBN[19] and $WSe_2$[20] can enable wafer-scale fabrication of a large array of devices. A 3D schematic of the device and an optical microscope image in Fig. 1a, b show the complete device architecture with bottom gates and top S/D contacts. The thickness of the hBN and $WSe_2$ flakes was determined to be ~15 and ~7 nm (around 10 layers) respectively, using atomic force microscopy scans as shown in Fig. 1c. Each of the three gates controls the local carrier density in the $WSe_2$ region above it through electrostatic doping. Besides controlling the carrier density (doping) in the $WSe_2$ side regions, the SGs also control the potential barriers between the $WSe_2$ side regions and the middle channel. Hence, by choosing appropriate MG and SG voltages, it is possible to selectively attain $p_S–n_M–p_S$ or $n_S–p_M–n_S$ band configurations along the SG–MG–SG direction (in-plane and perpendicular to the S/D channel). As a result, the SGs can influence the middle FET action in dark and under illumination. Device simulations with a representative three-gate structure on a thin $Si/SiO_2$ system demonstrate the effect of MG- and SG-induced electrostatic doping on the channel width under dark and the photocarrier population under illumination for varying $V_{SG}$ as shown in Supplementary note 2.

### Dark characteristics and photoresponsivity.
The formation of lateral p–n junctions at the two SG–MG interfaces, along the sides of the middle channel between S/D, is critical to the demonstration of SG modulation of the photoresponse. To demonstrate electrostatic doping and p–n diode action along the lateral MG–SG direction (diodes are in-plane and perpendicular to the direction of S/D conduction), side contacts ($C_{SG}$), one on each side of the $WSe_2$ flake over each SG, were fabricated on a separate device having the same architecture of the back gates and S/D contacts. The device schematic is shown in Supplementary Fig. 3a along with a schematic representation of the p–n diode in Fig. 2b inset. The current–voltage (IV) characteristics shown in Supplementary Fig. 3b along with the lateral MG (S contact)-to-SG ($C_{SG}$) contact change from that of a resistor (n–n) to that of a diode (p–n) with a rectification ratio of ~10$^4$, when the SG is kept at a fixed bias ($V_{SG} = 2$ V) and the MG is tuned from 2 to −2 V. With $V_{SG}$ fixed at 2 V, the $WSe_2$ region over the SG becomes n doped while the doping in the $WSe_2$ region over the MG changes from n to p type when $V_{MG}$ is changed from 2 to −2 V. This clearly shows the formation of a gate tunable lateral p–n junction along the side of the S/D channel. Electron energy band diagrams corresponding to the IV characteristics are shown in Supplementary Fig. 3c. They depict the change from n–n to p–n with appropriate bias voltages at MG and SG.

Figure 2a shows p-type $WSe_2$ FET transfer characteristics ($I_D V_{MG}$) at $V_D = 1$ V where the source terminal is grounded,

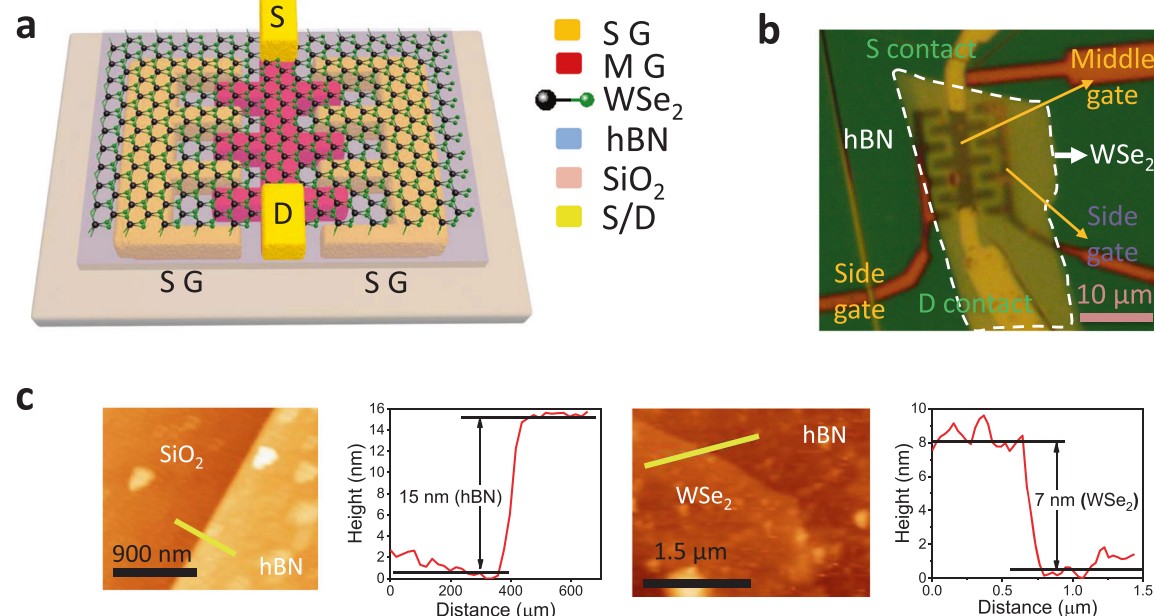

**Fig. 1 Device architecture and material thicknesses. a** 3D schematic of the WSe₂ phototransistor with middle (MG) and side gates (SG). The transparent blue layer is hBN on top of metal back gates on SiO₂/Si and underneath WSe₂ (channel material). The top layer in the green ('Se' atom) and black ('W' atom) ball-stick structure represents WSe₂. **b** Optical microscope image of the device with MG and SGs defined at the bottom and S/D contacts on the top. **c** AFM scans indicate that the WSe₂ channel is 7 nm thick and the hBN gate dielectric is 15 nm thick.

under dark, when $V_{SG}$ is not applied. An on/off current ratio of ~10⁵ and a hole mobility of 5 cm²/Vs indicate good MG control and hole transport respectively. The output characteristics ($I_D V_D$) in the inset indicate Schottky barrier-dominated transport across the S/D contacts to the WSe₂ channel over the MG. High drain on-current (in nA) for positive voltage bias at the metal source contact to the WSe₂ channel, and low off-current (~pA) confirm the formation of a p-type Schottky contact between Pt and WSe₂ that is favourable for hole injection. Dark $I_D V_{MG}$ characteristics in Fig. 2b show a monotonous decrease in drain current as $V_{SG}$ changes from −2 to 2 V. This is due to increasing depletion of the hole concentration, resulting in the formation of a lateral p$_M$–n$_S$ junction across the MG and SG WSe₂ regions, with increasing (more positive) $V_{SG}$. Hence the width of the conducting S/D channel decreases, leading to a reduction in current. For $V_{MG} = 0$ V, when $V_{SG}$ is not applied, a p$_S$–p$_M$–p$_S$ configuration is formed in WSe₂ along the lateral SG–MG–SG direction, as shown in Fig. 2c. Under this condition, since the S/D contacts are patterned with an overlap with only the MG, the S/D conduction channel width is defined by the physical width of the MG (2 μm).

The band diagram shown in Fig. 2c is considered to be flat under No-$V_{SG}$ condition, for a simple understanding of the device operation. On the other hand, when $V_{SG} = 2$ V is applied for $V_{MG} = 0$ V, the band configuration changes from p$_S$–p$_M$–p$_S$ to n$_S$–p$_M$–n$_S$, as $V_{SG} = 2$ V induces electrons in both SG WSe₂ regions making them n-type. Therefore, with p$_M$–n$_S$ configuration between the MG and both SGs, as the space charge region (depletion) encroaches the MG area, it effectively decreases the conduction channel width to < 2 μm. Hence, the absolute S/D current ($I_{dark}$) decreases in the dark state with positive $V_{SG}$. Based on this reasoning, band alignment for different $V_{SG}$ conditions for a fixed $V_{MG} = 0$ V is depicted in Fig. 2c to further understand the trend in $I_{dark}$ when $V_{SG}$ is tuned from −2 to No-$V_{SG}$ to 2 V. Figure 2d shows the relative change in dark current with respect to the channel carrier concentration (p$_{WSe_2}$). It can be seen that SG modulation of the channel current is higher for lower p$_{WSe_2}$ in

the range of 10¹³ to 10¹⁵ cm⁻³ (sub-threshold region), as it is easier to deplete at lower S/D channel doping density. The calculation of p$_{WSe_2}$ is shown in Supplementary note 4.

Photoresponse of the WSe₂ transistor was obtained under 532 nm laser illumination. The light was incident on the entire device area to ensure photogeneration of electron-hole (e-h) pairs in WSe₂ over the MG as well as the adjacent SG regions. From the total current upon illumination ($I_{light}$), the photocurrent is obtained as $I_{ph} = I_{light} - I_{dark}$. The $I_{ph}$ vs $V_{MG}$ plot in Fig. 3a shows a monotonous increase in $I_{ph}$ with $V_{SG}$ increasing from −2 to 2 V in the ON state. This trend in $I_{ph}$ with $V_{SG}$ is explained through band diagrams in Fig. 3b. When $V_{MG} = 0$ V and $V_{SG} = 2$ V (n$_S$–p$_M$–n$_S$ configuration) electric fields (from SG to MG) in the lateral depletion regions drive electrons photogenerated near and in the MG depletion region of WSe₂ out towards the SG regions. Similarly, holes photogenerated near and in the SG depletion regions of WSe₂ are driven towards the MG channel. These additional holes provide increased photocurrent in the n$_S$–p$_M$–n$_S$ configuration. Even a single SG will show a similar but smaller increase in photocurrent through the n$_S$–p$_M$ configuration under the ON state. Detailed advantages of using a three-gate structure as compared to the conventional single-gate FET are given in Supplementary note 5. On the other hand, the p$_S$–p$_M$–p$_S$ configuration ($V_{MG} = 0$ V, No-$V_{SG}$) has a relatively flatter band alignment in the SG–MG–SG direction, while the p⁺$_S$–p$_M$–p⁺$_S$ configuration ($V_{MG} = 0$ V, $V_{SG} = -2$ V) has depletion electric fields directed from the MG towards the SG regions that drive photogenerated holes out from the MG channel towards the SG regions. Consequently, $I_{ph}$ decreases progressively with decreasing $V_{SG}$. The photoresponsivity ($R$) in Fig. 3c is calculated using the relation

$$R = I_{ph}/P_{in} \qquad (1)$$

where $P_{in}$ is the input optical power. Since $R$ is directly proportional to $I_{ph}$, for a fixed $P_{in} = 33$ nW, $R$ can be modulated

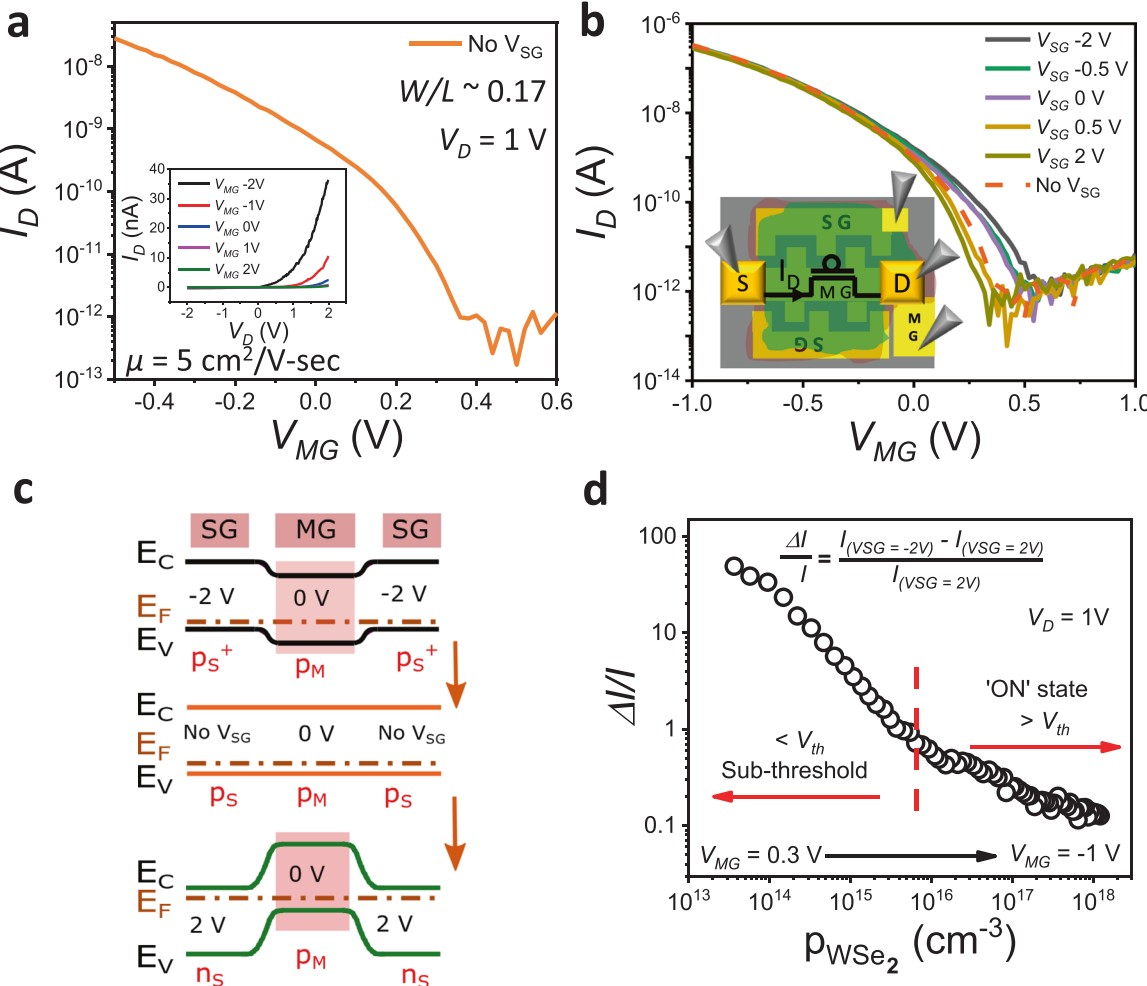

**Fig. 2 Device characterization and band alignment under dark. a** The transfer characteristic ($I_D V_{MG}$) of FET in dark condition shows an on/off current ratio close to $10^5$ when $V_{SG}$ is not applied. The inset shows the output characteristics ($I_D V_D$) without $V_{SG}$. **b** Transfer characteristics ($I_D V_{MG}$) for $V_{SG}$ varying from $-2$ ($p^+_S$–$p_M$–$p^+_S$) to 2 V ($n_S$–$p_M$–$n_S$), for fixed $V_D = 1$ V, under dark condition. $I_D$ decreases, going from $p^+_S$–$p_M$–$p^+_S$ to $n_S$–$p_M$–$n_S$ configuration. The inset shows the FET measurement configuration, where MG is the only gate controlling S/D conduction. **c** Energy band diagrams across the SG–MG–SG direction illustrating the transition from $p^+_S$–$p_M$–$p^+_S$ to $n_S$–$p_M$–$n_S$ as $V_{SG}$ changes from $-2$ to 2 V for fixed $V_{MG} = 0$ V (sub-threshold region). The semi-transparent box in pink shows the effective source-to-drain channel width under p–p$^+$ and p–n depletion from both sides of the channel. **d** Normalized change in $I_D$ vs source-to-drain channel carrier concentration ($p_{WSe_2}$), showing much higher change at lower concentrations in the sub-threshold region.

by nearly 24x from 1 A/W at $V_{SG} = -2$ V to a high value of 24 A/ W at $V_{SG} = 2$ V, as the band configuration changes from $p^+_S$–$p_M$–$p^+_S$ to $n_S$–$p_M$–$n_S$. Also, $R$ is higher by 2.4x for $V_{SG} = 2$ V in comparison to the conventional No-$V_{SG}$ configuration. This significant increase in $R$ can be attributed to a two-fold enhancement in $I_{ph}$ as discussed earlier, (a) a decrease in $I_{dark}$ and (b) an increase in $I_{light}$ for the same change in $V_{SG}$ that changes the band configuration in SG–MG–SG direction from $p_S$–$p_M$–$p_S$ to $n_S$–$p_M$–$n_S$. It is also interesting to note that while $V_{SG}$ modulation of the dark current is maximum in the sub-threshold region, both $R$ and $I_{ph}$ show maximum modulation and maximum or minimum values depending on $V_{SG}$ being positive or negative, when the channel is on ($V_{MG} = -0.5$ V). This indicates that $V_{SG}$ modulation of $I_{light}$ dominates the photo-response at high hole concentrations in the channel.

Figure 3d shows the dependence of $I_{ph}$ on $P_{in}$ for fixed $V_{MG} = -0.5$ V and $V_{SG}$ varying from $-2$ to 2 V. The power exponent ($\alpha$) increases from 0.48 to 0.61 as $V_{SG}$ goes from $-2$ to 2 V and $\alpha = 0.52$ for No-$V_{SG}$. $\alpha < 1$ indicates the presence of traps in the S/D channel[8], which could lead to significant recombination of the photogenerated e-h pairs in the channel before they reach the S/D

contacts. The larger the value of $\alpha$, the lesser is the number of trap states participating in current conduction[8,9]. Along with a decrease in channel width, an opposite and stronger reverse bias internal electric field builds up as $V_{SG}$ increases from $-2$ to 2 V. This helps in sweeping out electrons from the conduction band over MG to the SG regions and accumulating holes from the SG regions into the MG channel for $V_{SG} = 2$ V. On the other hand, for $V_{SG} = -2$ V, the electrons and holes flow in opposite directions, causing the accumulation of electrons and depletion of holes in the MG area. This leads to more pronounced e-h pair recombination in the case of $V_{SG} = -2$ V. Hence a smaller number of photogenerated carriers reach the contacts. This is consistent with a lower value of $\alpha$ for $V_{SG} = -2$ V indicating a higher effective trap density (higher recombination rate) and is one of the factors behind the lower responsivity and photo-conductive gain at $V_{SG} = -2$ V.

**Temporal photoresponsivity and detectivity.** Next, temporal measurements were carried out to extract the switching speed and analyse the responsivity-speed trade-off under varying $V_{SG}$ conditions. Further, the temporal measurements also enable the

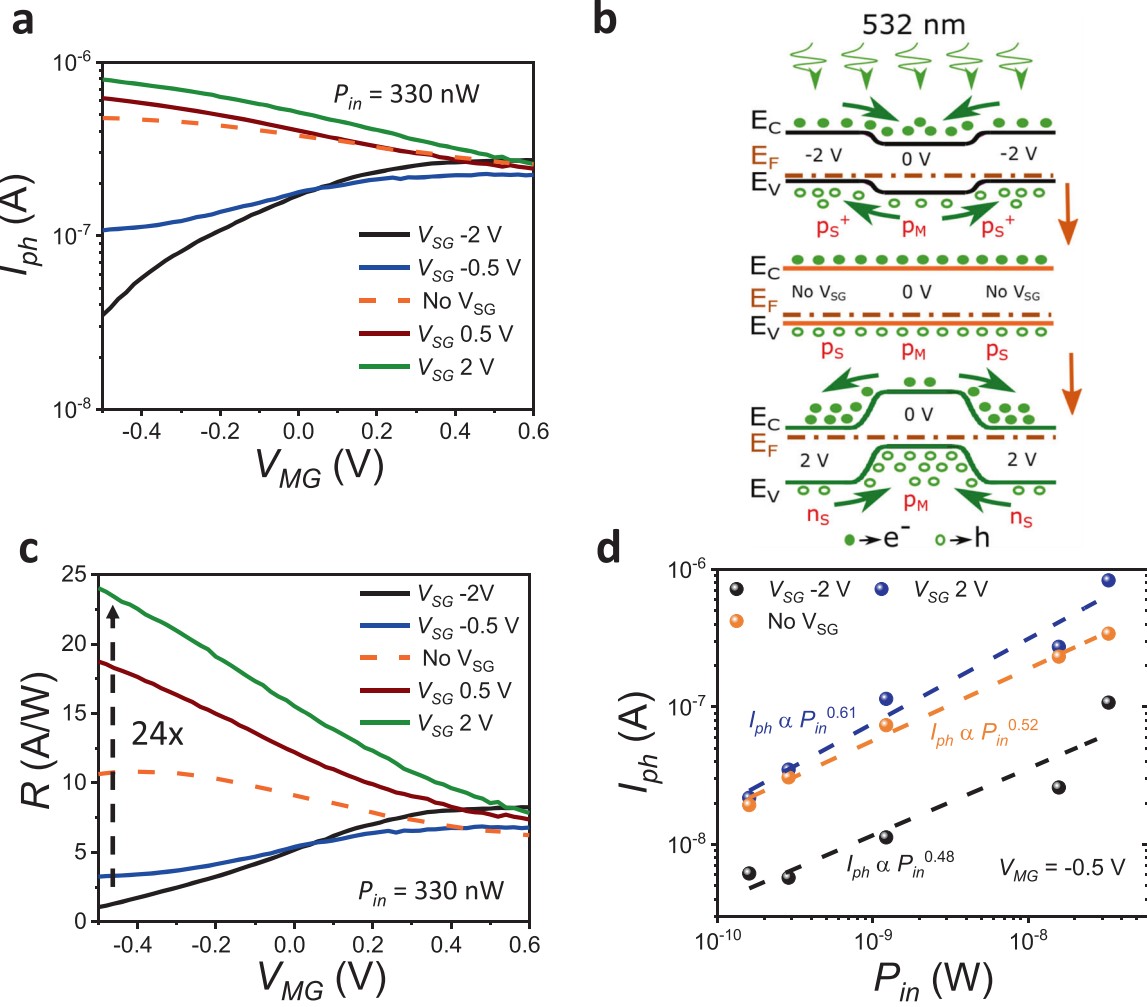

**Fig. 3 Device characterization and carrier dynamics under illumination. a** $I_{ph}$ under light (532 nm laser) vs $V_{MG}$, with $V_{SG}$ changing from −2 to 2 V. An increase in $I_{ph}$ is observed due to charge separation at the lateral p–n junction leading to an inflow of holes into the MG channel. **b** Band diagrams across SG-MG-SG showing efficient (holes towards the channel and electrons towards the SG regions) photogenerated e-h separation at the diode depletion regions, going from $p^+_S$-$p_M$-$p^+_S$ to $n_S$-$p_M$-$n_S$ configuration. **c** Responsivity calculated from $I_{ph}$ shows a 24x increase (for fixed $V_{MG} = -0.5$ V and $V_D = 1$ V) in going from $p^+_S$-$p_M$-$p^+_S$ to $n_S$-$p_M$-$n_S$ configuration along the SG-MG-SG direction. **d** $I_{ph}$ against input optical power shows a sublinear dependency for $V_{SG} = -2$ and 2 V and without $V_{SG}$ indicating the presence of trap states in the WSe$_2$ channel.

extraction of carrier lifetime ($\tau_{life}$), photoconductive gain ($G$), and external quantum efficiency (EQE) values. Representative photoswitching characteristics obtained under 33 nW incident power at 5 kHz are shown in Supplementary Fig. 9a.

We first present and analyse the data under varying $V_{SG}$, fixed $V_{MG}$ conditions as shown in Fig. 4a, b, c. Figure 4a shows $R$ and fall time ($\tau_f$) data for varying $V_{SG}$ at fixed $V_{MG} = -0.5$ V (maximum $R$ and $I_{ph}$ point in the $I_{ph}V_{MG}$ transfer curves of Fig. 3a). $R$ increases from 5.5 to 17 A/W with $V_{SG}$ increasing from 0 to 2 V and, as expected from the $R$-$\tau_f$ trade-off, $\tau_f$ increases from 14.6 to 15.8 μs. These trends are explained through Fig. 4b which depicts two cases- (1) $p_S$-$p_M$-$p_S$ ($V_{SG} = V_{MG} = -0.5$ V) and (2) $n_S$-$p_M$-$n_S$ ($V_{SG} = 2$ V, $V_{MG} = -0.5$ V). In case (2), lateral $n_S$-$p_M$ and $p_M$-$n_S$ band bendings across the SG–MG–SG direction form reverse-biased p–n junctions between MG and the SG WSe$_2$ regions. Hence, as explained in detail in the Supplementary Information (note 8), the equilibrium minority electron concentration in dark condition for case (2), $n_2$, is less than in case (1), $n_1$ (as explained in Fig. 4c), due to a depletion of electrons near the depletion edge in the MG region under reverse bias. Under illumination, these band bendings cause excess photogenerated holes to flow into the S/D channel (opposite for electrons) from

both sides and give rise to (i) a very high density of photogenerated holes, and (ii) a large difference in electron and hole concentrations in the S/D channel. These factors decrease the e-h recombination rate ($R_{SRH}$ in Fig. 4c) in case (2) when the laser is turned off, leading to high responsivity but slower speed. In case (1), lateral injection of holes and ejection of electrons is not possible due to a lack of band bending. Hence, compared to case (2), the transistor switches faster but with lower responsivity. Figure 4b plots $R$ vs $\tau_f$ trade-off indicating a slope of 9.8 A/Ws.

Next, we present and analyse the conventional case of varying $V_{MG}$ without any applied $V_{SG}$ (No-$V_{SG}$) in Fig. 4d, e. Figure 4d shows $R$ and $\tau_f$ data for $V_{MG}$ varying from −0.5 (ON, heavily p-doped channel) to 0.4 V (OFF, low n-doped channel). $R$ decreases as $V_{MG}$ increases towards 0.4 V from −0.5 V. $\tau_f$ also decreases indicating faster switching. These trends, consistent with published reports[8,9,21], are explained in detail in Supplementary note 9, and band diagram schematics under light for $V_{MG}$ at (1) −0.5 V, and, (2) 0.4 V are shown in Supplementary Fig. 11. Figure 4e plots the $R$ vs $\tau_f$ trade-off indicating a slope of 4.2 A/Ws.

Comparison of Figs. 4b and 4e indicates a 2x increase in the $R$ vs $\tau_f$ slope (9.8 vs 4.2 A/Ws) when $V_{SG}$ is applied. This indicates that the lateral p–n junction enables higher $R$ for the same speed

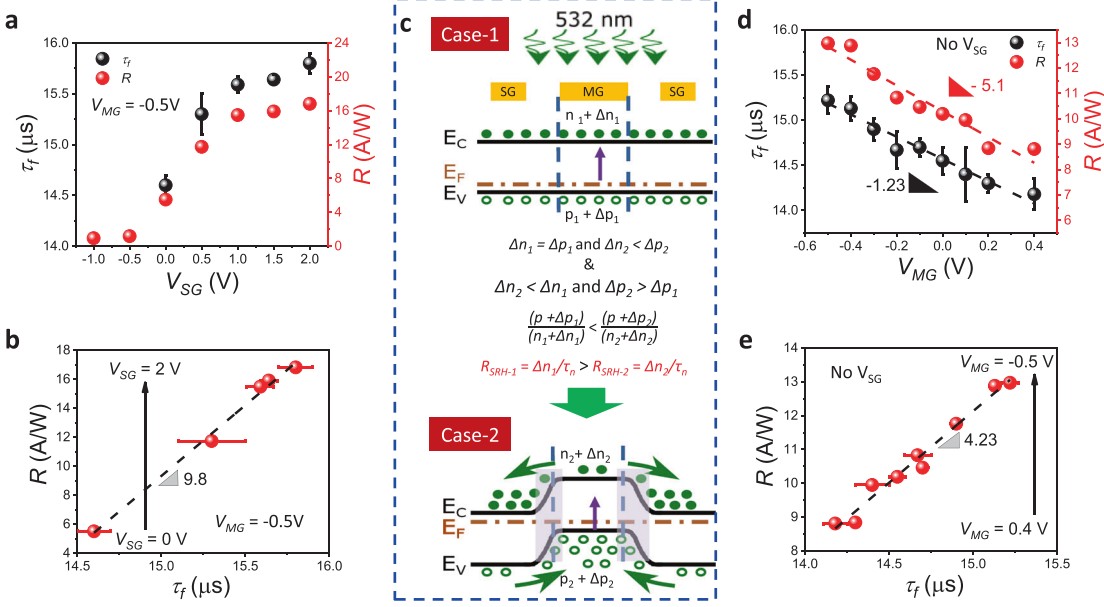

**Fig. 4 Temporal photoresponse of the WSe$_2$ phototransistor.** For fixed $V_{MG} = -0.5$ V and varying $V_{SG}$ (**a**) Responsivity ($R$) and fall time ($\tau_f$) data. **b** $R$ vs $\tau_f$ trade-off indicating a slope of 9.8 A/Ws. **c** Schematic band diagrams depicting recombination rate ($R_{SRH}$) dependence on carrier concentrations for (1) $p_S$-$p_M$-$p_S$ ($V_{SG} = V_{MG} = -0.5$ V) and (2) $n_S$-$p_M$-$n_S$ ($V_{SG} = 2$ V, $V_{MG} = -0.5$ V) configurations. More charge (e-h) separation under $n_S$-$p_M$-$n_S$ configuration leads to slower recombination and correspondingly higher lifetime, gain, $R$, and $\tau_f$. For varying $V_{MG}$ and No-$V_{SG}$. **d** $R$ and $\tau_f$ data. **e** $R$ vs $\tau_f$ trade-off indicating a slope of 4.2 A/Ws.

of operation, a better $R$ vs $\tau_f$ trade-off as compared to conventional means of varying $V_{MG}$, trap density, and other physical[8] or chemical[9] processes. It is worth noting that the modulation in $R$ at 5 kHz is smaller than under DC operation ($R$ and $D^*$ as a function of operating frequency are shown in Supplementary note 11). This is due to the absence of photogating in the MG, SG, and MG-SG depletion regions at high frequency (discussed in Supplementary note 12)[22,23]. Reduced photogating results in less efficient photogeneration of holes due to a higher recombination rate. The reduction in photogenerated hole concentration in the MG region, as well as in the number of holes travelling from SG regions to the MG area, results in lower $I_{ph}$ and $R$ under AC operation. Nevertheless, increasing $V_{SG}$ allows an additional knob, besides reduced trap density, for increasing $R$ under AC operation through enhanced photogenerated hole movement into the MG region. Further, the rise or response time ($\tau_r$) reduces from 23.8 to 21 μs, when $V_{SG}$ is increased from 0 to 2 V with $V_{MG} = -0.5$ V (see Supplementary Fig. 9b). $\tau_r$ represents the time required for the photogenerated carriers to reach steady-state by trapping and de-trapping in the energetically distributed trap states within the WSe$_2$ bandgap inside the WSe$_2$ channel and at the WSe$_2$/hBN interface[24]. At $V_{SG} = 2$ V the photogenerated channel hole concentration is much higher than at $V_{SG} = 0$ V. Hence the available trap states get filled up faster to reach a steady state. Additionally, a monotonic increase in $\alpha$ with $V_{SG}$ (see Supplementary Fig. 8b) indicates decreasing an effective number of participating traps, thereby reducing $\tau_r$. Because of both these reasons, $\tau_r$ is the lowest for $V_{SG} = 2$ V.

Supplementary note 10 details the methodology for calculating carrier lifetimes ($\tau_{life}$), transit times ($\tau_{transit}$), photoconductive gain ($G$), and EQE. Figure 5a shows that $\tau_{life}$ and $G = \tau_{life}/\tau_{transit}$ increase when $V_{SG}$ is increased from 0 to 2 V at fixed $V_{MG} = -0.5$ V. This is due to reduced recombination of e-h pairs as the device configuration changes from $p_S$-$p_M$-$p_S$ to $n_S$-$p_M$-$n_S$. Figure 5b shows a much smaller increase in $G$ for roughly the same change in $\tau_{life}$ as in Fig. 5a, when $V_{MG}$ is decreased from 0.4

to $-0.5$ V without $V_{SG}$. This observation is consistent with the larger change in $R$ for the same change in $\tau_f$ with varying $V_{SG}$ as compared to varying $V_{MG}$ (Fig. 4b, e). Although the mobility is relatively low for this device, the high responsivity reported in this study can be attributed to high photoabsorption and photogating effect in WSe$_2$, as reported for other TMD-based phototransistors[25–27]. Higher mobility with lower S/D resistance via S/D contact engineering could further improve the $\tau_{transit}$, $G$, and $R$ values. Figures 5c and 5d show EQE vs $V_{SG}$ and EQE vs $V_{MG}$ respectively. A maximum EQE of 21.2% is obtained at $V_{SG} = 1$ V for $V_{MG} = -0.5$ V, similar to the No-$V_{SG}$ case, since the lateral p–n junction does not affect the fundamental photon absorption and photogeneration processes.

Figure 6a demonstrates low power photoswitching data for measurements carried out for $V_{SG} = 2$ V and No-$V_{SG}$ at 1 Hz frequency for a laser power of 0.25 pW. Clear switching behaviour was observed for $V_{SG} = 2$ V, unlike the No-$V_{SG}$ case, reinforcing the benefit of a lateral p–n junction in enhancing $D^*$ of the phototransistor. $D^*$ is another important parameter that represents the detector's ability to measure signals with reference to its noise level. Here in this study $D^*$ is calculated using

$$D^* = \frac{\sqrt{A}}{\text{NEP}} \tag{2}$$

In the above equation, $A$ is the channel area over the MG. NEP is the noise equivalent power, which is extracted from the measured noise power spectral density ($S$) data at 1 Hz shown in Fig. 6b and $I_{ph}$ vs $P_{in}$ plots (shown in Supplementary Figure 15). The $S$ data was determined from the Fourier transform of the dark current time traces measured for fixed drain and varying $V_{MG}$ and $V_{SG}$ voltages, on a similar device (see Supplementary note 14). Figure 6c shows the comparison of $D^*$ for No-$V_{SG}$ and $V_{SG} = 2$ V, at $V_{MG} = +/- 0.4$ V (OFF/ON conditions). A 25x enhancement in $D^*$ is obtained for $V_{SG} = 2$ V, at $V_{MG} = 0.4$ V. This can be attributed to a substantial reduction in NEP due to i) enhanced $R$ (~1.45x) resulting from efficient photogenerated hole

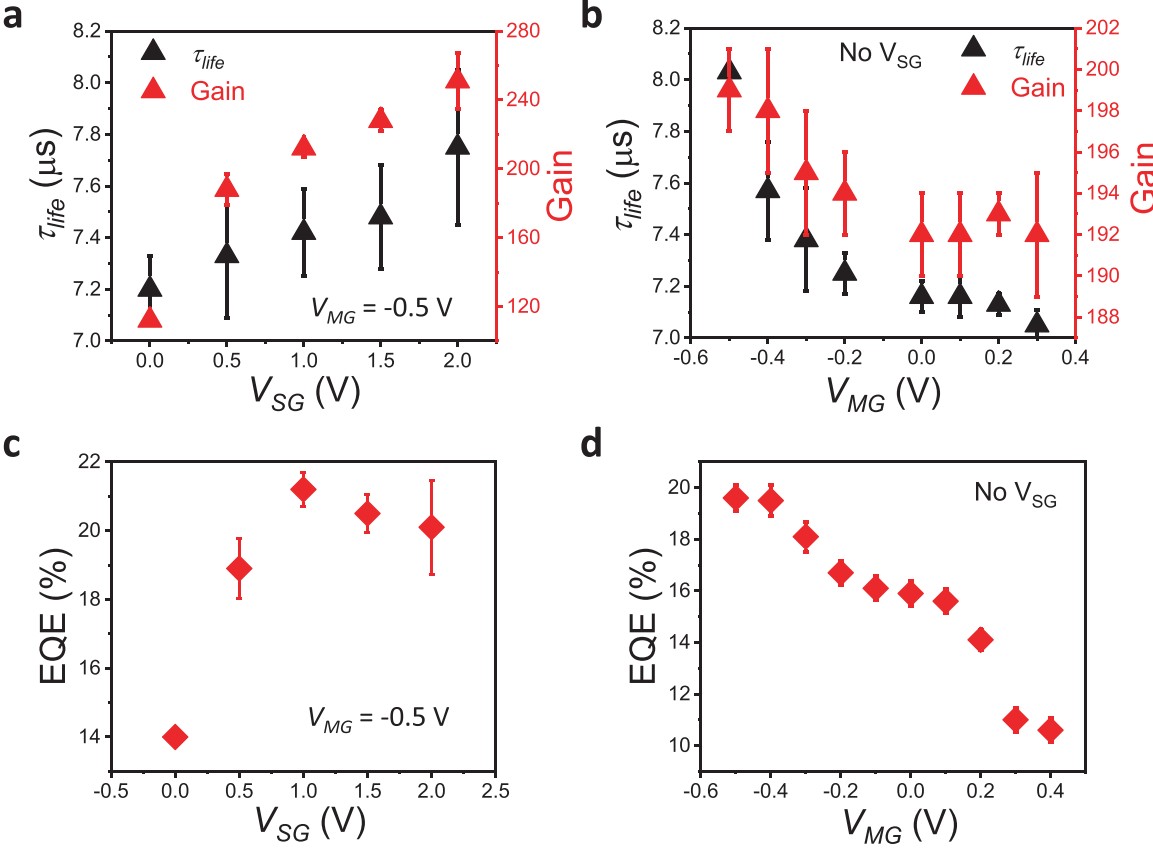

**Fig. 5 Gate voltage dependence of gain, carrier lifetime and EQE of WSe₂ phototransistor. a** Carrier lifetimes ($\tau_{life}$) extracted from photoconductive decay fits and calculated photoconductive gain (G) are plotted against $V_{SG}$ for fixed $V_{MG} = -0.5$ V. Minority electron ejection from and majority hole injection into the S/D channel under $n_S$–$p_M$–$n_S$ configuration increases e-h recombination time thereby increasing $\tau_{life}$ and G (nearly 2x) vs the $p_S$–$p_M$–$p_S$ case. **b** $\tau_{life}$ and G vs $V_{MG}$, without $V_{SG}$. The increase in G is much smaller compared to the increase seen for varying $V_{SG}$, for nearly the same change in $\tau_{life}$. **c** EQE vs $V_{SG}$ for $V_{MG} = -0.5$ V. **d** EQE vs $V_{MG}$ without $V_{SG}$. Maximum EQE of 21.5 % is obtained for $V_{SG} = 1$ V, $V_{MG} = -0.5$ V. The error bars indicate variability (s.e) in photoswitching data.

separation, as well as, ii) decrease in $I_{dark}$ (greater than 10x, see Supplementary note 14) resulting in a decrease in S by nearly an order of magnitude[12,13] at 1 Hz for $V_{SG} = 2$ V when compared to No-$V_{SG}$. The reduction in NEP due to both these factors increases $D^*$. A maximum $D^*$ value of $1.1 \times 10^{12}$ Jones is obtained.

Finally, Figs. 6d and 6e benchmark responsivity and $D^*$ values vs speed respectively, for the WSe₂ phototransistor reported in this work with other single 2D material-based (such as MoS₂, MoSe₂, WSe₂, InSe, SnS₂, etc.) phototransistors that report the values of all three parameters ($R$, $D^*$ and $\tau_f$). A clear trade-off between $R$ and $\tau_f$ is observed over a wide range of $R$ ($10^{-3}$ A/W to $10^5$ A/W) and $\tau_f$ (μs to seconds) (Fig. 6d)[10,11,28–46]. When $R$ is high, $\tau_f$ also tends to be very high[10,28] indicating slow detector speed. On the other hand, fast speed (low $\tau_f$) phototransistors show lower $R$[37,40]. Our work shows the benefit of high $R$ without slowing down the phototransistor by employing a lateral p–n homojunction. Figure 6e benchmarks both flicker noise- and shot noise-limited (theoretical maximum) $D^*$ vs $\tau_f$ values for the same set of reports as shown in Fig. 6d. With high flicker noise-limited ($1.1 \times 10^{12}$ Jones) and shot noise-limited ($5 \times 10^{13}$ Jones) $D^*$, high $R$ (94 A/W) and low $\tau_f$ (14 μs) data, this work reports one of the best combination of $R$, $D^*$ and $\tau_f$ values, enabled by a lateral p–n junction.

## Discussion

This work demonstrates a high-performance all-2D layered materials (WSe₂/hBN) based phototransistor with a responsivity

of 170 A/W at 300 pW laser power, and μs speed of operation, along with the flicker noise-limited measured specific detectivity of $1.1 \times 10^{12}$ Jones. These high-performance metrics are enabled by a facile electrostatic doping technique that employs a lateral p–n homojunction to control photocarrier population and the width of the conducting channel thereby improving the fundamental responsivity-speed trade-off by nearly 2x. The combined effect of enhanced responsivity and reduction in dark current also improves the detectivity as seen in the ability to detect sub-1 pW laser switching with the help of the lateral p–n junction. From a broader perspective, this study demonstrates a simple optoelectronic device architecture for achieving high responsivity and detectivity without compromising speed, which can be realized in any optically active 2D layered or thin-film semiconductor material that is amenable to electrostatic doping. Further, a lateral heterojunction device with different n and p-type materials could additionally enable a broader spectral response and wavelength selectivity to achieve enhanced performance. This can have promising applications in integrated photonics and optoelectronic devices.

## Methods

**Device fabrication.** A single side polished 4" p⁺-Si wafer with 285 nm SiO₂ grown on top was used as the substrate. Side gates (SG) and a middle gate (MG) were patterned first on top of the Si/SiO₂ wafer by electron beam lithography (EBL, Raith 150-Two). This was followed by metal (Cr 2 nm/ Au 30 nm) sputtering (AJA, ATC sputter system) and lift-off. The length and width of the MG channel were 14 and 2 μm respectively. The MG had 3 μm long, 1 μm wide and 3 μm apart leads

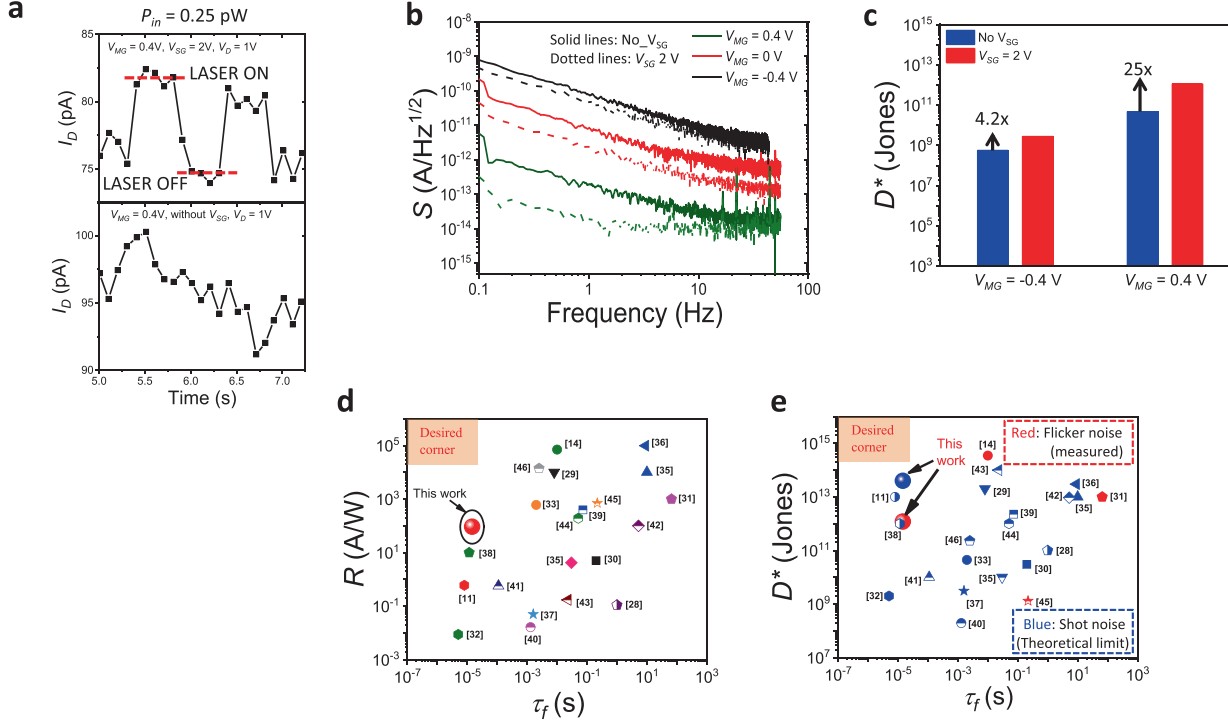

**Fig. 6 Measured noise, detectivity, and benchmarking of WSe₂ phototransistor. a**. Photoswitching observed for $V_{SG}$ = 2 V at an ultra-low laser power of 0.25 pW, unlike the No-$V_{SG}$ case. **b** Noise power spectral density ($S$) from flicker noise measurement for No-$V_{SG}$ and $V_{SG}$ = 2 V shows lower noise floor (by nearly 10x) for $V_{SG}$ = 2 V. **c** Specific detectivity ($D^*$) at $V_{MG}$ = ± 0.4 V for No-$V_{SG}$ and $V_{SG}$ = 2 V shows 25x enhancement in $D^*$ with $V_{SG}$. Benchmarking plots of (**d**) $R$ and (**e**) $D^*$ vs fall time ($\tau_f$).

extending on either side. A total of six such leads (three on each side) were fabricated on both sides of the MG channel to form an inter-digited structure with similar SG leads. The separation between the MG and SG leads was 1.5 μm. Bulk and single-crystal WSe₂ and hBN were purchased from SPI supplies. An hBN flake was exfoliated using the micro-mechanical exfoliation technique with 3 M magic scotch tape. The hBN flake was then transferred from the scotch tape onto a polydimethylsiloxane (PDMS) stamp. The PDMS stamp was fixed onto a glass slide and the glass slide was attached to a micromanipulator. The hBN was transferred selectively on top of the pre-fabricated MG and SG structure using the micro-manipulator under an Olympus BX-63 microscope. During the transfer process, the Si/SiO₂ substrate with its MG and SG pattern was kept on top of a microheater. After the hBN was aligned and placed on top of the SG and MG pattern, the entire structure - Si/SiO₂ substrate and the hBN flake along with the PDMS stamp and the glass slide, was heated to 60 °C to weaken the adhesion between the PDMS stamp and hBN. The temperature was then allowed to come down to 50 °C to release the glass slide and the PDMS stamp from the Si/SiO₂ substrate, so that only hBN stays on top of the MG and the SG pattern. The same transfer process was followed for aligning and placing WSe₂ on top of hBN. The WSe₂ flake was placed in such a way, that it covered the entire MG and SG area without touching the extended MG leads going to the source/drain contact pads. Next, source and drain contacts were patterned using EBL on top of WSe₂ with an MG overlap such that the actual, physical source-to-drain channel length remained at ~10 μm. Finally, source/drain sputter metallization (Cr 2 nm/ Pt 30 nm/ Au 80 nm) and lift-off were carried out to complete the device fabrication. On a separate sample (Supplementary Information Fig. 3a), two additional lateral contacts on top of the WSe₂ transistor, and overlapping with the SG area, were fabricated along with the source/drain contacts, to demonstrate a p–n junction between the source (or drain) and SG contacts.

**Device characterization**. Before optoelectrical characterization, the device was placed on a PCB with gold contact pads on it. It was then wire bonded using gold wire from the device contact pads to the large PCB contact pads using a wedge bond system. All electrical measurements were done in ambient conditions under a BX-63 Olympus microscope using a Keysight B1500A semiconductor device analyzer. The photo-response measurements were carried out using a 532 nm fiber-coupled diode laser. Steady-state photo-response was measured using Keysight B1500A and the temporal response was measured using a 4 GHz Keysight DSOS404A oscilloscope and current probe (Keysight 2825A). Input laser power was modulated from off to on using square pulses of 1 V peak-to-peak to the laser power controller unit from an Agilent 33220A function generator. The laser was incident on the device through the objective lens of the BX-63 Olympus

microscope. Input optical power was varied by inserting suitable optical density filters from HOLMARC in the path of the 532 nm laser beam.

## Data availability
All relevant data that support the findings of this study are available from the corresponding authors upon request.

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

## Acknowledgements

The authors thank Shankar Kesarwani for wire bonding the samples and Himani Jawa for her contribution in device fabrication and simulation. The authors also thank Prof. Ageeth Bol and Prof. Peter Steeneken for useful discussions. A.V. thanks the IITB-Monash Research Academy for his fellowship. K.T. acknowledges the Visvesvaraya PhD Scheme from the Ministry of Electronics and Information Technology (MeitY), Govt. of India for his fellowship. The authors acknowledge the Indian Institute of Technology Bombay Nanofabrication Facility (IITBNF) for the usage of its facilities for device fabrication and characterization. This work was funded by the Department of Science and Technology, Govt. of India through the grant DST/SJF/ETA-01/2016-17 and is part of the project 'Agricultural and environmental 2D gas sensors' with project number 483.20.029 of the research programme "Science for Diplomacy" which is (partly) financed by the Dutch Research Council (NWO).

## Author contributions

S.G. and S.L. conceived the idea. S.G., A.V., and S.D. created the experiment plan and fabricated the WSe$_2$ phototransistors. S.G., A.V., and S.D. carried out the steady-state and temporal photoresponse measurements. K.T. did the AFM measurements for the hBN and WSe$_2$ flakes and participated in technical discussions. S.G. and K.T. simulated the device structure. S.G., K.T., and S.L. analysed the simulated data. S.G. and S.L. analysed the experimental data and benchmarked it against previous reports. All authors contributed to the technical discussion and S.G. and S.L. wrote the manuscript.

## Competing interests

The authors declare no competing interests.
