## [Peer Review File · Nature Communications]

Reviewers' comments:

Reviewer #1 (Remarks to the Author):

The co-authors demonstrate an interesting photodetector structure which shows obvious attenuation of the trade-off between photoresponsivity and response speed. I think the novelties of this work is 1) the device structure and 2) the high responsivity plus the fast response time.

However, these novelties are questionable because

In general, the R under positive VSG is c.a. 3 times than that under zero VSG. From the optical microscope (Fig. 2b inset), the total area (2 side gates + 1 middle gate) is roughly 4 times than the middle gate area. It seems that the improved R arise from the collection of photo carriers generated from the side-gate area, which agrees with the picture shown in Fig. 3b. In this regard, why not simple employ a wider FET channel by depositing one piece of big gate electrode and wider S/D electrodes?

In Supplementary Fig. 4a, temporal ID is measured under $VMG = -0.5\text{ V}$, $VSG = 2\text{ V}$, $VD = 1\text{ V}$, $P = 330\text{ nW}$. Under this condition, R is expected to be 24 A/W according to Fig. 3c. Namely, ID should be $330\text{ nW} * 24\text{ A/W} = 7.92\text{ }\mu\text{A}$. However, the ID in Supplementary Fig. 4a is only c.a. 170 nA . Does this mean that the ID in Supplementary Fig. 4a is far from saturated? Namely, the response time regarding to $7.92\text{ }\mu\text{A}$ should be much longer?

Despite the above two problems, their physical picture is unclear and some of their results is not clearly described. Therefore I suggest not publishing this manuscript unless the above two issues and the following issues are addressed.

The authors claim that $VSG=0\text{ V}$ is equivalent to floating VSG. Indeed this argument is supported by Supplementary Fig. 3a which shows similar transfer characteristics under the two conditions. However, this picture is incorrect because of the following arguments.

When VSG is floating, we can reasonably conclude $VSG = VMG$ when assuming WSe₂ is ideal conductor.

I have to presume in this work the source electrode is grounded as in other literatures. Namely, under $VSG = 0\text{ V}$ condition, $VSG = VS$.

Based on b, when $VSG = 0\text{ V}$, $VSG \neq VMG$ when measuring the transfer characteristics except for one data point ($VMG = 0\text{ V}$).

I have to presume $VSD = 1\text{ V}$ in the results of Supplementary Fig. 3a, which is the same as other transfer characteristics in this manuscript.

In Fig. 2d, the authors did not describe how they varied and calculated $pWSe_2$. Was it varied by applying different VD?

In Fig. 4b, First, the authors defined recombination rate as $R_{SRH} = \Delta n / \tau$. This definition is different from the regular definition $R = 1 / \tau$. Next, the authors claim that $(p + \Delta p_1) / (n + \Delta n_1) < (p + \Delta p_2) / (n + \Delta n_2)$. This argument is based on another assumption that (in dark, $n_1 = n_2$ and $p_1 = p_2$) under the two conditions. This is an acceptable but not strict assumption because when changing the applied bias, the fermi level that determines n could be changed. The authors should clearly address this assumption.

In Page 8, the authors argued "...the number of traps participating in the photocurrent transport along the S/D direction also decreases. This ... is one of the factors behind lower responsivity (photoconductive gain) at $VSG = -2\text{ V}$." Please explain why.

In Page 9, "there is no significant imbalance in photogenerated electron and hole concentrations, leading to low τ life and hence low G and R. " Please explain why.

In 2nd paragraph of Page 10, the authors compared Fig. 4c with Fig. 4f. However, the results from the two figures are acquired under two experiment configurations. In Fig. 4c, VMG is controlled while VSG is changeable while VSG is controlled while VMG is changeable. Is this comparison fair?

In Fig. 5a, τ_{life} is plotted against V_{SG} . In this result, what is V_{D} ? Maybe $V_{D} = 1\text{ V}$? At least, I assume it should be constant. Then, my next question is about the gain ($G = \tau_{\text{life}} / \tau_{\text{transit}}$ in which $\tau_{\text{transit}} \propto 1 / V_{D}$). Suppose V_{D} is constant, why G and τ_{life} donot coincide with each other in Fig. 5a?

As shown in manuscript, there are three gates in a device including one middle gate and two side gates. However the shape of the side gates is sometimes interdigitated and sometimes rectangle (Fig. 2b). What's difference between them?

In Fig 2a inset, the Schottky barrier-dominated transport behavior across the S/D contacts was shown. Why an obvious asymmetric electrical phenomenon appears in the device with symmetric

S/D contacts? What is the difference between the S and D electrodes?

The performance comparison between the two types of devices was shown in Fig 4. Actually, we can see when the fall time is 14.5 μ s, the R in the device without applied side gate is higher than the fixed side gate and middle gate one; Even in the time interval from 14 to 16 μ s that the authors claimed in the abstract, the R is similar in the two types devices (Fig 4c and f). Therefore, the author should clearly determine the advantage for the design of three lateral in-plane gate electrodes than that of only one middle gate.

Other problems to be checked carefully include

Typo: "WSe2" in Fig 2d;

The SiO₂ layer thickness is claimed to be 285 nm in Methods and 280 nm in Results. Which is correct?

Reviewer #2 (Remarks to the Author):

This paper reports on the fabrication and characterization of npn and p+pp+ phototransistors enabled by gate tunable homojunctions formed in WSe₂ channels.

The work presents a thorough characterization of the devices and the paper is well written.

However I think that in its current form the paper is not suitable for publication because the claims are not supported by the data.

In particular:

1. The authors state on page 1 that the noise of the detectors is determined by the dark current. This is only partially true because such detectors are known to be severely limited by 1/f noise. Then in all the calculations of the D* the authors assume the shot noise limit, they actually do not measure the noise current. Therefore the claims of such high D* values are not supported. Related to this the minimum detectable power that their detectors can reach is 300 pW, this taken together with the area of the detector of approx. 10 μ m x 10 μ m, through the equation of $D^* = \sqrt{(\text{Area} \times \text{BW})/\text{NEP}}$, yields a D* of 1e7 Jones, about 7 orders of magnitude lower than the claimed one.
2. In Fig 6 the authors plot the D* as a function of incident power, this is meaningless, because D* defines the minimum detectable power of the detector and should be reported only as a function of temperature, frequency, wavelength, applied bias.
3. The spectral response of the photodetectors is missing.
4. I would like to see some control experiments to strongly prove that the use of those homojunctions npn is really necessary to achieve high performance. What happens for example if you would have only one heterojunction across the channel?
5. The authors make use of hBN a material that at the moment is not scalable, why is this needed? What is the performance of those structures by replacing hBN with another dielectric, through a scalable process?

Reviewer #3 (Remarks to the Author):

In this manuscript, the authors explored the photo responsivity, speed, and sensitivity of a WSe₂ phototransistor. While the ideas presented by the authors are interesting, there are a few remaining issues that need to be addressed.

1. A WSe₂ phototransistor is explored under the gate voltages ranging from -2V and 2V; therefore, it is important to know its electrical transport properties in the same range. In other words, it would be necessary to extend the I_d curvatures as a function of V_{mg} to (-2V – 2V) in Fig. 2a and 2b and to calculate both the hole and electron mobility. Since the channel mobility play an important role in both photo responsivity and speed, it would be interesting to know whether electron and hole have different mobility and whether the mobility is important in the following experiments (different "cases").
2. The hole mobility of the device in this manuscript is about 5 cm²/Vs, which is lower than other

reports. As a result, the device is expected to have relatively low photo responsivity and long response time. Some discussion would strengthen the manuscript.

3. From the schematic diagram in Fig. 1a, the WSe₂ channel between the source and drain electrodes mainly depends on the middle gate, which is confirmed by Fig. 2b. When the WSe₂ channel is in the ON state, the side gates have limited influence on the electrical transport properties of the channel. When the WSe₂ channel is in the OFF state, the side gates can slightly change the doping of the WSe₂ channel. Therefore, it may be inappropriate to claim it as "a gate tunable lateral p-n junction".

4. Since the two side-gates are short, the photocurrent signals generated on two side-middle interfaces are expected to have opposite polarities. Under a zero source-drain bias, the photocurrent responses of the entire device are expected to be zero. Therefore, it would be interesting to know the photocurrent signals (photo responsivity and speed) as a function of the source-drain bias.

5. "no-Vsg" is a little bit confused. "no-Vsg" means "ground", "floating", or something else.

Response to reviewers' comments including revisions in the main manuscript and supporting information for the manuscript titled "Enhanced responsivity and detectivity of fast WSe₂ phototransistor enabled by electrostatically tunable in-plane P-N homojunction"

Based on the reviewer's comments the revised manuscript and supplementary information now addresses all the concerns raised by the reviewers and incorporates their suggestions. We have made the following key changes in this submission:

- a. Thorough justification of the advantage of our device architecture in mitigating the responsivity-speed trade-off by incorporating:
 - new text in the main manuscript
 - additional section in supplementary information (section 5) comprising new data, calculations and schematics along with explanations (reviewers 1 and 2)
- b. New band diagrams and analysis of carrier dynamics (supplementary information, section 8) to improve the physical picture responsible for the enhanced performance- high responsivity at high speed (reviewers 1, 2 and 3)
- c. Updated figure showing detectivity as a function of side-gate bias for DC as well as AC operating conditions (Figure 6b, main manuscript) to demonstrate performance tunability (reviewer 2)
- d. New figure and calculations for estimation of detectivity using shot noise and thermal noise in supplementary information (section 10) (reviewers 2 and 3)
- e. New section with figures and associated text on frequency-dependent responsivity and detectivity data in the supplementary information (section 6) highlighting differences under DC and AC operations (reviewers 1 and 2)
- f. Additional band diagrams to provide good understanding of the variation of response time under DC and AC operations in supplementary information section 9 (reviewer 1)
- g. Improved explanations on the contribution of traps and related photogating effect on the device operation (reviewer 1)
- h. Overall clarity of the text in the manuscript and supporting information has been enhanced (reviewers 1, 2 and 3)

In the revised manuscript and supporting information, the text changes have been shown in **red**. Further, modified figures have also been marked with a **red bounding box**. In response letter the responses have been shown in **red** and the changes in main manuscript and Supplementary Information have been shown in **blue**.

Reviewer 1

The co-authors demonstrate an interesting photodetector structure which shows obvious attenuation of the trade-off between photoresponsivity and response speed. I think the novelties of this work is 1) the device structure and 2) the high responsivity plus the fast response time. However, these novelties are questionable because

1. In general, the R under positive V_{SG} is c.a. 3 times than that under zero V_{SG} . From the optical microscope (Fig. 2b inset), the total area (2 side gates + 1 middle gate) is roughly 4 times than the middle gate area. It seems that the improved R arise from the collection of photo carriers generated from the side-gate area, which agrees with the picture shown in Fig. 3b. In this regard, why not simple employ a wider FET channel by depositing one piece of big gate electrode and wider S/D electrodes?

We thank the reviewer for the comments and highlighting the novelty of this study. The schematics below in Fig. 1a and b show the three-gate (SG-MG-SG) structure alongside the reviewer's proposed structure.

Fig. 1: Device architecture and carrier conduction path from source to drain are shown for a) our SG-MG-SG and b) single extended gate (as per reviewer's proposal) FET structures.

The proposed structure is worse than our device for the following reasons:

(i) **Lack of Electrical Tunability:** In the SG-MG-SG architecture, the SG is employed specifically to widely tune the device performance. Such wide tunability is not possible in the wide-gate FET structure. This performance tunability is important in the following ways,

- a) We can use the multi-gate device for high responsivity (to operate even at high intensity input optical power) [$V_{SG}= 2$ V, $V_{MG}= -0.5$ V] as well as,
- b) at high detectivity (to capture weak optical signal) [$V_{SG}= 2$ V, $V_{MG}= 0.4$ V], as per requirement.

Depletion increase for positive V_{SG} applied, not only helps in accumulation of holes in the MG channel, but also reduces the number of electrons in conduction band cause less recombination. Thereby it results in an increment in α value. Hence, this tunability under positive V_{SG} leads to less recombination of photo-generated carriers giving an advantage in responsivity.

(ii) Higher OFF current: In comparison to the SG-MG-SG architecture, the FET channel area in case of the single wide-gate device will be larger. Hence the dark state current in ‘ON’ or ‘OFF’ state will be higher. This will result in poor detectivity.

(iii) Non-scalability of R with Area: Next, here we have tried to estimate the change in responsivity between only MG and extended SG-MG-SG single FET structure. As per the figure 1 above, if we consider the MG channel length = L and width = W, then area under the middle gate (which takes part in S/D transport), $A = W * L$.

Now, according to the reviewer, if we consider a larger area single FET with a width of 4W (as the total width including SG-MG-SG is approximately 4 times more than MG width), then the area of the larger FET will be $A' = 4W * L = 4A$.

Considering extended S/D contacts as well, with the same voltage conditions (V_D and V_{MG}) the current (dark and light) in wider FET will be 4 times higher than the FET with only MG.

Now, as we know, responsivity, $R = I_{ph} / (P_{in} * (A_{device} / A_{laser}))$,

Then, for only MG device, $R1 = I_{ph} / (P_{in} * (A / A_{laser}))$, and

For extended FET, $R2 = 4 * I_{ph} / (P_{in} * (4 * A / A_{laser})) = R1$. So fundamentally, we don't expect any change in R for extended FET and FET only over MG.

Similarly, specific detectivity is expressed as, $D^* = R * \frac{A_{device}^{1/2}}{S_n}$

S_n from shot noise is expressed as, $S_n = 2qI_{dark}^{1/2}$

Then for only MG device operation, $S_{n1} = 2qI_{dark}^{1/2}$ and

for extended FET, $S_{n2} = 2q(4 * I_{dark})^{1/2} = 2 * S_{n1}$

Then, $D^*_1 = R * \frac{A^{1/2}}{S_{n1}}$ and,

for extended FET, $D^*_2 = R * \frac{2A}{S_{n2}} = D^*_1$

(iv) NPN ($V_{SG} = 2 V$) still better than larger effective width ($V_{MG} = V_{SG}$) Furthermore, here we also present a comparison between no- V_{SG} and $V_{MG} = V_{SG}$ conditions, and look at the effect in the device responsivity and detectivity. In case of $V_{MG} = V_{SG}$, effectively the channel area expands, as the whole area under MG and SG is under same bias. Hence, in this case, a good number of carriers (dark or photo) participate in conduction and reach S/D contacts. On the other hand, in case of no- V_{SG} , only the area under MG takes part in S/D conduction, as shown in figure 2 below. Hence the $V_{MG} = V_{SG}$ case presents a close approximation to the reviewer's proposed structure.

Fig. 2: Flow of holes from source to drain under transistor ‘ON’ state is demonstrated for (a) no- V_{SG} , (b) $V_{MG} = V_{SG}$ [approximation to wider FET proposed by reviewer]

Fig. 3: A performance comparison between $V_{SG} = V_{MG}$, no- V_{SG} and $V_{SG} = 2V$ conditions for (a) responsivity and (b) detectivity under $P_{in} = 33$ nW.

Clearly it is observed that although the net responsivity number for $V_{MG} = V_{SG}$ conditions is higher from no- V_{SG} , but the maximum detectivity is much lower. Also, under applied $V_{SG} = 2V$, in three-gate configuration, both responsivity and detectivity is higher than $V_{MG} = V_{SG}$. This shows the advantage of having a separate SG electrical control.

Changes to supplementary information: The following section has been added to the Supplementary Information Section 5.

In this work, the MG FET channel area is less than the total device area. Enhanced photogeneration can be achieved if the total device area (area over SG-MG-SG) were to be considered as a single FET channel (Supplementary Figure 5). However, the three-gate (SG-MG-SG) structure is still superior due to the following aspects,

(i) **Non-scalability of R and D^* with Area:** It is possible to compare the responsivities between the only MG FET and the extended SG-MG-SG single FET structure. As per the Supplementary Figure 5 below, if we consider the MG channel length = L and its width = W, then the area above the middle gate (which takes part in S/D transport), $A = W \cdot L$.

Supplementary Figure 5. Device architecture and carrier conduction path from source-to-drain are shown for a) the SG-MG-SG FET with only MG channel is active, and, b) the single extended gate FET structure.

Now, if we consider a larger area single FET with a width of $4W$ (as the total width including SG-MG-SG is approximately 4 times more than the MG width), then the area of the larger FET will be $A' = 4W * L = 4A$.

Considering extended S/D contacts as well, with the same voltage conditions (V_D and V_{MG}), the currents (dark and light) in the wider FET will be 4 times higher than the FET with only MG.

Now, as we know, responsivity, $R = I_{ph} / (P_{in} * (A_{device} / A_{laser}))$,

Then, for only MG device, $R_1 = I_{ph} / (P_{in} * (A / A_{laser}))$, and,

for extended FET, $R_2 = 4 * I_{ph} / (P_{in} * (4 * A / A_{laser})) = R_1$. Hence, fundamentally we don't expect any difference in R between the extended FET and the FET only over MG.

Similarly, specific detectivity is expressed as, $D^* = R \times \frac{A_{device}^{1/2}}{S_n}$

S_n from shot noise is expressed as, $S_n = 2qI_{dark}^{1/2}$

Then for only MG device operation, $S_{n1} = 2qI_{dark}^{1/2}$ and

for extended FET, $S_{n2} = 2q(4 * I_{dark})^{1/2} = 2 * S_{n1}$

Then, $D^*_1 = R \times \frac{A^{1/2}}{S_{n1}}$ and,

for extended FET, $D^*_2 = R \times \frac{2A}{S_{n2}} = D^*_1$

(ii) NPN ($V_{SG} = 2V$) FET is still better than larger effective width ($V_{MG} = V_{SG}$) FET

Furthermore, we also present a comparison between no- V_{SG} , $V_{MG} = V_{SG}$ and $V_{SG} = 2V$ conditions, and look at their effect on device responsivity and detectivity. In case of $V_{MG} = V_{SG}$, effectively the channel area expands, as the entire area above the MG and the SGs is under the same gate bias. Hence, in this case, a good number of carriers (dark or photo) participate in conduction and reach the S/D contacts. On the other hand, in case of no- V_{SG} condition, only the area above the MG takes part in S/D conduction, as shown in Supplementary Figure 6 below.

Hence the $V_{MG} = V_{SG}$ case presents a close approximation to the single extended gate FET structure.

Supplementary Figure 6. Flow of holes from source-to-drain under transistor ‘ON’ state is shown for (a) no- V_{SG} , (b) $V_{MG} = V_{SG}$ [approximation to wider, single extended gate FET]

Supplementary Figure 7. Performance comparison between $V_{SG} = V_{MG}$, no- V_{SG} and $V_{SG} = 2V$ conditions for (a) responsivity and (b) detectivity under $P_{in} = 33$ nW.

Although net responsivity for $V_{MG} = V_{SG}$ condition is higher than the no- V_{SG} case as shown in Supplementary Figure 7, the maximum detectivity is much lower. Also, under applied $V_{SG} = 2$ V, in the three-gate configuration, both, responsivity and detectivity are higher than $V_{MG} = V_{SG}$. This shows the advantage of having separate SG electrical control instead of a single gate wide FET structure.

2. In Supplementary Fig. 4a, temporal I_D is measured under $V_{MG} = -0.5$ V, $V_{SG} = 2$ V, $V_D = 1$ V, $P = 330$ nW. Under this condition, R is expected to be 24 A/W according to Fig. 3c. Namely, I_D should be 330 nW * 24 A/W = 7.92 μ A. However, the I_D in Supplementary Fig. 4a is only c.a. 170 nA. Does this mean that the I_D in Supplementary Fig. 4a is far from saturated? Namely, the response time regarding to 7.92 μ A should be much longer? Despite the above two problems, their physical picture is unclear and some of their results is not clearly described. Therefore, I suggest not publishing this manuscript unless the above two issues and the following issues are addressed.

We thank the reviewer for the inputs.

(i) Firstly, the value of input optical power, $P = 33 \text{ nW}$, instead of 330 nW . So, the I_{ph} will be 792 nA . This is in line with the experimental data presented. We sincerely apologize for this unintentional typing mistake. We have changed the value from 330 nW to 33 nW in all places in the main manuscript. This typographical error however does not affect the trends and key conclusions of the manuscript.

(ii) The two values of I_{ph} - under DC ($\sim 792 \text{ nA}$) and under 5 kHz frequency of operation ($\sim 170 \text{ nA}$) differ by around $4.7\times$. The I_{ph} at 5 kHz is away from saturation. A similar difference in I_{ph} under DC and at high-frequency operation is observed in literature (see examples below); due to a dominant photogating (due to trapping of photo generated carriers) effect under DC and absence of the same at high frequency.¹⁻²

Fig. 4: Frequency dependent I_{ph} shows I_{ph} decrease with an increase in frequency. These references are taken from, (a) M. M. Furchi, *ACS Nano Lett.*, 2014 and (b) H. P. Hsu, *Materials*, 2019.

(iii) In our SG-MG-SG device architecture too, we have performed additional photoswitching measurements for varying frequency (0 to 5 kHz), under $V_{\text{MG}} = -1.5 \text{ V}$ (ON state) and $V_{\text{SG}} = 1 \text{ V}$ and at $P_{\text{in}} = 75 \text{ nW}$. Normalized R vs frequency data in Figure 5a below shows a similar decrease in R as we move to higher frequencies. This can happen due to the following reason.

The MG channel region can have fast as well as slow electron traps. The slow electron traps near the conduction band edge may not be able to respond at 5 kHz frequency, hence there is reduced trapping of photogenerated electrons. Consequently, there is a reduction in photogating and gain with increasing frequency leading to lower photocurrent under AC (5 kHz) conditions.

Fig. 5: Frequency dependent photoswitching characteristics. **a** Normalized R and **b** normalized D^* vs frequency. R and D^* have been normalized to their DC values (at 0 Hz). Consistent with data shown in Fig. 4 (from response letter), both R and D^* show a monotonic decrease with frequency.

Figure 6 illustrates photogenerated carrier dynamics under DC and high frequency operation.

Fig. 6: Carrier dynamics for photogenerated electrons and holes are shown through band diagrams under (a) DC mode of operation (device is illuminated under a constant light source of 532 nm laser) and (b) under modulated light intensity switching from ON to OFF at high frequency (say, 5 kHz).

Furthermore, it is important to note that 1) the I_{ph} at 5 kHz operating frequency is lower than the I_{ph} under DC operation, and 2) the fall time is limited by the rate of recombination as discussed in the main manuscript on page 9 in the discussion on Fig. 4c. Therefore, the response time (fall time) should be faster in the case of AC operation.

Changes to the supplementary information: The following section has been added to the supplementary information section 6.

Supplementary Figure 9. Frequency dependent photoswitching characteristics. **a** Normalized R and **b** normalized D^* vs frequency. R and D^* have been normalized to their DC values (at 0 Hz). Both R and D^* show a monotonic decrease with frequency.

Frequency dependent photocurrent (I_{ph}) measurements show a decrease in R and D^* with frequency in Supplementary Figure 9. This monotonic decrease in R and D^* with frequency is consistent with literature.[1]-[2] This happens because of the presence of fast as well as slow electron traps in the MG channel region.

The slow electron traps near the conduction band edge may not be able to respond at 5 kHz frequency, hence there is reduced trapping of photogenerated electrons. Consequently, there is a reduction in photogating and gain with increasing frequency leading to lower photocurrent under AC (5 kHz) conditions and therefore a lower R ($R = \frac{I_{ph}}{P_{in}}$). A pictorial representation of electron and hole dynamics under DC and high frequency operation is presented in Supplementary Figure 10a and b.

Supplementary Figure 10. Carrier dynamics for photogenerated electrons and holes are shown through band diagrams under (a) DC mode of operation (device is illuminated under a constant light source of 532 nm laser) and (b) under modulated light intensity switching from ON to OFF at high frequency (say, 5 kHz).

3. The authors claim that $V_{SG}=0$ V is equivalent to floating V_{SG} . Indeed this argument is supported by Supplementary Fig. 3a which shows similar transfer characteristics under the two conditions. However, this picture is incorrect because of the following arguments.

a) When V_{SG} is floating, we can reasonably conclude $V_{SG} = V_{MG}$ when assuming WSe_2 is ideal conductor.

b) I have to presume in this work the source electrode is grounded as in other literatures. Namely, under $V_{SG} = 0$ V condition, $V_{SG} = V_S$.

c) Based on b, when $V_{SG} = 0$ V, $V_{SG} \neq V_{MG}$ when measuring the transfer characteristics except for one data point ($V_{MG} = 0$ V).

d) I have to presume $V_{SD} = 1$ V in the results of Supplementary Fig. 3a, which is the same as other transfer characteristics in this manuscript.

The authors thank the reviewer for the input. To answer the queries of the reviewer we mention the following points:

(i) We assume that when the reviewer says $V_{MG} = V_{SG}$, he/she is referring to MG and SG WSe_2 being at the same potential. This is not possible since WSe_2 is not an ideal conductor since it has a semiconducting bandgap of 1.2 eV for few layers and will not form an equipotential surface.

(ii) In the device, the dielectric hBN is 15 nm thick. The separation between MG to SG is 1 μ m. So, a strong local gate field (\sim MV/cm) will build up in WSe_2 over the MG and SG area, such that the WSe_2 over MG and SG will be controlled by MG and SG individually.

(iii) In case of floating V_{SG} , V_{SG} is not the same as V_{MG} , which the reviewer has correctly assumed in point c. This is supported by Figure 7. The figure shows a comparison of dark currents between no- V_{SG} and $V_{SG} = V_{MG}$ conditions. Here currents are very similar only when V_{MG} is close to 0 V.

(iv) According to the measurement configuration, for the source electrode $V_S = 0$ V is applied. Also, under $V_{SG} = 0$ V condition, $V_{SG} = V_S$.

(v) V_{SD} or $V_D = 1$ V is applied for Supplementary Figure 3a.

Fig. 7: A comparison between No- V_{SG} and $V_{MG} = V_{SG}$ conditions in a) log and b) linear scale shows similar current values near $V_{MG} = 0$ V. The currents deviate when V_{MG} moves away from 0 V.

Estimation of channel capacitance (individual control of MG and SG over channel): We have estimated the channel capacitance in the three-gate device under No- V_{SG} and $V_{MG} = V_{SG}$ conditions. This will help us to estimate the influence of MG and SG over the channel. The MG and SG capacitances are in parallel. So, total capacitance, $C_{total} = C_{MG} + 2C_{SG}$.

Fig. 8: A cross-sectional schematic of the device along SG-MG-SG direction, showing individual channel capacitances over SG and MG. The capacitances are in parallel.

Now, to calculate channel capacitance, we first calculated channel sheet charge density ($Q_{channel}$ C/cm²).

Then the channel capacitance can be calculated as, $C_{channel} = \frac{dQ_{channel}}{dV_{MG}}$

From this, if we see the $C_{channel}$ for No- V_{SG} and $V_{MG} = V_{SG}$ conditions at $V_{MG} = -0.5$ V, then, $C_{channel(No-VSG)} = C_{MG} = 2.54 \text{ e-}14 \text{ F}$ and $C_{channel(VMG = VSG)} = 1.1\text{e-}13 \text{ F} = C_{total}$.

So, $2C_{SG} = C_{total} - C_{MG} = 8.53 \text{ e-}14 \text{ F}$.

This clearly shows a **4.5 times increment** in total capacitance from MG channel capacitance. This closely matches with the area difference between only MG channel area and extended SG-MG-SG channel area.

This confirms individual control of MG and two SGs over their respective WSe₂ channel regions, and there is no interference/cross-coupling of MG over SGs or vice-versa.

Changes to the supplementary information: In supplementary, Fig. 3a has been modified by adding the information of V_D as follows,

Old figure

Modified figure

4. In Fig. 2d, the authors did not describe how they varied and calculated p_{WSe_2} . Was it varied by applying different V_D ?

The calculation is done for $V_D = 1$ V and extracted over an entire range of V_{MG} .

First, we have calculated the mobility using, $\mu = \frac{L}{W * V_D \left(\frac{\epsilon_0 \epsilon_r}{d} \right)} \times \frac{dI_D}{dV_G}$, derived from MOSFET current equation, operating in the linear region.

Then, we have used the drift current equation, $I = p_{WSe_2} * e * \mu * E * A$, to extract the values of p_{WSe_2} where, p_{WSe_2} is hole concentration in the channel, e is electronic charge in Coulomb, $E = \frac{V_D}{L}$, (V_D is drain voltage and L is the channel length between source and drain) is the electric field and A is the cross-sectional channel area. Here, we have used a constant $\mu \sim 5$ cm²/V-s obtained from the inversion region (beyond threshold voltage) over the entire range of V_{MG} .

Changes to the main manuscript page no. 6 are as follows:

Original text: It can be seen that SG modulation of the channel current is higher for lower p_{WSe_2} in the range of 10^{13} to 10^{15} cm⁻³ (sub-threshold region), as it is easier to deplete at lower S/D channel doping density.

Modified text: It can be seen that SG modulation of the channel current is higher for lower p_{WSe_2} in the range of 10^{13} to 10^{15} cm⁻³ (sub-threshold region), as it is easier to deplete at lower S/D channel doping density. The calculation of p_{WSe_2} is shown in Supplementary Information Section 7.

Changes to the supplementary information:

Section 7 of the Supplementary Information has been added with the detail calculation of channel carrier density, as below:

The calculation of channel carrier density with varying gate voltage under No- V_{SG} condition (only MG is functioning) is presented below.

First, we have calculated hole mobility using the standard FET equation, $\mu = \frac{L}{W * V_D \left(\frac{\epsilon_0 \epsilon_r}{d} \right)} \times \frac{dI_D}{dV_G}$, for linear region operation. Next, we have used the current equation, $I = p_{WSe_2} * e * \mu * E * A$, to extract the values of p_{WSe_2} , where p_{WSe_2} is hole concentration in the channel, e is electronic charge in Coulomb, $E = \frac{V_D}{L}$, (V_D is drain voltage and L is the channel length between source and drain) is the electric field and A is the cross-sectional channel area. Here, we have used a constant $\mu \sim 5$ cm²/V-s obtained from the inversion region (beyond threshold voltage) over the entire range of V_{MG} .

5. In Fig. 4b, First, the authors defined recombination rate as $R_{SRH} = \Delta n/\tau$. This definition is different from the regular definition $R = 1/\tau$. Next, the authors claim that $(p+\Delta p_1)/(n+\Delta n_1) < (p+\Delta p_2)/(n+\Delta n_2)$. This argument is based on another assumption that (in dark, $n_1 = n_2$ and $p_1 = p_2$) under the two conditions. This is an acceptable but not strict assumption because when changing the applied bias, the fermi level that determines n could be changed. The authors should clearly address this assumption.

Authors thank the reviewer for this valuable input. By including the suggestion made by the reviewer in our calculations, we have been able to strengthen the conclusions further as detailed below:

(i) **Reason for using SRH equation:** The fast decay time in a phototransistor is related to recombination lifetime. Under this consideration, as WSe_2 is an indirect bandgap semiconductor, we have used SRH rate for carrier recombination, $R_{SRH} = \Delta n/\tau$ [3]. Here WSe_2 transistor is in ON state with p-type conduction and $p \gg n$. Ref. 3 shows a similar case where SRH recombination rate has been used.

(ii) **Assumption and relation between n_1, n_2 and p_1, p_2 :** We had initially considered electron and hole concentrations under dark conditions for case (1) and (2) to be the same, i.e. $n_1 = n_2$ and $p_1 = p_2$. Now, for the two SG bias conditions as shown in Figure 9 below, the applied MG voltage is kept constant. The doping levels, and hence the Fermi levels of the WSe_2 areas over the MG and the SGs, are controlled solely by MG and SG voltages as shown earlier in this response letter. Then the carrier concentrations for both holes and electrons in the quasi-neutral region (flat band area) over MG should be the same in both cases.

Fig. 9: Schematic energy band diagrams depicting recombination rate (R_{SRH}) dependence on carrier concentrations for (1) ps-pm-ps ($V_{SG} = V_{MG} = -0.5$ V) and (2) ns-pm-ns ($V_{SG} = 2$ V, $V_{MG} = -0.5$ V) configurations.

Now, for case-(2), under $V_{MG} = -0.5 \text{ V}$ and $V_{SG} = 2 \text{ V}$, the p-n junctions formed between MG and the two SGs are reverse biased. We know that, under reverse bias there is a deficiency of minority carrier concentration near the depletion boundary (in this case, electrons in the MG quasi-neutral region), as shown in Figure 10 below. Hence assuming a constant electron concentration throughout the quasi neutral region over MG will lead to a slight decrease in the average electron concentration, such that $n_1 > n_2$, as shown pictorially below. Although the majority hole concentration will remain constant in both cases ($p_1 = p_2 = p$).

Fig. 10: Electron concentration profiles for two different V_{SG} conditions showing $n_1 > n_2$.

Further, band bending from MG towards both SG directions results in an internal electric field from the SG to the MG in case (2). Due to this, under illumination, the photogenerated carriers move in (holes) and out (electrons) from MG to SG, which does not happen in case (1). This leads to $\Delta n_1 > \Delta n_2$, $\Delta p_1 < \Delta p_2$ and $\Delta n_2 < \Delta p_2$, $\Delta n_1 = \Delta p_1$ and $\Delta p_2 > \Delta n_2$. Here Δn_1 , Δp_1 and Δn_2 , Δp_2 are additional carriers generated and accumulated in the conduction channel under illumination. Hence, we still have,

$$\frac{p + \Delta p_1}{n_1 + \Delta n_1} < \frac{p + \Delta p_2}{n_2 + \Delta n_2}$$

Because, n_2 is now less than what we had originally assumed, **the inequality is even stronger**. This shows a higher imbalance in electron and hole concentrations (under light illumination) in case-(2). Less number of electrons available against a large number of accumulated holes over the middle gate makes the recombination of e-h pair in case of case-(2) less probable than case-(1).

Changes to the manuscript:

1. Fig. 4 is changed with revised Fig. 4b, with different electron concentrations n_1 , n_2 , and hole concentration p , for case (1) and case (2).

Old figure:

New figure:

2. Text in the main manuscript on page 9 has been modified as follows.

Original text: These trends are explained through Fig. 4b which depicts two cases- (1) $ps-p_M^+-ps$ ($V_{SG} = V_{MG} = -0.5\text{V}$) and (2) $ns-p_M^-ns$ ($V_{SG} = 2\text{V}, V_{MG} = -0.5\text{V}$). In case (2), lateral $ns-p_M$ and

p_M-n_S band bendings across the SG-MG-SG direction cause excess photogenerated holes into the S/D channel (opposite for electrons) and give rise to i) a very high density of photogenerated holes, and ii) a large difference in electron and hole concentrations in the S/D channel. These factors decrease the e-h recombination rate (R_{SRH} in Fig. 4b) in case (2) when the laser is turned off, leading to high responsivity but slower speed.

Modified text: These trends are explained through Fig. 4b which depicts two cases- (1) $p_S-p_M-p_S$ ($V_{SG} = V_{MG} = -0.5$ V) and (2) $n_S-p_M-n_S$ ($V_{SG} = 2$ V, $V_{MG} = -0.5$ V). In case (2), lateral n_S-p_M and p_M-n_S band bendings across the SG-MG-SG direction form reverse biased p-n junctions between MG and the SG WSe_2 regions. Hence, as explained in detail in the Supplementary Information (section 8), the equilibrium minority electron concentration in dark condition for case (2), n_2 , is less than in case (1), n_1 (as explained in Fig. 4b), due to a depletion of electrons near the depletion edge in the MG region under reverse bias. Under illumination, these band bendings cause excess photogenerated holes to flow into the S/D channel (opposite for electrons) from both sides and give rise to i) a very high density of photogenerated holes, and ii) a large difference in electron and hole concentrations in the S/D channel. These factors decrease the e-h recombination rate (R_{SRH} in Fig. 4b) in case (2) when the laser is turned off, leading to high responsivity but slower speed.

Changes in supplementary information:

We have added a figure in Supplementary Information section 8 and text explanation as shown below:

Reasons for the difference in effective MG minority carrier (electron in this case) concentrations for case (1) and (3) are explained below.

Supplementary Figure 11: Electron concentration profiles for two different V_{SG} conditions showing $n_1 > n_2$.

Minority carrier concentration profile across a reverse biased MG-SG pn junction [case (2) in Fig. 4b of main manuscript] is shown in the Supplementary Figure 11. The channel over the MG region is the main region of interest since carrier conduction takes place primarily over the MG area. As it is well-known in pn junction theory that minority carriers get depleted in the quasi-neutral region near the boundary of the pn depletion region, the equilibrium (dark condition) electron carrier concentration in the MG region dips down near the depletion edge. When $V_{MG} = -0.5$ V is applied, and the channel becomes p-type, electrons act as minority carriers in the MG channel. The actual electron concentration profile (n_{p0} or $n_{2actual}$) is shown above. Now for simplification, if we consider a uniform and average electron concentration n_{2avg} throughout the MG channel then, $n_2 < n_1$, where n_1 is the electron concentration for case (1) in Fig. 4b of the main manuscript.

6. In Page 8, the authors argued “...the number of traps participating in the photocurrent transport along the S/D direction also decreases. This ... is one of the factors behind lower responsivity (photoconductive gain) at $V_{SG} = -2$ V.” Please explain why.

The authors thank the reviewer for raising this query. At first, we need to look at the complete section from the main manuscript to discuss the above-mentioned sentence.

(i) The section in the manuscript is as follows,

“Since the channel width decreases with increasing lateral depletion width as V_{SG} increases from -2 V to 2 V, the number of traps participating in the photocurrent transport along the S/D direction also decreases. This is consistent with the lower value of α for $V_{SG} = -2$ V and is one of the factors behind lower responsivity (photoconductive gain) at $V_{SG} = -2$ V.”

The loss of electrons from the middle channel appears as an “apparent” decrease in recombination rate, hence a lower effective trap density and a higher alpha from a conventional photocurrent vs laser power stand point. Along with that, due to the flow of photogenerated holes from the side channel to the middle, we get an additional boost to responsivity. This two-fold enhancement in responsivity and photocurrent is discussed in detail below.

Under $V_{SG} = -2$ V and 2 V and $V_{MG} = -0.5$ V, we have these following observations.

Case-(1): Lower alpha (“apparent” higher effective trap density)

Case-(2): Higher alpha (“apparent” lower effective trap density)

Fig. 11: Total number of electrons available in the conduction band under the two SG bias conditions. The probability of recombination is shown by dotted arrows for case-(1) and case-(2).

1. For -2 V and 2 V V_{SG} cases (1) and (2) above, the band bending will be in opposite directions along SG-MG-SG, while $V_{MG} = -0.5$ V is held constant.

2. In case of $V_{SG} = 2$ V (p_M-n_S configuration) the depletion region encroaches the MG area in contrast to the $V_{SG} = -2$ V case ($n_M-n_S^+$ configuration). This leads to a smaller S/D channel width for $V_{SG} = 2$ V resulting in fewer electrons in the conduction channel.

3. In addition, photogenerated electrons from the MG channel go away to the SG area under the action of the reverse bias depletion field at $V_{SG} = 2$ V. This results in further lowering of the electron concentration in the conduction channel. This decreases the probability of recombination of e-h pairs. Whereas in case of $V_{SG} = -2$ V, electrons come in to the channel over the MG area, and the recombination of e-h pairs gets enhanced, resulting in higher effective trap density and hence a lower α value and lower responsivity, than $V_{SG} = 2$ V [4].

(ii) Secondly, the statement “This ... is one of the factors behind lower responsivity (photoconductive gain) at $V_{SG} = -2$ V” can be explained by depletion formation and strong internal electric field for positive V_{SG} . Again, if we go back to Figure 11, for $V_{SG} = 2$ V, there is large band bending, which will separate out electrons and hole efficiently. This large accumulation of holes in the MG channel from SG also contributes to the photocurrent and the responsivity. On the other hand, movement of electrons from MG to SG increases e-h recombination time. That means in the absence of enough number of photogenerated electrons in the channel, a photogenerated hole gets more time to circulate through the channel multiple times before it recombines with an electron leading to photoconductive gain.

Changes to manuscript:

Original sentence in page 8 of the manuscript: Since the channel width decreases with increasing lateral depletion width as V_{SG} increases from -2 V to 2 V, the number of traps participating in the photocurrent transport along the S/D direction also decreases. This is consistent with the lower value of α for $V_{SG} = -2$ V and is one of the factors behind lower responsivity (photoconductive gain) at $V_{SG} = -2$ V.

Modified: Along with a decrease in channel width, an opposite and stronger reverse bias internal electric field builds up as V_{SG} increases from -2 V to 2 V. This helps in sweeping out electrons from the conduction band over MG to the SG regions and accumulating holes from the SG regions into the MG channel for $V_{SG} = 2$ V. On the other hand, for $V_{SG} = -2$ V, the electrons and holes flow in opposite directions, causing accumulation of electrons and depletion of holes in the MG area. This leads to more pronounced e-h pair recombination in case of $V_{SG} = -2$ V. Hence a smaller number of photogenerated carriers reach the contacts. This is consistent with a lower value of α for $V_{SG} = -2$ V indicating a higher effective trap density (higher recombination rate), and is one of the factors behind the lower responsivity and photoconductive gain at $V_{SG} = -2$ V.

7. In Page 9, “there is no significant imbalance in photogenerated electron and hole concentrations, leading to low τ_{life} and hence low G and R. “Please explain why.

The authors thank the reviewer for this query. This can be explained with the following points.

(i) Dark condition under OFF state and filling of traps: Under transistor ‘OFF’ state ($V_{\text{MG}} = 0.4 \text{ V}$), the transistor channel under MG is n-doped (hole depleted and electron rich). In this condition under dark state, most of the electron trap states near the conduction band will get filled with dark state electrons.

(ii) No vacant trap states under illumination leads to balanced photogenerated electron and hole concentrations: Now, under illumination, very few photogenerated electrons will get a chance to occupy the few remaining unfilled electron trap states. As we know, 1) under photogeneration same number of electron and holes are generated and 2) concentrations of photogenerated electrons and holes dominate over dark state concentrations. So, under illumination, there will be negligible imbalance in photogenerated electron and hole concentrations. Similar results have been reported in MoSe_2 phototransistors in reference [5].

Therefore, a smaller number of available empty traps upon illumination make photogating effect less influential under this bias condition (OFF state) leading to lower R, hence lower G, since G is directly proportional to R as shown in the expression below [6].

$$R = \frac{\lambda q}{hc} \times \text{EQE} \times G$$

Here, λ is the incident wavelength, q is electronic charge, h is Planck’s constant and c is the speed of light.

8. In 2nd paragraph of Page 10, the authors compared Fig. 4c with Fig. 4f. However, the results from the two figures are acquired under two experiment configurations. In Fig. 4c, V_{MG} is controlled while V_{SG} is changeable while V_{SG} is controlled while V_{MG} is changeable. Is this comparison fair?

The authors thank the reviewer for asking this question. In this study our aim is to show a reduction in responsivity and speed trade-off with our novel device architecture compared to **conventional phototransistors**. Hence a comparison in responsivity-speed trade-off has been established between,

1) Conventional trap-based limitation, which is limited by accessible trap density in a device (No- V_{SG} , variable V_{MG}) against, 2) a photo carrier density dependent trade-off (fixed V_{MG} , variable V_{SG}), as observed in our device. We have compared changes in device performance against a conventional phototransistor control sample.

Control experiment with variable V_{MG} and No- V_{SG} , to determine responsivity-speed trade-off: A well-established observation in layered materials-based phototransistors suggests that for

high responsivity (due to the presence of trap states and photogating) values, response time and or decay time get longer. On the other hand, response or decay time reduces with fewer traps in the conduction channel, which on the other hand decreases responsivity.

Now, in our study, in case (2) (Fig. 4e in the main manuscript, also shown in Figure 12 in this response) V_{SG} is fixed and V_{MG} is changing. As V_{MG} changes, the Fermi level within WSe_2 moves from the conduction band to the valence band or vice-versa. This causes either electron or hole traps to be active, based on the position of the Fermi level (near conduction or valence band edge). So, case (2) is **a conventional study** for the role of traps in responsivity and speed. The No- V_{SG} condition ensures fair comparison with **conventional phototransistors** that do not have a SG electrode/bias.

Our device architecture, responsivity-speed governed by photogenerated hole density:

Further in case (1) (Fig. 4b) we have fixed the V_{MG} and V_{SG} is varied. This makes the Fermi level fixed within WSe_2 bandgap in the MG channel, and hence there is no change in accessible trap density (with the change in V_{SG}), while there is change in responsivity and speed due to accumulation of holes over MG channel with varying positive V_{SG} .

Therefore, a comparison between control conventional phototransistor vs. our device architecture to assess responsivity-speed trade-off with two separate measurement conditions on the same device makes it reasonable.

Fig. 12: Schematic band diagrams for temporal response of the WSe_2 phototransistor. b) Schematic band diagrams depicting recombination rate (R_{SRH}) dependence on carrier concentrations for (1) $p_s-p_M-p_s$ ($V_{SG} = V_{MG} = -0.5V$) and (2) $n_s-p_M-n_s$ ($V_{SG} = 2V, V_{MG} = -0.5V$) configurations and e) Schematic band diagrams along S/D direction depicting electron, hole and available empty trap concentrations for $V_{MG} = -0.5V$ (ON, heavily p-doped channel) and $V_{MG} = 0.4V$ (OFF, low n-doped channel) conditions. (Figure labels kept as **b** and **e** to maintain consistency with above discussion and with main manuscript Fig. 4.)

9. In Fig. 5a, τ_{life} is plotted against V_{SG} . In this result, what is V_{D} ? Maybe $V_{\text{D}} = 1$ V? At least, I assume it should be constant. Then, my next question is about the gain ($G = \tau_{\text{life}} / \tau_{\text{transit}}$ in which $\tau_{\text{transit}} \propto 1/V_{\text{D}}$). Suppose V_{D} is constant, why G and τ_{life} do not coincide with each other in Fig. 5a?

The authors thank the reviewer for this query. For the τ_{life} and G plot in Fig. 5a, applied $V_{\text{D}} = 1$ V and $V_{\text{MG}} = -0.5$ V. Reviewer has correctly pointed out that τ_{life} and G did not coincide.

This is due to the slight difference in apparent mobility, for different applied V_{SG} , since ON currents are different for different V_{SG} . This is due to a variation in channel width due to applied V_{SG} , so the change in on-current due to this change in channel width appears as an apparent change in mobility. This leads to a change in the transit time of carriers from source to drain, as shown in the equation below.

$$\tau_{\text{transit}} = \frac{L^2}{\mu V_{\text{D}}}$$

Hence, there is a slight variation in gain from carrier lifetime.

Changes to the manuscript: In order to clarify the mismatch in G and τ_{life} , the following changes have been done in the main manuscript page 11:

Original sentence in page 11 of the manuscript: Mobility was calculated from the $I_{\text{D}}V_{\text{G}}$ characteristics for no- V_{SG} condition (Fig. 2a) using,

$$\mu = \frac{L}{W \times V_{\text{D}} (\epsilon_0 \epsilon_r / d)} \times \frac{dI_{\text{D}}}{dV_{\text{G}}} \dots \dots \dots (2)$$

Here W and d are the channel width and dielectric (hBN) thickness, respectively. ϵ_0 and ϵ_r are the dielectric constants for vacuum and hBN.

Modified text: Mobility was calculated from the $I_{\text{D}}V_{\text{G}}$ characteristics for no- V_{SG} and positive V_{SG} conditions (Fig. 2a, b) using,

$$\mu = \frac{L}{W \times V_{\text{D}} (\epsilon_0 \epsilon_r / d)} \times \frac{dI_{\text{D}}}{dV_{\text{G}}} \dots \dots \dots (2)$$

Here W and d are the channel width and dielectric (hBN) thickness, respectively. ϵ_0 and ϵ_r are the dielectric constants for vacuum and hBN. Fundamentally, hole mobility in the WSe_2 channel should not change with V_{SG} ; however, the mobility values calculated from the on-currents show some variation with V_{SG} due to the varying channel width. As a result, τ_{transit} also varies with V_{SG} .

10. As shown in manuscript, there are three gates in a device including one middle gate and two side gates. However the shape of the side gates is sometimes interdigitated and sometimes rectangle (Fig. 2b). What's difference between them?

The authors thank the reviewer for pointing out this discrepancy. In our fabricated device architecture, we have used interdigitated side gate structure. In Fig. 2b, the non-interdigitated representation was a simpler way to show the electrical configuration of the device. We have now changed it to an interdigitated structure for the side gates in Fig. 2b to maintain consistency with fabricated devices.

Changes to the main manuscript: We have changed Fig. 2b inset in the main manuscript, with the figure below.

Old figure:

Modified figure:

Changes to the supplementary: We have changed Supplementary Figure 2b inset with the figure below.

Old figure:

Modified figure:

11. In Fig 2a inset, the Schottky barrier-dominated transport behavior across the S/D contacts was shown. Why an obvious asymmetric electrical phenomenon appears in the device with symmetric S/D contacts? What is the difference between the S and D electrodes?

The authors thank the reviewer for the comment. From the fabrication aspect, the source and drain contacts are symmetric (simultaneous metal deposition for both S/D contacts during fabrication).

The $I_D V_{DS}$ are highly asymmetric and Schottky in nature. This is likely due to non-Ohmic Schottky contact formation between metal S and D electrodes with WSe_2 . Due to the high reverse bias field at the source end, the carriers get swept out to the contacts easily, as shown in the band diagram below. The barrier at the drain end controls the carrier injection. A negative V_D increases the Schottky barrier for holes from metal (drain) to WSe_2 , causing a very low hole current, and a positive V_D decreases metal- WSe_2 Schottky barrier leading to an increase in hole injection. So, even though both the contacts are symmetric and Schottky in nature, the IV is Schottky barrier dominated majorly due to the drain contact. However, we would like to point out that although the contacts are Schottky in nature, that does not affect the main focus/result of this work which is to show a fundamental benefit of using the side-gate device, and not the fastest phototransistor.

Fig. 13: Source-drain energy band diagrams for positive and negative V_D – showing low current for negative V_D and high current for positive V_D .

12. The performance comparison between the two types of devices was shown in Fig 4. Actually, we can see when the fall time is 14.5 μs , the R in the device without applied side gate is higher than the fixed side gate and middle gate one; Even in the time interval from 14 to 16 μs that the authors claimed in the abstract, the R is similar in the two types devices (Fig 4c and f). Therefore, the author should clearly determine the advantage for the design of three lateral in-plane gate electrodes than that of only one middle gate.

The authors thank the reviewer for this question. The R data in the R vs. τ_f plots in Figs. 4c and f in the main manuscript was collected under a laser switching frequency of 5 kHz. This should not be compared with DC operation (please see response to Q.2 on dependence of I_{ph} on switching frequency), where R increases by 2x when V_{SG} is applied in contrast to the no- V_{SG} case. **However, we note that even under AC operation there is a 1.3x increase in R for $V_{SG} = 2\text{ V}$ with respect to no- V_{SG} condition.** Having said that, the comparison was mainly aimed to highlight the reduction in “**responsivity vs. speed trade-off**” in case of V_{SG} variation with respect to control sample measurement (variation of V_{MG} , with No- V_{SG}). AC measurement is needed for the measurement of speed.

R under AC operation can be lower than DC operation: The value of R in Figure 4c and f is calculated from the photocurrent (I_{ph}) that is generated by an incident laser excitation at 5 kHz.

- **Prominent photogating and higher modulation in I_{ph} under DC operation:** In the main manuscript as well as in the response to question 2, we have discussed the impact of slow electron traps near the conduction band of WSe₂ and the effect of photogating. Under DC condition, electrons get trapped even in the slow electron trap states easily. Therefore, effectively, a greater number of photogenerated holes get accumulated and are collected by the contacts. This helps in achieving high I_{ph} .

- **Less photogating and poorer modulation of I_{ph} for AC operation:** In the case of AC operation at 5 kHz, photogating is not prominent as the trapping time is likely to be slower than 5 kHz. M. M. Furchi et al. and P. H. Hsu et al. have reported similar observations under high frequency operation, as shown in Figure 4 in response to question 2 [1], [2]. Now, such electron traps are present throughout the device in the WSe₂. That is, in WSe₂ over MG, over SG, as well as over the depletion region in-between MG and SG as shown in Figure 14. Therefore, lesser number of photogenerated holes will be available over SG and the area between SG and MG (Figure 14) under AC operation. Owing to this, the photocurrent does not reach its maximum value under 5 kHz photoswitching. The plot ' R_{norm} vs frequency' also shows the same effect in Figure 5 in response to question 2. Also, because of this reason the modulation in R values with and without side gate at 5 kHz is lower than for DC operation.

Fig. 14: WSe₂ band diagram along SG-MG-SG direction for DC and AC (5 kHz) operation. Under DC operation electron trap states get filled and the rate of e-h recombination decreases. This enables a large number of holes to come in from SG to MG. High frequency (5 kHz) AC operation does not allow enough electron trapping resulting in a larger e-h recombination rate. This leads to a smaller number of holes travelling from SG to MG.

Advantage of the three-gate structure: Although the modulation in R, and its absolute value, for AC operation is smaller than for DC operation, there is a clear advantage for the three-gate structure under AC operation in the following cases.

➤ Advantages under AC operation:

1. **Less number of trap states in the channel material:** Less number of trap states can decrease the number of trapped photocarriers in the channel and thereby, weaken the photogating effect that causes the DC-AC imbalance. This will result in faster photo carrier dynamics [7] and extend the advantage of having three gates to a higher frequency.

2. *Modulating V_{SG}* : For a fixed trap density, one may achieve the same boost in R at high frequency (like DC) operation at a higher V_{SG} . Say, we get a 2x boost in R for $V_{SG} = 1$ V with respect to No- V_{SG} . Then, the same boost in R may be achieved in case of a fixed high frequency operation at $V_{SG} = 2$ V. Therefore, modulating V_{SG} can help in achieving the same R value at different operating frequencies which makes this device structure unique.
3. *Lower operating frequency can replicate DC operation*: For a fixed V_{SG} and trap density, a lower frequency of operation (in this study less than 5 kHz) in the three-gate device can give a boost close to DC operation. At lower frequencies, the effect of photogating increases.

Along with the aforementioned advantages this device architecture can provide few more advantages as listed below.

➤ Advantages under DC operation:

4. In this three-gate structure the device R can be modulated electrically with side gate voltage.
5. Lower off-current for three-gate structure vs single gate structure resulting in higher detectivity.

These have been explained in detail in the response to question 1.

Additional text on page 10 of main manuscript after the sentence “This indicates that the lateral p-n junction enables higher R for the same speed of operation, a better R vs τ_f trade-off as compared to conventional means of varying V_{MG} , trap density and other physical⁸ or chemical⁹ processes.”: It is worth noting that the modulation in R at 5 kHz is smaller than under DC operation. R and D^* as a function of operating frequency are shown in Supplementary Information section 6. This is due to the absence of photogating in the MG, SG and MG-SG depletion regions at high frequency (discussed in Supplementary section 9).^{40,41} Reduced photogating results in less efficient photogeneration of holes due to a higher recombination rate. The reduction in photogenerated hole concentration in the MG region, as well as in the number of holes travelling from SG regions to the MG area, results in lower I_{ph} and R under AC operation. However, increasing V_{SG} allows an additional knob, besides reduced trap density, for increasing R under AC operation through enhanced photogenerated hole movement into the MG region.

Changes in supplementary information:

We have added the above discussion in Supplementary Information section 9 as below:

This section highlights the differences in device performance under DC and AC operating conditions. Also, the advantages of the three-gate structure over a conventional single gate phototransistor under AC and DC operating conditions have been discussed.

R under AC operation can be lower than DC operation: The value of R in Figure 4c and f is calculated from the photocurrent (I_{ph}) that is generated by an incident laser excitation at 5 kHz.

- *Prominent photogating and higher modulation in I_{ph} under DC operation:* In the main manuscript, we have discussed the impact of slow electron traps near the conduction band of WSe₂ and the effect of photogating. Under DC condition, electrons get trapped even in the slow electron trap states easily. Therefore, effectively, a greater number of photogenerated holes get accumulated and are collected by the contacts. This helps in achieving high I_{ph} .
- *Less photogating and poorer modulation of I_{ph} for AC operation:* In the case of AC operation at 5 kHz, photogating is not prominent as the trapping time is likely to be slower than 5 kHz. M. M. Furchi et al. and P. H. Hsu et al. have reported similar observations under high frequency operation [1], [2]. Now, such electron traps are present throughout the device in the WSe₂. That is, in WSe₂ over MG, over SG, as well as over the depletion region in-between MG and SG as shown in Figure 12. Therefore, lesser number of photogenerated holes will be available over SG and the area between SG and MG, to travel to the MG region (Supplementary Figure 12) under AC operation. Owing to this, the photocurrent does not reach its maximum value under 5 kHz photoswitching. The plot ' R_{norm} vs frequency' also shows the same effect in Figure 10 of Supplementary Information. Also, because of this reason the modulation in R values with and without side gate at 5 kHz is lower than for DC operation.

Supplementary Figure 12. WSe₂ band diagram along SG-MG-SG direction for DC and AC (5 kHz) operation. Under DC operation electron trap states get filled and the rate of e-h

recombination decreases. This enables a large number of holes to come in from SG to MG. High frequency (5 kHz) AC operation does not allow enough electron trapping resulting in a larger e-h recombination rate. This leads to a smaller number of holes travelling from SG to MG.

Advantage of the three-gate structure: Although the modulation in R, and its absolute value, for AC operation is smaller than for DC operation, there is a clear advantage for the three-gate structure under AC operation in the following cases.

➤ Advantages under AC operation:

1. *Less number of trap states in the channel material:* Less number of trap states can decrease the number of trapped photocarriers in the channel and thereby, weaken the photogating effect that causes the DC-AC imbalance. This will result in faster photo carrier dynamics [3] and extend the advantage of having three gates to a higher frequency.
2. *Modulating V_{SG} :* For a fixed trap density, one may achieve the same boost in R at high frequency (like DC) operation at a higher V_{SG} . Say, we get a 2x boost in R for $V_{SG} = 1$ V with respect to No- V_{SG} . Then, the same boost in R may be achieved in case of a fixed high frequency operation at $V_{SG} = 2$ V. Therefore, modulating V_{SG} can help in achieving the same R value at different operating frequencies which makes this device structure unique.
3. *Lower operating frequency can replicate DC operation:* For a fixed V_{SG} and trap density, a lower frequency of operation (in this study less than 5 kHz) in the three-gate device can give a boost close to DC operation. At lower frequencies, the effect of photogating increases.

Along with the aforementioned advantages this device architecture can provide few more advantages as listed below.

➤ Advantages under DC operation (as described in Supplementary Information section 5):

4. In this three-gate structure the device R can be modulated electrically with side gate voltage.
5. Lower off-current for three-gate structure vs single gate structure resulting in higher detectivity.

13. Other problems to be checked carefully include

a. Typo: “WSe₂” in Fig 2d;

b. The SiO₂ layer thickness is claimed to be 285 nm in Methods and 280 nm in Results. Which is correct?

a. In p_{WSe_2} , WSe₂ is in subscript.

Change to the manuscript: p_{WSe_2} has been changed in the following sections.

- i. In page 5 under Fig. 2 caption and,
- ii. In page 6.

Original manuscript: p_{WSe_2}

Modified: We have changed p_{WSe_2} in the manuscript to D_{WSe_2} .

b. The thickness is 285 nm.

Change to the manuscript: We have corrected the SiO_2 thickness to 285 nm from 280 nm on page 3 of the manuscript.

Reviewer 2:

This paper reports on the fabrication and characterization of npn and p^+pp^+ phototransistors enabled by gate tunable homojunctions formed in WSe_2 channels. The work presents a thorough characterization of the devices and the paper is well written. However I think that in its current form the paper is not suitable for publication because the claims are not supported by the data. In particular:

1. The authors state on page 1 that the noise of the detectors is determined by the dark current. This is only partially true because such detectors are known to be severely limited by $1/f$ noise. Then in all the calculations of the D^* the authors assume the shot noise limit, they actually do not measure the noise current. Therefore the claims of such high D^* values are not supported. Related to this the minimum detectable power that their detectors can reach is 300 pW, this taken together with the area of the detector of approx. $10 \mu\text{m} \times 10 \mu\text{m}$, through the equation of $D^* = \sqrt{(\text{Area} \times \text{BW})/\text{NEP}}$, yields a D^* of $1e7$ Jones, about 7 orders of magnitude lower than the claimed one.

Shot noise, a common consideration to calculate D^* in TMD phototransistor: We agree with the reviewer's comment, and thank him/her for the suggestion. Typically, the overall noise is a combination of shot noise, thermal (Johnson) noise and flicker noise. In several 2D layered materials-based phototransistor studies it is very common to consider shot noise measured from dark current as the major contributor among the above noise components mentioned. [5], [8]-[9]

Calculation of thermal noise: Although shot noise is commonly considered in detectivity (D^*) calculations for TMD phototransistors, we have further extended our calculation and determined D^* from thermal noise as well.

D^* quantifies the phototransistor's ability to measure the weakest possible signal and noise equivalent power (NEP) is the incident light power at which a detector yields a signal-to-noise

ratio of unity. Considering a bandwidth B of the phototransistor, the NEP depends on responsivity (R) and the noise current (I_N) in the phototransistor as $NEP = \frac{I_N}{R}$. The noise can be a sum of the shot noise (I_S), thermal (Johnson) noise (I_T) and flicker noise. Shot noise is calculated from I_{dark} as $I_S = (2qBI_{dark})^{\frac{1}{2}}$, whereas thermal noise is calculated using the value of shunt resistance as $I_T = \left(\frac{4k_B T}{R_{ch}} B\right)^{\frac{1}{2}}$. R_{ch} is the output channel resistance, $\frac{V_D}{I_{dark}}$. D^* is then computed as $D^* = \frac{R(AB)^{\frac{1}{2}}}{I_N}$, where A is the area of the active channel over the MG.

In Figure 15 we have presented the D^* value calculated from shot noise as well as thermal noise.

Fig. 15: Detectivity vs. V_{MG} plotted for input optical power of 33 nW using a) shot noise limit and b) thermal noise limit.

From Fig. 15 above it is evident that at 300K, the detectivity calculated from shot noise and thermal noise is in the same range. D^* calculated from shot noise is slightly lower than that calculated using thermal noise. Therefore, we can conclude that D^* is limited by shot noise, and not by thermal noise.

$D^* 1e7$ Jones is not reasonable: In the D^* calculation, the reviewer has assumed 300 pW to be the NEP for our detector. However, this is only a low optical power at which the fast photoswitching measurements were carried out due to measurement limitations and should not be taken as the NEP since we also demonstrate photoswitching at further low incident power of 0.25 pW.

Changes in the manuscript: The changes made in the manuscript on page no. 11 are as follows.

Original sentence in the page 11 of the manuscript: D^* is another important performance parameter which measures the detector's ability to detect weak optical signals with reference to its own noise level. D^* is calculated using,

$$D^* = R \times \frac{\sqrt{A}}{(2qI_{\text{dark}})^{\frac{1}{2}}} \dots\dots\dots (4)$$

where A is the channel area over the MG. Fig 6a shows D* vs V_{MG} at 330 nW of laser power, for V_{SG} varying from -2 V to 2 V and for no-V_{SG}. D* increases monotonically with V_{SG}, with a change of ~1.5 order (5x10⁹ Jones for V_{SG} = -2 V to 1x10¹¹ Jones at V_{SG} = 2 V) for maximum R at V_{MG} = -0.5 V. A 3x enhancement in maximum detectivity (D*_{max}) is obtained for V_{SG} = 2 V over no-V_{SG} at V_{MG} = 0.4 V. When the SG-MG-SG band configuration goes from p_S-p_M-p_S to n_S-p_M-n_S, the dark current noise (shot noise = (2qI_{dark})^{1/2}) decreases due to a decrease in I_{dark} (Fig. 2b) and R increases (Fig. 3c). Therefore, increasing V_{SG} gives a two-fold enhancement in D*.

Modified: D* is another important performance parameter that measures the detector's ability to detect weak optical signals with reference to its own noise level. D* is calculated using,

$$D^* = R \times \frac{\sqrt{A}}{(2qI_{\text{dark}})^{\frac{1}{2}}} \dots\dots\dots (4)$$

where, A is the channel area over the MG. Here shot noise = (2qI_{dark})^{1/2} has been calculated using the dark current and considered to be the major contributor to D*.^{14,31,34} Fig 6a shows D* vs V_{MG} at 33 nW of laser power, for V_{SG} varying from -2 V to 2 V and for no-V_{SG}. D* increases monotonically with V_{SG}, with a change of ~1.5 order (5x10⁹ Jones for V_{SG} = -2 V to 1x10¹¹ Jones at V_{SG} = 2 V) for maximum R at V_{MG} = -0.5 V. A 3x enhancement in maximum detectivity (D*_{max}) is obtained for V_{SG} = 2 V over no-V_{SG} at V_{MG} = 0.4 V. When the SG-MG-SG band configuration goes from p_S-p_M-p_S to n_S-p_M-n_S, the dark current noise (shot noise) decreases due to a decrease in I_{dark} (Fig. 2b) and an increase in R (Fig. 3c). Therefore, increasing V_{SG} gives a two-fold enhancement in D*. Since shot noise is not the only contributor to detector noise, D* calculations accounting for thermal noise are shown in Supplementary Information section 10. Comparison of D* calculated from both sources of noise shows that shot noise is the more dominant source amongst the two.

References:

14. Lee, H., Ahn, J., Im, S., Kim, J. & Choi, W. High-Responsivity Multilayer MoSe₂ Phototransistors with Fast Response Time. *Sci. Rep.*, vol. 8, pp. 11545, (2018).

31. Xu, H., Zhu, J., Zou, G., Liu, W., Li, X., Li, C., Ryu, G. H., Xu, W., Han, X., Guo, Z., Warner, J. H., Wu, J. & Liu, H. Spatially Bandgap - Graded MoS₂(1-x)Se₂x Homojunctions for Self - Powered Visible – Near - Infrared Phototransistors. *Nano-Micro Lett.*, pp. 1–14 (2020).

34. Choi, W., Cho, M. Y., Konar, A., Lee, J. H., Cha, G., Hong, S. C., Kim, S., Kim, J., Jena, D., Joo, J. & Kim, S. High-detectivity multilayer MoS₂ phototransistors with spectral response from ultraviolet to infrared. *Adv. Mater.*, vol. 24, no. 43, pp. 5832–5836, (2012).

Changes to the Supplementary information:

We have added the calculations of D^* based on thermal noise and a comparison of D^* calculated from shot noise as well as thermal noise in Supplementary section 10.

D^* quantifies the phototransistor's ability to measure the weakest possible signal and noise equivalent power (NEP) is the incident light power at which a detector yields a signal-to-noise ratio of unity. Considering a bandwidth B of the phototransistor, the NEP depends on responsivity and the noise current (I_N) in the phototransistor as $NEP = \frac{I_N}{R}$. The noise can be a sum of the shot noise (I_S), thermal (Johnson) noise (I_T) and flicker noise. Shot noise is calculated from I_{dark} as $I_S = (2qBI_{dark})^{\frac{1}{2}}$, whereas thermal noise is calculated using the value of shunt resistance (R_{SH}) as $I_T = \left(\frac{4k_B T}{R} B\right)^{\frac{1}{2}}$. R is the output channel resistance, $\frac{V_D}{I_{dark}}$. D^* is then computed as $D^* = \frac{R(AB)^{\frac{1}{2}}}{I_N}$, where A is the area of the active channel over the MG.

Figure 13 depicts D^* values calculated from shot noise as well as thermal noise.

Supplementary Figure 13. Detectivity vs. V_{MG} plotted for input optical power=33 nW using a) shot noise limit, and, b) thermal noise limit.

From Supplementary Figure 13, it is evident that at 300 K, the detectivity calculated from shot noise and thermal noise is in the same range. D^* calculated from shot noise is slightly lower than that calculated using thermal noise. Therefore, it can be concluded that D^* is limited by shot noise, and not by thermal noise.

2. In Fig 6 the authors plot the D^* as a function of incident power, this is meaningless, because D^* defines the minimum detectable power of the detector and should be reported only as a function of temperature, frequency, wavelength, applied bias.

The authors thank the reviewer for this input. We agree with the reviewer and have removed the plot of D^*_{max} vs P_{in} in Fig. 6b. Instead, we have added D^*_{max} for different V_{SG} , as shown in

Figure 16. In the inset of Fig. 6b we have presented the effect of V_{SG} on D^* for DC and 5 kHz operation under $P_{in} = 33$ nW. Also, we have shown the changes in D^* in ‘ D^*_{norm} vs frequency’ plot in Supplementary Figure 5b.

Fig. 5: (Same figure as in response to reviewer 1, question 2) Frequency dependent photoswitching characteristics. **a** Normalized R and **b** normalized D^* vs frequency. R and D^* have been normalized to their DC values (at 0 Hz). Consistent with literature [1], [2], both R and D^* show a monotonic decrease with frequency.

Changes to the main manuscript: We have replaced Fig. 6b (D^*_{max} vs. P_{in}) in the main manuscript with D^*_{max} vs. applied V_{SG} . In the inset of Fig. 6b we have presented the effect of V_{SG} on D^* for DC and 5 kHz operation under $P_{in} = 33$ nW.

Old figure:

Modified figure:

Original sentence in the page 11 text of the manuscript: Fig. 6b shows power dependent D^*_{\max} (at $V_{MG} = 0.4$ V) for varying V_{SG} . D^*_{\max} decreases monotonically with increase in laser power.

Modified sentence: Fig. 6b shows D^*_{\max} for varying V_{SG} at $V_{MG} = 0.4$ V. Inset shows the effect of V_{SG} on D^* under DC and 5 kHz operating frequency, estimated from photoresponsivity values measured at $P_{in}=33$ nW. These plots clearly demonstrate the advantage of the three-gate structure under different operating frequencies.

Original sentence in Fig. 6 caption: **b** D^*_{\max} vs input optical power. D^*_{\max} decreases monotonically with increase in P_{in} and a maximum D^*_{\max} of 1.6×10^{14} Jones is obtained at $P_{in} = 300$ pW.

Modified text in Fig. 6 caption: **b** D^*_{\max} for varying V_{SG} . A maximum D^*_{\max} of 1.6×10^{14} Jones is obtained for $V_{SG} = 2$ V at $V_{MG} = 0.4$ V. Inset shows V_{SG} vs. D^* under DC and 5 kHz operating frequency, estimated from photoresponsivity values measured at $P_{in}=33$ nW.

3. The spectral response of the photodetectors is missing.

We thank the reviewer for bringing up the spectral response. Several studies have shown the spectral response of few layer WSe_2 phototransistors as shown in Figure 16.¹⁰⁻¹¹. The response is nearly constant in the range of 500 nm to 800 nm and it reaches a maximum at around 800 nm.

Fig. 16: Responsivity vs. wavelength and energy is shown for few layer WSe_2 from a) Yang. Y. et. al., *J. Mater. Chem. C*, (2018), b) Hsu. H. P., *Materials*, (2019).

Impact of wavelength on device performance:

- i) **No impact of wavelength for single material homojunction:** A common observation is a slight change in responsivity against wavelength as the material's (here in this case WSe₂) photo absorption changes with wavelength. However, this can only change the photoresponse or responsivity number. The goal of our study is to improve responsivity and detectivity with V_{SG}, as well as to reduce the responsivity-speed trade-off, in comparison to No-V_{SG} (conventional phototransistor operating condition). This won't get affected with a change in wavelength within the range of WSe₂'s light absorption limit.
- ii) **Wavelength dependence for lateral heterojunctions:** However, if the lateral three-gate structure shown in this work were to be realized using different n and p-type materials, placed laterally in an n-p-n configuration in the SG-MG-SG direction, then the heterostructure could have a different and possibly wider range of wavelength absorption as compared to a lateral homojunction (this work). This could result in a wavelength dependent impact of the SG bias on the responsivity and detectivity.

Changes to the manuscript: The following changes have been incorporated at the end of the **Discussion** section in the main manuscript.

Original statement in the page 13 of the manuscript: From a broader perspective, this study demonstrates simple optoelectronic device architecture for achieving high responsivity and detectivity without compromising speed, which can be realised in any optically active 2D layered or thin film semiconductor material that is amenable to electrostatic doping.

Modified: From a broader perspective, this study demonstrates a simple optoelectronic device architecture for achieving high responsivity and detectivity without compromising speed, which can be realized in any optically active 2D layered or thin film semiconductor material that is amenable to electrostatic doping. Further, a lateral heterojunction device with different n and p type materials could additionally enable a broader spectral response and wavelength selectivity to achieve enhanced performance. This can have promising applications in integrated photonics and optoelectronics devices.

4. I would like to see some control experiments to strongly prove that the use of those homojunctions npn is really necessary to achieve high performance. What happens for example if you would have only one heterojunction across the channel?

We thank the reviewer for the question. Our study is completely based on homojunctions of WSe₂. We believe the reviewer is asking about "only one p-n junction" on the side when he/she is referring to "**only one hereojunction**". In the study we have done control experiments to evaluate performance with respect to only MG (No-V_{SG}) and SG applied- similar to conventional phototransistors.

One SG reduces the advantage of V_{SG} to half: Now both the SGs in the device are symmetrically fabricated with the same continuous and uniform dielectric (hBN) and channel

material (WSe_2). Both the SGs are equidistant from the MG, fabricated simultaneously and operate independently of each other. Due to this if a single SG is employed instead of two, it can be estimated that the effect of SG will be halved. A single SG would lead to a single p-n junction instead of two (like the double SGs in our work) thereby reducing the carrier separation to half as shown in the Fig. 16(a) schematic. With this consideration we have divided the responsivity and detectivity numbers by two and compared in figure 16.

Fig. 17: The figure shows (a) electrical measurement connection configurations and formation of p-n homojunction for No- V_{SG} , double V_{SG} and single V_{SG} and (b) a comparison for maximum R and D^* for No- V_{SG} , and $V_{SG} = 2$ V with one and two side gates. Values for R and D^* presented in case of single side gate architecture are calculated based on the assumption discussed above.

The above Fig. 17 shows that even with a single SG the responsivity and detectivity numbers are still higher for positive V_{SG} under ON state than for the No- V_{SG} case due to the high internal electric field. This clearly shows the benefit of the lateral p-n homojunction in separating out electrons and holes and enhancing responsivity.

Changes in the manuscript: We have made changes in the main manuscript page 7, as follows.

Original sentence in page 7 in the manuscript: Hence, the S/D channel now carries the original as well as additional photogenerated holes supplied from the SG regions to the channel. These additional holes are also collected by the S/D contacts due to the presence of a source-to-drain electric field, thereby providing additional photocurrent in the n_S - p_M - n_S configuration.

Modified: Hence, the S/D channel now carries the original as well as additional photogenerated holes supplied from the SG regions to the channel. These additional holes are also collected by the S/D contacts due to the presence of a source-to-drain electric field, thereby providing additional photocurrent in the n_S - p_M - n_S configuration. Even a single SG will show a similar but

smaller increase in photocurrent through the n_s - p_M configuration under ON state. Detailed advantages of using a three-gate structure as compared to the conventional single gate FET are given in Supplementary Information section 5.

Changes to the Supplementary Information: We have discussed the effect of single side gate in Supplementary Information section 5.

Advantage of single SG over no-SG: The two device SGs have been symmetrically fabricated with the same, continuous and uniform dielectric (hBN) and channel material (WSe_2) on top of the dielectric. Both the SGs are equidistant from the MG, fabricated simultaneously and operate independently of each other. Due to this, if a single SG is employed instead of two, it can be estimated that its effect will be reduced to half, with respect to the double SGs as shown in Fig. 8(a) schematic. A single SG would lead to a single p-n junction instead of two thereby reducing the carrier separation to half. With this consideration we have divided the responsivity and detectivity numbers by two and compared in Fig. 8(b).

Supplementary Figure 8. The figure shows (a) electrical measurement connection configurations and formation of p-n homojunction for No- V_{SG} , double V_{SG} and single V_{SG} and (b) a comparison for maximum R and D^* for No- V_{SG} , and $V_{SG} = 2$ V with one and two side gates. Values for R and D^* presented in case of single side gate architecture are calculated based on the assumption discussed in the text.

Supplementary Figure 8b shows that even with a single SG, the responsivity and detectivity numbers are still higher for positive V_{SG} than No- V_{SG} . This clearly shows the benefit of the lateral p-n homojunction in separating out electrons and holes and enhancing responsivity.

5. The authors make use of hBN a material that at the moment is not scalable, why is this needed? What is the performance of those structures by replacing hBN with another dielectric, through a scalable process?

The use of hBN is advantageous because of the following reasons.

Less interface traps: hBN is another layered vdW material. Hence, use of hBN gives a high quality semiconductor-dielectric interface between hBN and WSe₂ due to the presence of very few dangling bonds at the interface. This leads to better mobility and on current.

Flexible: The flexibility of hBN makes it useful in 2D materials based flexible electronics which might not be possible for other standard dielectrics like Al₂O₃, HfO₂ etc.

Scalability of hBN: Although we have used mechanical exfoliation from bulk crystals to get hBN as well as WSe₂ in our study, well-controlled CVD growth of hBN and WSe₂ has been demonstrated in the literature,[12]-[13] which can be used for scalable electronics. Since the main aim of this study is to demonstrate the performance benefit of a gated, lateral p-n junction device architecture, we have not employed CVD-grown materials to demonstrate scalability.

Changes to the manuscript: A sentence regarding scalability of hBN has been added to the manuscript on page 3, as follows:

Original text in the manuscript page 3: The fabrication process flow has been described in detail in Supplementary Fig. 1.

Modified: The fabrication process flow has been described in detail in Supplementary Fig. 1. Although mechanically exfoliated flakes from bulk crystals have been employed in this work, use of CVD grown large area flakes/layers of hBN³⁵ and WSe₂³⁶ can enable wafer-scale fabrication of a large array of devices.

Reference:

35. Stehle, Y., Meyer, H. M., Unocic, R. R., Kidder, M., Polizos, G., Datskos, P. G., Jackson, R., Smirnov, S. N. & Vlassiuk, I. V. Synthesis of Hexagonal Boron Nitride Monolayer: Control of Nucleation and Crystal Morphology, *Chem. Mater.* vol. 27, no. 23, pp. 8041-8047, (2015).
36. Wang, X., Li, Y., Zhuo, L., Zheng, J., Peng, X., Jiao, Z., Xiong, X., Han, J. & Xiao, W. Controllable growth of two-dimensional WSe₂ using salt as co-solvent, *Cryst. Eng. Comm.* vol. 20, no. 40, pp. 6267-6272, (2018).

Reviewer 3:

In this manuscript, the authors explored the photo responsivity, speed, and sensitivity of a WSe₂ phototransistor. While the ideas presented by the authors are interesting, there are a few remaining issues that need to be addressed.

1. A WSe₂ phototransistor is explored under the gate voltages ranging from -2V and 2V; therefore, it is important to know its electrical transport properties in the same range. In other words, it would be necessary to extend the I_D curvatures as a function of V_{MG} to (-2V – 2V) in Fig. 2a and 2b and to calculate both the hole and electron mobility. Since the channel mobility play an important role in both photo responsivity and speed, it would be interesting to know whether electron and hole have different mobility and whether the mobility is important in the following experiments (different “cases”).

The authors thank the reviewer for asking this question. We have extended the $I_D V_{MG}$ range from -0.5V-to-0.6V to -1V-to-1V.

Fig. 18: Extended $I_D V_{MG}$ curves under dark state from -1 V to 1 V V_{MG} . for (a) only for No- V_{SG} and (b) varying V_{SG} s.

However these extended IVs in Figure 17, do not add any extra value to the study.

Hole mobility is more important for responsivity and speed in p-FET: The reviewer has mentioned about the importance of hole as well as electron mobility. In this study the FET is a p-FET and the S/D contacts selectively inject holes into the channel, not electrons. This is evident from the $I_D V_D$ plot in Fig. 2a inset and is also clearly seen from the $I_D V_{MG}$ plot in Fig. 2a and 2b, presented here again in Figure 17. Please also refer to the energy band diagram in response to Q. 11 (reviewer 1) showing the large electron barrier and small hole barrier at the source-drain contacts to WSe₂ channel.

Hence we can say that this study is based on hole conduction. There is no significant conduction of electrons in the S/D direction. The only electron movement that takes place is in the lateral SG to MG direction and vice-versa depending on the SG potential. Thus, only hole mobility plays an important role in dark and photo current transport. Also, under positive V_{MG} , when the Fermi

level over MG moves towards the conduction band and electron conduction is expected to dominate, the FET is under ‘OFF’ state. Hence, electron mobility is not relevant for this device.

Both electron and hole mobilities will be important, if a study is carried out on ambipolar devices, where we get significant electron as well as hole current. This can be obtained by tuning the hole and electron Schottky contact barriers to roughly midgap of the channel TMD semiconductor.

Changes to the manuscript: We have changed the transfer characteristics in manuscript Fig. 2(a) and (b). We have extended the $I_D V_{MG}$ range from -0.5V-to-0.6V to -1V-to-1V.

2. The hole mobility of the device in this manuscript is about $5 \text{ cm}^2/\text{Vs}$, which is lower than other reports. As a result, the device is expected to have relatively low photo responsivity and long response time. Some discussion would strengthen the manuscript.

The high responsivity value is likely due to the following reasons.

High photo absorption causes high responsivity: The laser used for photo excitation is a 532 nm laser. In this range, WSe_2 has a high absorption coefficient; nearly $10^5/\text{cm}$.¹¹ High responsivity is a product of quantum efficiency which depends on absorbed light and photoconductive gain.

The total absorption of light also depends on channel material thickness, which is 7 nm (~ 10 layers) here. Total absorbed optical power can be expressed as, $P_{\text{abs}} = P_0 (1 - \exp^{-\alpha t})$.¹³ Here P_0 is the incident optical power, α is the absorption coefficient and t is the WSe_2 thickness. The WSe_2 thickness in this work is sufficiently large (few-layer) to ensure high photo absorption.

High channel trap density causes photoconductive gain and high responsivity: The photoconductive gain is typically attributed to the existing trap states in the channel, which has been discussed in our study as well as many other studies.¹⁴⁻¹⁵ A high trap density on our WSe_2 can be another likely reason for significant photogating and high responsivity. This is also further confirmed by a substantial difference in the DC and AC responsivity values as can be seen from Fig. 5.

Fast recombination and fast transit time cause high speed: The transit time has been calculated for $V_D = 1$ V, and $\mu = 5$ cm²/V-s. The calculated transit time for a hole to pass from source to drain for a channel length of 10 μ m, is in sub-microsecond. Again, the recombination limited fall time is typically very fast, as also reported by other studies. Therefore, a fast response time can be credibly achieved with a mobility value of 5 cm²/V-s.

Improved mobility can boost responsivity further: Furthermore, in future studies, one may work on lowering S/D resistance, thereby improving the mobility. The improved mobility can lower $\tau_{transit}$. Hence, the photoconductive gain and R can be boosted further, as shown in the equations below.

$$\tau_{transit} = \frac{L^2}{\mu V_D} \dots\dots\dots (1) ,$$

$$G = \frac{\tau_{life}}{\tau_{transit}} \dots\dots\dots (2)$$

$$R = \frac{\lambda q}{hc} \times EQE \times G \dots\dots\dots (3)$$

Changes to the main manuscript: We have added sentence in the main manuscript page 11.

Original text in the manuscript page 11: This observation is consistent with the larger change in R for the same change in τ_f with varying V_{SG} as compared to varying V_{MG} (Figs. 4c, f).

Modified: This observation is consistent with the larger change in R for the same change in τ_f with varying V_{SG} as compared to varying V_{MG} (Figs. 4c, f). Although the mobility is relatively low for this device, the high responsivity reported in this study can be attributed to high photo absorption and photogating effect in WSe₂, as reported for other TMD based phototransistors.³⁷⁻³⁹ Higher mobility with lower S/D resistance via S/D contact engineering could further improve the $\tau_{transit}$, G and R values.

Reference:

37. Cao, Z., Harb, M., Lardhi, S., & Cavallo, L. Impact of Interfacial Defects on the Properties of Monolayer Transition Metal Dichalcogenide Lateral Heterojunctions. *J. Physical Chem. Letts.*, vol. 8, pp. 1664-1669, (2017).

38. Liu, E., Long, M., Zeng, J., Luo, W., Wang, Y., Pan, Y., Zhou, W., Wang, B., Hu, W., Ni, Z., You, Y., Zhang, X., Qin, S., Shi, Y., Watanabe, K., Taniguchi, T., Yuan, H., Y. Hwang, H., Cui, Y., Miao, F., & Xing, D. High Responsivity Phototransistors Based on Few-Layer ReS₂ for Weak Signal Detection. *Adv, Func. Mater.* vol. 26, pp. 1938-1944, (2016).

39. Buscema, M., O. Island, J., J. Groenendijk, D., I. Blanter, S., A. Steele, G., S. J. van der Zant, H., & Castellanos-Gomez, A. Photocurrent generation with two-dimensional van der Waals semiconductors. *Chem. Soc. Rev.* vol. 44, pp. 3691, (2015).

3. From the schematic diagram in Fig. 1a, the WSe₂ channel between the source and drain electrodes mainly depends on the middle gate, which is confirmed by Fig. 2b. When the WSe₂ channel is in the ON state, the side gates have limited influence on the electrical transport properties of the channel. When the WSe₂ channel is in the OFF state, the side gates can slightly change the doping of the WSe₂ channel. Therefore, it may be inappropriate to claim it as “a gate tunable lateral p-n junction”

The authors thank the reviewer for asking this question. Here, we still think that the device is gate tunable in the lateral SG-MG-SG direction, for the following reasons.

Dark IV of p-n junction show lateral gate tunability: Firstly, in SI Fig. 2a, we have shown gate tunable p-n action, re-presented in Figure 19 below. This shows clear gate dependent tunability from n-n to p-n junction in the lateral direction.

Fig. 19: Electrostatically doped, lateral (along the direction from MG to SG), in-plane p-n junction IV characteristics. The IV shows a change in ON and OFF currents approximately by two orders when V_{MG} is changed from 2 V to -2 V ($n_M - n_S$ to $p_M - n_S$ configuration).

The role of the gate tunable lateral p-n junction in current modulation can be described as follows:

1. **Lower current modulation in ON state:** If we look at the dark IV under ON state, there is only a small change in the current level as we change the V_{SG} . Since the doping density of the channel is high in the ON state, ability of the depletion region to encroach into the MG quasi-neutral region is limited and the change in current is small. This has been correctly pointed out by the reviewer.

2. **Large change in current under OFF state:** However, in case of OFF state operation, say at $V_{MG} = 0.5$ V and $V_{SG} = -2$ V, under $p^+_S - n_M - p^+_S$ configuration (in the lateral direction of SG-MG-SG) electrons coming in from the SG regions to the MG region can change the doping level by effectively changing the net concentration of electrons in the channel. This can further increase the n-doping of the channel and result in a higher current. But when $V_{MG} = 0.5$ V and $V_{SG} = 2$ V, under $n^+_S - n_M - n^+_S$ configuration, the effect on MG doping will be the opposite. Hence there is lower current. In both these conditions a significant modulation (>100x) in the sub-

threshold current flowing in the transverse S/D direction is seen (dark $I_D V_{MG}$ characteristics in Figure 20 below). This demonstrates a substantial role of the gate tunable lateral p-n junction.

Fig. 20: Dark state $I_D V_{MG}$ with V_{SG} variation showing a large modulation in sub-threshold current in the transverse S/D direction.

3. Modulation in photo-responsivity (R) in ON state: Further we have observed a significant modulation in R ($>10x$) under ON state of the FET. Although under dark state the carrier modulation is small and hence the change in current, the change in R under illumination is a result of efficient photo carrier separation under changing internal electric field in lateral SG-MG-SG direction with varying V_{SG} . This again proves the formation of a lateral p-n junction and its effect on transverse S/D transport. Figure 21 shows R vs V_{MG} plot with varying V_{SG} .

Fig. 21: R vs V_{MG} with varying V_{SG} showing the effect of a lateral p-n junction in I_{ph} and R in ON state.

Therefore, we believe that the side gate modulation of the OFF state/sub-threshold dark current ($>100x$) and the photoresponsivity ($>10x$) in the ON state are significant enough to justify the choice of “gate tunable lateral p-n junction”.

4. Since the two side-gates are short, the photocurrent signals generated on two side-middle interfaces are expected to have opposite polarities. Under a zero source-drain bias, the photocurrent responses of the entire device are expected to be zero. Therefore, it would be interesting to know the photocurrent signals (photo responsivity and speed) as a function of the source-drain bias.

The authors thank the reviewer for asking this interesting question.

1. Symmetry of the device leads to zero current: The device is completely symmetric in the transverse S/D direction as well as in the lateral SG-MG-SG direction. So, it is expected that for zero drain bias there is no current flow from the source-to-drain under illumination.

2. S/D contact asymmetry can cause photo current: The source and drain contacts are Schottky in nature, and if there is any asymmetry between the two contacts due to slight variations in the metal/WSe₂ interfaces and/or contact area, we may still get some photocurrent under zero S/D bias due to photocarrier separation near the contacts. This indicates the presence of a photovoltaic effect in the device.

5. “no- V_{sg} ” is a little bit confused. “no- V_{sg} ” means “ground”, “floating”, or something else.

“no- V_{SG} ” means floating, where no side gate voltage has been applied. But “no- V_{SG} ” and $V_{SG} = 0$ V are the same and have been demonstrated and supported by data in Supplementary Fig. 3a, as shown below in Figure 22.

Fig. 22: $I_D V_{MG}$ under No- V_{SG} and $V_{SG} = 0$ V at $V_D = 1$ V establish No- V_{SG} is similar to $V_{SG} = 0$ V.

So, we may consider the “no- V_{SG} ” condition as equivalent to zero volts applied in the SG.

Changes to the main manuscript: We have modified a sentence on page 6 of the manuscript as follows.

Original text on page 6 in the manuscript: A good match between $I_D V_{MG}$ transfer curves for $V_{SG} = 0$ V and no- V_{SG} applied (see Supplementary Fig. 3a) further supports the band diagram for the no- V_{SG} case in Fig. 2c, where bands are considered to be flat.

Modified: A good match between $I_D V_{MG}$ transfer curves for $V_{SG} = 0$ V and no- V_{SG} (no applied voltage to the side gates) biasing conditions (see Supplementary Fig. 3a) further supports the band diagram for the no- V_{SG} case in Fig. 2c, where bands are considered to be flat.

References:

1. Furchi, M. M., Polyushkin, D. K., Pospischil, A., & Mueller, T. Mechanisms of Photoconductivity in Atomically Thin MoS₂. *ACS Nano. Lett.* vol. 14, pp. 6165 – 6170, (2014).
2. Hsu, H. P., Lin, D. Y., Jheng, J. J., Lin, P. C., & Ko, T. S. High Optical Response of Niobium-Doped WSe₂ layered crystal. *Materials.* vol. 12, no. 1161, pp. 1-8, (2019).
3. Frisenda, R., Molina-Mendoza, A. J., Mueller, T., Castellanos-Gomez, A., & van der Zant, H. S. J. Atomically thin p–n junctions based on two-dimensional materials, *Chem. Soc. Rev.*, vol: 47, pp: 3339, (2018).
4. Zhang, W., Chiu, M. H., Chen, C. H., Chen, W., Li, L. J., & Wee, A. T. S. Role of metal contacts in high-performance phototransistors based on WSe₂ monolayers. *ACS Nano*, vol. 8, no. 8, pp. 8653–8661, (2014).
5. Lee, H., Ahn, J., Im, S., Kim, J. & Choi, W. High-Responsivity Multilayer MoSe₂ Phototransistors with Fast Response Time. *Scientific Reports.* vol. 8, no. 11545, pp. 1-7, (2018).
6. Huo, N. & Konstantatos, G. Ultrasensitive all-2D MoS₂ phototransistors enabled by an out-of-plane MoS₂ PN homojunction. *Nat. Commun.* vol. 8, no. 572, pp. 1–6, (2017).
7. Chow, P. C. Y., Matsuhisa, N., Zalar, P., Koizumi, M., Yokota, T., & Someya, T. Dual-gate organic phototransistor with high-gain and linear photoresponse. *Nat. Commun.*, vol. 9, no. 4546, pp. 1–8 (2018).
8. Xu, H., Zhu, J., Zou, G., Liu, W., Li, X., Li, C., Ryu, G. H., Xu, W., Han, X., Guo, Z., Warner, J. H., Wu, J. & Liu, H. Spatially Bandgap - Graded MoS₂(1-x)Se_{2x} Homo Junctions for Self - Powered Visible – Near - Infrared Phototransistors. *Nano-Micro Lett.*, pp. 1–14 (2020).
9. Choi, W., Cho, M. Y., Konar, A., Lee, J. H., Cha, G., Hong, S. C., Kim, S., Kim, J., Jena, D., Joo, J. & Kim, S. High-detectivity multilayer MoS₂ phototransistors with spectral response from ultraviolet to infrared. *Adv. Mater.*, vol. 24, no. 43, pp. 5832–5836, (2012).
10. Yang, Y., Huo, N. & Li, J. Gate-tunable and high optoelectronic performance in multilayer WSe₂ P–N diode, *J. Mater. Chem. C.* vol. 6, pp. 11676, (2018).
11. Hsu, H. P., Lin, D. Y., Jheng, J. J., Lin, P. C., & Ko, T. S. High Optical Response of Niobium-Doped WSe₂ layered crystal. *Materials.* vol. 12, no. 1161, pp. 1-8, (2019).
12. Stehle, Y., Meyer, H. M., Unocic, R. R., Kidder, M., Polizos, G., Datskos, P. G., Jackson, R., Smirnov, S. N. & Vlassiouk, I. V. Synthesis of Hexagonal Boron Nitride Monolayer: Control of Nucleation and Crystal Morphology, *Chem. Mater.* vol. 27, no. 23, pp. 8041-8047, (2015).

13. Wang, X., Li, Y., Zhuo, L., Zheng, J., Peng, X., Jiao, Z., Xiong, X., Han, J. & Xiao, W. Controllable growth of two-dimensional WSe₂ using salt as co-solvent, *Cryst. Eng. Comm.* vol. 20, no. 40, pp. 6267-6272, (2018).
14. Cao, Z., Harb, M., Lardhi, S., & Cavallo, L. Impact of Interfacial Defects on the Properties of Monolayer Transition Metal Dichalcogenide Lateral Heterojunctions. *J. Physical Chem. Letts.*, vol. 8, pp. 1664-1669, (2017).
15. Liu, E., Long, M., Zeng, J., Luo, W., Wang, Y., Pan, Y., Zhou, W., Wang, B., Hu, W., Ni, Z., You, Y., Zhang, X., Qin, S., Shi, Y., Watanabe, K., Taniguchi, T., Yuan, H., Y. Hwang, H., Cui, Y., Miao, F., & Xing, D. High Responsivity Phototransistors Based on Few-Layer ReS₂ for Weak Signal Detection. *Adv, Func. Mater.* vol. 26, pp. 1938-1944, (2016).
16. Buscema, M., O. Island, J., J. Groenendijk, D., I. Blanter, S., A. Steele, G., S. J. van der Zant, H., & Castellanos-Gomez, A. Photocurrent generation with two-dimensional van der Waals semiconductors. *Chem. Soc. Rev.* vol. 44, pp. 3691, (2015).

Reviewers' comments:

Reviewer #1 (Remarks to the Author):

The authors have addressed many of my concerns, especially the my comments #1 and #2. However, the following questions have not been satisfactorily addressed yet.

1. The authors still did not clearly state which electrode is grounded. The readers have to assume the source is grounded.

2. In my previous comment #3, I wanted to say $V_{SG}=0$ V is not equivalent to floating V_{SG} . In my reasoning I argued $V_{SG} = V_{MG}$ by assuming WSe_2 is ideal conductor. I appreciate that this argument is inaccurate because WSe_2 is semiconductor, as pointed by the authors. However, at least V_{SG} is tightly related to V_{MG} and $V_{SG} \neq 0$ V, namely $V_{SG}=0$ V is not equivalent to floating V_{SG} . I don't think this equivalence is related to the main point of this manuscript and therefore suggest the authors to consider not claiming this equivalence.

3. In the response to my comment #4, the authors clarified their calculation through the data from "the linear region". However, they did not state which data is this linear region. Is it Figure 2a in the region of $V_{MG} < -0.5$ V?

4. In the response to my comment #5, the authors suggested Ref. 3. However, I did not see this equation in Ref. 3. Note, my point in comment #5 is not the recombination mechanism. Instead I was questioning whether $R = \Delta n / \tau$ or $1 / \tau$. The equation employed in this manuscript leads to mismatch in quantity dimension (unit).

5. In the response to my comment #7, the authors satisfactorily answered my question. No vacant traps \rightarrow no photogating \rightarrow low G and R. This argument should be updated in the main text as well to improve the readability.

6. In the response to my comment #8 and #12, the authors stated that their primary novelty is the optimized "responsivity vs. speed trade-off". In this regard, I think the manuscript title is somehow misleading.

7. In the abstract and introduction sections, the manuscript mentioned a 2.4x and 5.3x improvement. But this improvement is not mentioned in the main text or the conclusion.

8. In the response to my comment #11, the explanation for the asymmetric IV curve is that "the drain end controls the carrier injection". I don't think so. In addition, this is in conflict to our experience with other similar devices. Although the authors think the asymmetry does not affect the main focus of this manuscript, I think such result is not proper in a high impact journal like Nat. Comm. Maybe the authors can consider redo their experiments.

Therefore I suggest reconsider this manuscript after the above points are addressed.

Reviewer #2 (Remarks to the Author):

The authors have raised several arguments to support their claims about the high D^* reported, which unfortunately remain unconvincing.

What they report is the values of D^* at the shot noise and thermal noise limit, which is the upper bound of D^* that these detectors can achieve. It is not supported that the detectors have actually achieved such D^* .

The referee insists that with NEP of 300 pW the D^* can be extracted and is 7 orders of magnitude lower than the claimed one.

The authors now state that they have managed to distinguish signal at 0.25pW which would make D^* 4 orders of magnitude lower than the claimed one. I couldn't find in the manuscript those data by the way to support detection at 300 pW and 0.25pW. It is true that this estimation of D^* is the strictest one and rarely used in scientific literature. But it is probably the most accurate. Otherwise D^* claims can be made by measuring the noise current in dark conditions at the exact same conditions (temperature, frequency, bias etc) as the responsivity. It is well documented and known in literature that 2d materials suffer from flicker or $1/f$ noise (not thermal as the authors have considered in their revision), and this can only be measured, not modeled. I do not claim that the paper should be rejected even if the D^* is lower, this is up to the editor to decide, but what I state is that the claims of the paper on this D^* value is not supported by the data.

On a different note I do remain confused how this structure achieves gain of ~ 200 with lifetime of 10 us and mobility $5\text{cm}^2/\text{Vs}$. Which equation governs gain in this structure? the authors present

equations 1 and 2 which seem not to yield the expected values. Then they state that responsivity is high due to efficient photogating and absorption (which is still debatable) but I am confused how does this support the reported gain values?

Response to reviewers' comments including revisions in the main manuscript and supporting information for the manuscript titled "Enhanced responsivity and detectivity of fast WSe₂ phototransistor enabled by electrostatically tunable in-plane P-N homojunction"

Based on the reviewer's comments, the revised manuscript and supplementary information now address the concerns raised by the reviewers and incorporates their suggestions. We have made the following key changes in this submission:

- a. Figure 6 has been modified with addition of specific detectivity data obtained from flicker noise measurements as suggested by reviewer 2.
- b. Section on detectivity has been revised to include improved explanations based on the new experimental data. (Reviewer 2)
- c. New benchmarking plots showing improved performance of our device w.r.t key performance parameters for TMD-based photodetectors- responsivity, detectivity, response time. (Reviewers 1, 2)
- d. Statements added in the manuscript based on recommendations from reviewer 1.

In the revised manuscript and supporting information, the text changes have been shown in red. Further, modified figures have also been marked with a red bounding box. In response letter the responses have been shown in red and the changes in main manuscript and Supplementary Information have been shown in blue.

Reviewer #1 (Remarks to the Author):

The authors have addressed many of my concerns, especially the my comments #1 and #2. However, the following questions have not been satisfactorily addressed yet.

1. The authors still did not clearly state which electrode is grounded. The readers have to assume the source is grounded.

We thank the reviewer for mentioning this point. We have added the following sentence to clarify that the source terminal is grounded for all our experiments.

Modified text in manuscript, page 5, line-1: “at $V_D = 1$ V where the source terminal is grounded”

2. In my previous comment #3, I wanted to say $V_{SG}=0$ V is not equivalent to floating V_{SG} . In my reasoning I argued $V_{SG} = V_{MG}$ by assuming WSe_2 is ideal conductor. I appreciate that this argument is inaccurate because WSe_2 is semiconductor, as pointed by the authors. However, at least VSG is tightly related to V_{MG} and $V_{SG} \neq 0V$, namely $V_{SG}=0$ V is not equivalent to floating V_{SG} . I don't think this equivalence is related to the main point of this manuscript and therefore suggest the authors to consider not claiming this equivalence.

We have modified the sentence in manuscript page 6, line 5 with the following one:

Original sentence in the main manuscript: “A good match between $I_D V_{MG}$ transfer curves for $V_{SG} = 0$ V and no- V_{SG} applied (no applied voltage to the side gates) biasing conditions (see Supplementary Fig. 3a) further supports the band diagram for the no- V_{SG} case in Fig. 2c, where bands are considered to be flat.”

Modified text: “The band diagram shown in Fig. 2c is considered to be flat under No- V_{SG} condition, for a simple understanding of the device operation.”

3. In the response to my comment #4, the authors clarified their calculation through the data from “the linear region”. However, they did not state which data is this linear region. Is it Figure 2a in the region of $V_{MG} < -0.5$ V?

The authors thank the reviewer for highlighting this. Firstly, we can determine the linear region from the $I_D V_D$ characteristics shown in Fig. 2a inset in the main manuscript. There it is clearly visible that when the

Fig. 17 Idealized drain characteristics of a MOSFET. For $V_D \geq V_{Dsat}$, the drain current remains constant.

Figure 1: The snapshot describing linear and saturation regions of an n-channel Si MOSFET.¹

The initial non-linearity for low V_D is due to a Schottky contact barrier. In addition to that, we can also determine linear region from $I_D V_G$ curve, which has been discussed in reference 2. Similarly, the linear region in Fig. 2a ($I_D V_{MG}$) of the main manuscript starts approximately from $V_{MG} < -0.45$ V, where I_{Dark} shows linear dependency with V_{MG} . Further the $I_D V_{MG}$ plot in linear scale is presented in Figure 2, and the straight line indicates the linear region.

Figure 2: $I_D V_{MG}$ characteristics in linear scale. The blue dotted line (a guide to eye) shows the linear region in the $I_D V_{MG}$ plot.

Reference:

1. S. M. Sze, *Semiconductor Devices Physics and Technology*, Chapter 6, pp. 191, 2002
2. Dobrescu, L., Petrov, M., Dobrescu, D., & Ravariu, C. Threshold voltage extraction methods for MOS transistors, *23rd International Semiconductor Conference*, vol. 1, pp. 371-374, (2000)

4. In the response to my comment #5, the authors suggested Ref. 3. However, I did not see this equation in Ref. 3. Note, my point in comment #5 is not the recombination mechanism. Instead I was questioning whether $R = \Delta n / \tau$ or $1 / \tau$. The equation employed in this manuscript leads to mismatch in quantity dimension (unit).

We would like to clarify this question in three parts:

(A) Equation not present in Ref. 3. of the “previous response letter”:

Ref. 3 in our previous response letter was referred to as an example of 2D-layered material-based reports that have considered SRH in their analysis. The text in Ref. 3 discusses SRH recombination.

(B) Standard SRH recombination equation: The standard textbook equation for SRH recombination from reference 3 (in this letter) is shown in the figure below and is the same as what we have used in the main manuscript.

$$\left. \frac{\partial p}{\partial t} \right|_{\text{i-thermal R-G}} \approx -\frac{\Delta p}{\tau_p} \quad \text{for holes in an } n\text{-type material} \quad (3.34a)$$

$$\left. \frac{\partial n}{\partial t} \right|_{\text{i-thermal R-G}} = -\frac{\Delta n}{\tau_n} \quad \text{for electrons in a } p\text{-type material} \quad (3.34b)$$

Equations (3.34) are the desired end result, the special-case characterization of R-G center (indirect thermal) recombination-generation. Although steady state or slowly varying conditions are implicitly assumed in the development of Eqs. (3.34), the relationships can be applied with little error to most transient problems of interest. It should be restated that a $\Delta p < 0$ is possible and will give rise to a $\partial p/\partial t|_{\text{i-thermal R-G}} > 0$. A positive $\partial p/\partial t|_{\text{i-thermal R-G}}$ simply indicates that a carrier deficit exists inside the semiconductor and generation is occurring at a more rapid rate than recombination. $\partial p/\partial t|_{\text{i-thermal R-G}}$ and $\partial n/\partial t|_{\text{i-thermal R-G}}$, it must be remembered, characterize the *net* effect of the thermal recombination and thermal generation processes.

As duly emphasized, the Eq. (3.34) relationships apply only to minority carriers and to situations meeting the low-level injection requirement. The more general steady state result⁽⁷⁾, valid for arbitrary injection levels and both carrier types in a nondegenerate semiconductor, is noted for completeness and future reference to be

$$\left. \frac{\partial p}{\partial t} \right|_{\text{i-thermal R-G}} = \left. \frac{\partial n}{\partial t} \right|_{\text{i-thermal R-G}} = \frac{n_i^2 - np}{\tau_p(n + n_1) + \tau_n(p + p_1)} \quad (3.35)$$

Figure 3: Figure shows the rate of SRH recombination when holes and electrons are minority carriers, in first two equations respectively.³ The final equation shows the generalized recombination equation.

(C) $R = \Delta n/\tau$, has a mismatch in quantity dimension (unit):

The recombination rate R for an indirect bandgap semiconductor is calculated from SRH recombination formula as shown above. $R = \Delta n/\tau$, is the simplified expression under low-level injection when electrons are the minority carriers. Here Δn , is the change in electron conc. in unit volume. τ , is the electron recombination time constant and the unit of τ is sec. So, unit of R becomes $1/\text{cm}^3\text{-sec}$, which represents the rate of change in carrier concentration per unit volume per second.

The equations for R employed in the main manuscript have the correct dimensions and units (of $1/\text{cm}^3\text{-sec}$) as per textbook definition above. Additionally, reference 3 here shows (see Figure 4 below) a clear mention of SRH recombination equation in 2D p-n junction with the same units of $1/\text{cm}^3\text{-sec}$.

This unusual interlayer recombination can be described by two possible physical mechanisms or a combination of both: (1) Shockley-Read-Hall (SRH) recombination ($R \approx n_M p_W / \tau (n_M + p_W)$) mediated by inelastic tunnelling of majority carriers into trap states in the gap; and (2) Langevin recombination ($R \approx B n_M p_W^2$) by Coulomb interaction. Here R is

Figure 4: Snapshot from reference 4 in this document shows equation for SRH recombination.

Reference:

3. R. F. Pierret, *Semiconductor Device Fundamentals*. Addison Wesley Publishing Company, Chapter 3, pp. 115, 1996.
4. Chul-Ho Lee, Gwan-Hyoung Lee, Arend M. van der Zande, Wenchao Chen, Yilei Li, Minyong Han, Xu Cui, Ghidewon Arefe, Colin Nuckolls, Tony F. Heinz, Jing Guo, James Hone & Philip Kim, Atomically thin p–n junctions with van der Waals heterointerfaces, *Nat. Nanotech.* vol. **9**, pp. 676–681, 2014.

5. In the response to my comment #7, the authors satisfactorily answered my question. No vacant traps → no photogating → low G and R. This argument should be updated in the main text as well to improve the readability.

The authors thank the reviewer for this suggestion.

We have added the following section in page 10 of the main manuscript:

Old text in the main manuscript: Under illumination, since most trap states are already filled, there is no significant imbalance in photogenerated electron and hole concentrations, leading to low τ_{life} and hence low G and R.

Modified text: Hence, under illumination, very few photogenerated electrons will get a chance to occupy the low number of remaining unfilled electron trap states. Further, we know that, 1) under photogeneration same number of electron and holes are generated, and, 2) concentrations of photogenerated electrons and holes dominate over dark state concentrations. Thus, under illumination, there will be negligible imbalance in photogenerated electron and hole concentrations. Similar results have been reported for MoSe₂ phototransistors in reference 14.

6. In the response to my comment #8 and #12, the authors stated that their primary novelty is the optimized “responsivity vs. speed trade-off”. In this regard, I think the manuscript title is somehow misleading.

Since the optimized responsivity vs speed trade-off enables higher responsivity and detectivity at the same speed, which we show at τ_f of 15.5 μs for both responsivity (13 A/W vs 17 A/W @ 5kHz operating frequency) and detectivity values (new flicker-noise based measured detectivity in this revision is enhanced by 25x with V_{SG} - 5×10^{10} Jones vs 1.26×10^{12} Jones) we believe that this data justifies the title.

Also, R vs τ_f trade-off optimization is described in the abstract. However, if the reviewer is still not agreeable, then we are willing to change the title to:

“Speed-neutral enhancement in responsivity and detectivity of WSe_2 phototransistor enabled by electrostatically tunable in-plane p-n homojunction”

7. In the abstract and introduction sections, the manuscript mentioned a 2.4x and 5.3x improvement. But this improvement is not mentioned in the main text or the conclusion.

The authors thank the reviewer for noting this. In the abstract we have mentioned the above numbers to indicate the improvement of device performance at $V_{SG} = 2\text{V}$ from No- V_{SG} condition. However, in the main text page 8, line 1, it is written as “R is higher by nearly 2x for $V_{SG} = 2\text{ V}$ in comparison to the conventional no- V_{SG} configuration”, where we had discussed about the modulation in R . Now, we have changed this to “**2.4x** from **nearly 2x**”. The detectivity part is completely rewritten with new flicker noise based measured specific detectivity data and changes have been made in abstract and introduction sections accordingly.

Original sentence in the main manuscript page 8, line 1: R is higher by nearly 2x for $V_{SG} = 2\text{ V}$ in comparison to the conventional no- V_{SG} configuration.

Modified sentence: R is higher by 2.4x for $V_{SG} = 2\text{ V}$ in comparison to the conventional no- V_{SG} configuration.

8. In the response to my comment #11, the explanation for the asymmetric IV curve is that “the drain end controls the carrier injection”. I don’t think so. In addition, this is in conflict to our experience with other similar devices. Although the authors think the asymmetry does not affect the main focus of this manuscript, I think such result is not proper in a high impact journal like Nat. Comm. Maybe the authors can consider redo their experiments. Therefore I suggest reconsider this manuscript after the above points are addressed.

The authors thank the reviewer for the comment. In the response below we show several examples of drain end-controlled carrier injection (including WSe_2 as in this work, with similar band diagrams) leading to asymmetric current in multiple high-impact (e.g. Adv. Func. Mater.) and highly reputed (Journal of Applied Physics) journals.

Drain end controls the carrier injection: To elaborate this, we split our discussion into two parts for better understanding.

- a. **Hole injection from drain end under $V_D > 0$ V:** Hole injection from drain end in WSe₂ p-type FET under $V_D > 0$ V is not unusual and has been well explained in reference 5. The following snapshot of figure 5 and associated explanations from reference 5 shows the injection of holes from drain side for $V_D > 0$ V with band diagrams similar to what we have demonstrated in Figure 6 of this response. Therefore, in good agreement with literature, our band diagrams clearly establish that for $V_G < 0$ V (ON state) and $V_D > 0$ V, hole injection occurs from drain end.

FIG. 3. Modification of band diagram with respect to change in V_G is shown for $V_D > 0$ at (a); panel (b) shows transistor characteristics at dark and under illumination for $V_D = 1$ V [the dashed red line shown in transport characteristics at (c)]. Here the filled and empty circles denote electrons and holes, respectively; black and red color represents electrically generated and photo-generated carriers, respectively. W_D and W_S denote the depletion regions of drain and source Schottky contacts, respectively. Similarly, panel (d) and (e) show modification of band diagram and transistor characteristics at dark and under illumination for $V_D = -1$ V, which is marked in (f).

Figure 5: Snapshot from reference 5, discussing about hole injection from drain terminal when the transistor is in ON state and $V_D > 0$ V. (Arnob Islam et. al., Effects of asymmetric Schottky contacts on photoresponse in tungsten diselenide (WSe₂) phototransistor, *J. Appl. Phys.*, (2017))

- b. **Drain end controls carrier injection:** Figure 6 below shows band diagrams for two different drain bias conditions ((i) $V_D = 1$ V and (ii) $V_D = -1$ V), and for both the cases, the source terminal has been grounded.

- (i) **$V_D = 1$ V:** For this condition depicted in Fig. 6(i) the barrier for holes at the drain end gets significantly thinner, cause an effective hole barrier decrease at the drain end, since the **applied drain bias majorly drops at the drain junction at which bias is applied.**⁶ This results in high injection of holes from drain to the channel.

- (ii) $V_D = -1V$: With the source terminal grounded, as shown in Fig. 6(ii), the field for hole transport will be in the direction from source to drain. But, as the source is grounded, **and the drain end is biased at -1 V, again, the maximum voltage drop will take place at the drain side.** Therefore, being grounded, the source barrier will not be modulated significantly to allow significant hole injection. Due to this large source to channel barrier, smaller number of holes will be injected from source to channel, hence, resulting in lower current at $V_D = -1V$.

Figure 6: Source-drain energy band diagrams for positive (i) and negative (ii) V_D – showing low current for negative V_D and high current for positive V_D . **Drain bias controls carrier injection.**

To further reinforce our case, we present another argument from reference 6, a study on MoS₂ n-FET system, shown in Figure 7 (a snapshot from Reference 6) which also demonstrates that the forcing terminal (which is drain terminal in our study) controls the current injection.

Figure 3. Band diagram based on two back-to-back Schottky barriers. The forward current observed for negative V_{ds} is due to the image force barrier lowering at the forced junction, while the lower (reverse) current at $V_{ds} > 0V$ is limited by the low electric field at the grounded junction. The red arrow represents the direction of the electron flow.

The above considerations lead to the qualitative energy band model shown in Figure 3, which fully explains the observed I - V curves. The key points are: i) The barriers are slightly asymmetric. ii) The gating effect of the grounded substrate shifts the band diagram closer to the junction connected to the forcing lead (the anode or drain) downward for negative V_{ds} (thus reducing the Schottky barrier) and upward for positive V_{ds} (thus increasing the Schottky barrier). iii) The applied bias drops mainly on the junction connected to the forcing lead, as proved by Kelvin force microscopy^[61] and confirmed by four-probe measurements in this study. This behavior is obvious when the forced junction is reverse biased, as it occurs for $V_{ds} < 0V$, and is caused by the gating effect, which empties the channel from free carriers close to the drain region, when the powered junction is forward biased, i.e. for $V_{ds} > 0V$. iv) The image force barrier lowering affects the current mainly when the forcing electrode is on the reverse biased junction thereby causing the high forward current at $V_{ds} \leq 0V$. v) When the grounded lead is on the reversed biased junction, the barrier is lightly affected by the image force barrier lowering. In such case, the low field at the grounded junction, which is about in a flat-band condition, strongly suppresses the current. vi) Due to the high barrier ($\geq 1eV$), the contribution of the holes, which are the minority carriers, can be neglected.

Figure 7: Snapshot from reference 6 shows the major voltage drop takes place at the forcing terminal and hence controls carrier injection. (Antonio Di Bartolomeo, et. al. Asymmetric Schottky Contacts in Bilayer MoS₂ Field Effect Transistors, *Adv. Funct. Mater.*, (2018))

Therefore, with the above explanations and band diagrams, we believe that the our argument, “**the drain end (V_D) controls the carrier injection**” holds, as for a Schottky contact transistor the applied voltage majorly drops across the forcing junction, causing different contact barriers for holes at the respective drain and source ends under two different drain bias conditions.

Asymmetry does not affect the main focus of the manuscript: The main focus of the manuscript is the extra carrier injection into the channel from side regions, due to the applied side gate voltage. This changes the carrier concentration and current in the channel with reference to no- V_{SG} condition, under illumination. As the reviewer has rightly pointed out, this phenomenon does not depend on the asymmetry of the current; rather the modulation depends on the absolute current in the channel under no- V_{SG} condition which is discussed in main manuscript Figure 2d.

In addition to this there are substantial high impact reports on 2D materials, where researchers have shown asymmetric currents in 2D transistors,⁶⁻⁷ including a study where Schottky barrier dependent asymmetric contacts have been used advantageously to separate out photogenerated carriers for high responsivity.⁸ In this regard, we believe that drain-bias dependent asymmetry in IVs is a common and acceptable feature of 2D material transistors, although not the mainstay of our manuscript.

Reference:

5. Arnob Islam and Philip X.-L. Feng, Effects of asymmetric Schottky contacts on photoresponse in tungsten diselenide (WSe₂) phototransistor, *J. Appl. Phys.*, vol. 122, pp. 085704-1 – 085704-5, (2017).
6. Antonio Di Bartolomeo, Alessandro Grillo, Francesca Urban, Laura Iemmo, Filippo Giubileo, Giuseppe Luongo, Giampiero Amato, Luca Croin, Linfeng Sun, Shi-Jun Liang, Lay Kee Ang, Asymmetric Schottky Contacts in Bilayer MoS₂ Field Effect Transistors, *Adv. Funct. Mater.*, vol. 28, no. 28, pp. 1-10, (2018).
7. Antonio Di Bartolomeo, Francesca Urban, Maurizio Passacantando, Niall McEvoy, Lisanne Peters, Laura Iemmo, Giuseppe Luongo, Francesco Romeo & Filippo Giubileo, A WSe₂ vertical field emission transistor, *Nanoscale*, vol. 11, pp. 1538-1548, (2019).
8. Wei Gao, Shuai Zhang, Feng Zhang, Peiting Wen, Li Zhang, Yiming Sun, Hongyu Chen, Zhaoqiang Zheng, Mengmeng Yang, Dongxiang Luo, Nengjie Huo, Jingbo Li, 2D

WS₂ Based Asymmetric Schottky Photodetector with High Performance, *Adv. Electron. Mater.*, vol. 2000964, pp. 1-7, 2020.

Reviewer #2 (Remarks to the Author):

The authors have raised several arguments to support their claims about the high D* reported, which unfortunately remain unconvincing.

What they report is the values of D* at the shot noise and thermal noise limit, which is the upper bound of D* that these detectors can achieve. It is not supported that the detectors have actually achieved such D*. The referee insists that with NEP of 300 pW the D* can be extracted and is 7 orders of magnitude lower than the claimed one.

The authors now state that they have managed to distinguish signal at 0.25pW which would make D* 4 orders of magnitude lower than the claimed one. I couldn't find in the manuscript those data by the way to support detection at 300 pW and 0.25pW. It is true that this estimation of D* is the strictest one and rarely used in scientific literature. But it is probably the most accurate. Otherwise D* claims can be made by measuring the noise current in dark conditions at the exact same conditions (temperature, frequency, bias etc) as the responsivity. It is well documented and known in literature that 2d materials suffer from flicker or 1/f noise (not thermal as the authors have considered in their revision), and this can only be measured, not modeled. I do not claim that the paper should be rejected even if the D* is lower, this is up to the editor to decide, but what I state is that the claims of the paper on this D* value is not supported by the data.

The authors like to thank the reviewer for bringing out this point strongly. We have the answered this question in the following parts:

(A) I couldn't find in the manuscript those data by the way to support detection at 300 pW and 0.25pW

Fig. 6c (photoswitching data) and Fig. 3d (I_{ph} vs P_{in}) in the reviewed manuscript clearly show data which support detection of input light at 0.25 pW and detectivity calculation from shot noise at 300 pW. These two plots are shown here in Figure 8a and b.

Figure 8: **a** Photoswitching data at 0.25 pW shows detection at low power. **b** I_{ph} data at 300 pW is used for calculated D^* .

(B) It is well documented and known in literature that 2d materials suffer from flicker or 1/f noise (not thermal as the authors have considered in their revision), and this can only be measured, not modeled.

Based on the reviewer’s suggestion, we have now **measured the specific detectivity (D^*) from flicker noise** and compared it with shot noise-limited specific detectivity values as shown in Figure 6 below. The flicker noise-limited D^* came down by **one to one and half order** under ON and OFF states of the transistor (-0.4 V and 0.4 V respectively) for both No- V_{SG} and $V_{SG} = 2$ V conditions. However, the gain in flicker noise-limited D^* with V_{SG} is nearly 25x as opposed to 5x for shot noise-limited D^* that we had reported in the previous revision (10x for this device as shown in Figure 6). **Hence the benefit of V_{SG} in enhancing detectivity is maintained, and actually higher, for flicker noise-limited D^* .**

Figure 9: Comparison of flicker noise- and shot noise-limited specific detectivity (D^*) under No- V_{SG} and $V_{SG} = 2$ V, for $V_{MG} = -0.4$ V and 0.4 V

Figure 10. Extraction of noise equivalent power (NEP) is shown for $V_{SG} = 2\text{ V}$ for on/off conditions of the transistor corresponding to $V_{MG} = \pm 0.4\text{ V}$. The NEP values are determined from the intersection points of the extrapolated fits of I_{ph} vs P_{in} with the measured noise floors under the same voltage conditions.

Further based on the new detectivity data we have changed the Figure 6 in the main manuscript in the following way.

- 1) Original Figure 6c- the photoswitching data at 0.25 pW, is now Figure 6a.
- 2) Original Figure 6b is replaced with newly measured 1/f noise data for No- V_{SG} and $V_{SG} = 2\text{ V}$ at different V_{MG} .
- 3) Figure 6c now shows measured flicker noise-based D^* data at different V_{MG} s, for no- V_{SG} and $V_{SG} = 2\text{ V}$.
- 4) Original Figure 6d (benchmarking) is now divided into Figures 6d (R vs τ_f) and 6e (D^* vs τ_f).

The modified Figure 6 is shown below.

Figure 11: Modified Figure 6 in the revised main manuscript. **a.** Photoswitching observed for $V_{SG} = 2$ V at extremely low laser power of 0.25 pW unlike the no- V_{SG} case. **b** Noise power spectral density (S) from flicker noise measurement for No- V_{SG} and $V_{SG} = 2$ V shows lower noise floor (by nearly 10x) for $V_{SG} = 2$ V. **c** Specific detectivity (D^*) at $V_{MG} = \pm 0.4$ V for No- V_{SG} and $V_{SG} = 2$ V shows 25x enhancement in D^* with V_{SG} . Benchmarking plots of **d.** R and **e.** D^* vs fall time (τ_f).

Changes to the manuscript:

1. Figure 6 on page 13 of the main manuscript page 13 is replaced with new Figure 6.

Old Figure 6:

New Figure 6:

2. Changes in the Figure 6 caption:

Original sentence in the Figure 6 caption: Specific detectivity (D^*) of the WSe₂ phototransistor. **a** D^* vs V_{MG} for varying V_{SG} at laser power of 33 nW. D^*_{max} ($V_{MG}=0.4$ V) increases by nearly 3x for $V_{SG} = 2$ V vs no- V_{SG} . **b** D^*_{max} for varying V_{SG} . A maximum D^*_{max} of 1.6×10^{14} Jones is obtained for $V_{SG} = 2$ V at $V_{MG} = 0.4$ V. Inset shows V_{SG} vs. D^* under DC and 5 kHz operating frequency, estimated from photoresponsivity values measured at $P_{in}=33$ nW. **c** Photoswitching observed for $V_{SG} = 2$ V (at D^*_{max}) at extremely low laser power of 0.25 pW unlike the no- V_{SG} case. **d** Benchmarking plot of R, D^* and fall time (τ_f).

Modified sentence in the Figure 6 caption: **a.** Photoswitching observed for $V_{SG} = 2$ V at extremely low laser power of 0.25 pW unlike the no- V_{SG} case. **b** Noise power spectral density (S) from flicker noise measurement for No- V_{SG} and $V_{SG} = 2$ V shows lower noise floor (by nearly 10x) for $V_{SG} = 2$ V. **c** Specific detectivity (D^*) at $V_{MG} = \pm 0.4$ V for No- V_{SG} and $V_{SG} = 2$ V shows 25x enhancement in D^* with V_{SG} . Benchmarking plots of **d.** R and **e.** D^* vs fall time (τ_f).

3. Changes in the main manuscript text:

Old text in the manuscript in page 13-14: D^* is another important performance parameter that measures the detector's ability to detect weak optical signals with reference to its own noise level. D^* is calculated using,

$$D^* = R \times \frac{\sqrt{A}}{(2qI_{dark})^{\frac{1}{2}}} \dots\dots\dots (4)$$

where, A is the channel area over the MG. Here shot noise = $(2qI_{dark})^{1/2}$ has been calculated using the dark current and considered to be the major contributor to D^* .^{13, 31, 34} Fig 6a shows D^* vs V_{MG} at 33 nW of laser power, for V_{SG} varying from -2 V to 2 V and for no- V_{SG} . D^* increases monotonically with V_{SG} , with a change of ~1.5 order (5×10^9 Jones for $V_{SG} = -2$ V to 1×10^{11} Jones at $V_{SG} = 2$ V) for maximum R at $V_{MG} = -0.5$ V. A 3x enhancement in maximum detectivity (D^*_{max}) is obtained for $V_{SG} = 2$ V over no- V_{SG} at $V_{MG} = 0.4$ V. When the SG-MG-SG band configuration goes from $p_S-p_M-p_S$ to $n_S-p_M-n_S$, the dark current noise (shot noise) decreases due to a decrease in I_{dark} (Fig. 2b) and an increase in R (Fig. 3c). Therefore, increasing V_{SG} gives a two-fold enhancement in D^* . Since shot noise is not the only contributor to detector noise, D^* calculations accounting for thermal noise are shown in Supplementary Information section 10. Comparison of D^* calculated from both sources of noise shows that shot noise is the more dominant source amongst the two. Fig. 6b shows D^*_{max} for varying V_{SG} at $V_{MG} = 0.4$ V. Inset shows the effect of V_{SG} on D^* under DC and 5 kHz operating frequency, estimated from photoresponsivity values measured at $P_{in}=33$ nW. These plots clearly demonstrate the advantage of the three-gate structure under

different operating frequencies. A maximum D^* of 1.6×10^{14} Jones is obtained at $P_{in} = 300$ pW for $V_{SG} = 2$ V, one of the highest reported values for 2D materials-based phototransistors. This can be attributed to the combined effect of better separation of photogenerated e-h pairs resulting in high gain and low noise level when the device operates at $V_{SG} = 2$ V. Further, to demonstrate the enhancement in detectivity, photoswitching measurements were carried out for $V_{SG} = 2$ V and no- V_{SG} at 1 Hz frequency for an extremely low laser power of 0.25 pW (Fig. 6c). Clear switching behaviour was observed for $V_{SG} = 2$ V unlike the no- V_{SG} case, directly reinforcing the benefit of a lateral p-n junction in enhancing the detectivity of the phototransistor.

Finally, Fig. 6d benchmarks the responsivity, detectivity and fall time, τ_f (which governs the photodetector speed), for the WSe_2 phototransistor reported in this work with other reported 2D materials-based phototransistors. The comparison is exclusively with single-TMD (MoS_2 , $MoSe_2$, WSe_2 , $InSe$, SnS_2 etc.) phototransistors. P-n heterostructures or materials such as graphene and black phosphorus have been excluded. A clear trade-off between R and τ_f is observed over a wide range of R (10^{-3} A/W to 10^5 A/W) and τ_f (μs to seconds). When R is high, τ_f also tends to be very high^{10,34}, indicating slow detector speed. On the other hand, fast speed (low τ_f) phototransistors show lower R ^{22,24}. An overall improvement in performance, with high detectivity (5.4×10^{13} Jones), high R (94 A/W) and low τ_f (14 μs), makes this work one of the highest reported R and D^* with τ_f in the μs range, enabled by a lateral p-n junction.

Modified text: Figure 6a demonstrates low power photoswitching data for measurements carried out for $V_{SG} = 2$ V and no- V_{SG} at 1 Hz frequency for a laser power of 0.25 pW. Clear switching behaviour was observed for $V_{SG} = 2$ V unlike the no- V_{SG} case, reinforcing the benefit of a lateral p-n junction in enhancing the detectivity of the phototransistor. D^* is another important parameter that represents the detector's ability to measure signals with reference to its own noise level. Here in this study D^* is calculated using

$$D^* = \frac{\sqrt{A}}{NEP} \dots\dots\dots (4)$$

In the above equation A is the channel area over the MG. NEP is the noise equivalent power, which is extracted from the noise power spectral density (S) data at 1 Hz shown in Figure 6b and I_{ph} vs P_{in} plots (shown in Supplementary Figure 14). The S data was determined by calculating the Fourier transform of the dark current time traces measured for fixed drain and varying V_{MG} and V_{SG} voltages, on a similar device (see Supplementary Information section 10). Fig. 6c shows the comparison of D^* for No- V_{SG} and $V_{SG} = 2$ V, at $V_{MG} = +/- 0.4$ V (off/on conditions). A 25x enhancement in detectivity is obtained for $V_{SG} = 2$ V, at $V_{MG} = 0.4$ V. This can be attributed to the decrease in S by an order of magnitude at 1 Hz for $V_{SG} = 2$ V when compared to No- V_{SG} . This further minimizes the NEP and hence increases detectivity. A maximum detectivity value of 1.1×10^{12} Jones is obtained. Finally, Figures 6d and 6e benchmark

responsivity and detectivity values vs speed respectively, for the WSe_2 phototransistor reported in this work with other single 2D material-based (such as MoS_2 , MoSe_2 , WSe_2 , InSe , SnS_2 etc.) phototransistors that report the values of all three parameters (R , D^* and τ_f). A clear trade-off between R and τ_f is observed over a wide range of R (10^{-3} A/W to 10^5 A/W) and τ_f (μs to seconds) (Figure 6d). When R is high, τ_f also tends to be very high^{10,34}, indicating slow detector speed. On the other hand, fast speed (low τ_f) phototransistors show lower R ^{22,24}. Our work shows the benefit of high R without slowing down the phototransistor by employing a lateral p-n homojunction. Figure 6e benchmarks both flicker noise- and shot noise-limited D^* vs τ_f values for the same set of reports as shown in Figure 6d. With high flicker noise-limited (1×10^{12} Jones) and shot noise-limited (5×10^{13} Jones) D^* , high R (94 A/W) and low τ_f (14 μs) data, this work reports one of the best combination of R , D^* and τ_f values, enabled by a lateral p-n junction.

Changes to the Supplementary Information: We have added the following figure in Supplementary section 11 (Figure 15 in Supplementary) which compares the value of measured D^* from flicker noise to the calculated D^* from shot noise and shows the method to extract NEP (Figure 14 in Supplementary).

Supplementary Figure 15. A comparison of flicker noise- and shot noise-limited specific detectivity (D^*) under No- V_{SG} and $V_{SG} = 2\text{V}$, for $V_{MG} = -0.4$ V and 0.4 V

Supplementary Figure 14. Extraction of noise equivalent power (NEP) is shown for $V_{SG} = 2$ V for on/off conditions of the transistor corresponding to $V_{MG} = \pm 0.4$ V. The NEP values are determined from the intersection points of the extrapolated fits of I_{ph} vs P_{in} with the measured noise floors under the same voltage conditions.

On a different note I do remain confused how this structure achieves gain of ~ 200 with lifetime of 10 ns and mobility $5\text{cm}^2/\text{Vs}$. Which equation governs gain in this structure? the authors present equations 1 and 2 which seem not to yield the expected values. Then they state that responsivity is high due to efficient photogating and absorption (which is still debatable) but I am confused how does this support the reported gain values?

The authors thank the reviewer for asking this.

Firstly, the equation that governs the photoconductive gain, $G = \tau_{life}/\tau_{transit}$, is embedded in the main manuscript text, page 12, line 1 (“Fig. 5a shows that τ_{life} and $G = \tau_{life}/\tau_{transit}$ increase when V_{SG} is increased from 0 to 2 V at fixed $V_{MG} = -0.5$ V.”).

The reported mobility ($5\text{cm}^2/\text{V-s}$) was extracted from dark current under No- V_{SG} condition. Under dark, a large number of trap states are empty leading to low mobility because of enhanced scattering. Under illumination the carrier concentration increases in the channel due to photogeneration and extra carrier injection from side gate, that helps to fill up these empty trap states, and they are also responsible for photogating in the device at the same time.^{9,10} Hence, due to the unavailability of such empty trap states under high carrier concentration, effective mobility in the channel increases. This leads to higher mobility ($\sim 19\text{cm}^2/\text{V-s}$) under illumination in comparison to the dark state. The increased mobility gives a transit

time of ~ 52 ns for a channel length of ~ 10 μm and $V_D = 1$ V that leads to a high gain value of ~ 200 for a lifetime value of ~ 10 μs .

Reference:

9. Bharti, D., Raghuwanshi, V., Varun, I., Mahato, A. K., & Tiwari, S. P. Photo-Response of Low Voltage Flexible TIPS-Pentacene Organic Field-Effect Transistors, *IEEE Sensors Journal*, vol. 17, no. 12, pp. 3689-3697, 2017.
10. Lee, H., Ahn, J., Im, S., Kim, J. & Choi, W. High-Responsivity Multilayer MoSe_2 Phototransistors with Fast Response Time. *Sci. Rep.*, vol. 8, pp. 11545, (2018).

REVIEWER COMMENTS

Reviewer #1 (Remarks to the Author):

Comments

I think the present version look much better. However, the illustrations and writing still needs to be improved. For example,

1. Fig. 4b and e should be enlarged and with less but more eye-catching contents.
2. "Patterned Fermi level modulation" could be better than "In-plane p-n homojunction". There isn't sufficient introduction to "electrostatic doping" and "in-plane homojunction". Therefore several related refs should be included, e.g., *Nanoscale*, 2019, 11, 22531-22538; *ACS Photonics*, 2017, 4, 823-829; *J. Mater. Chem. C*, 2019,7, 1182-1187.
3. Explanations to the working mechanism is too lengthy. Every word here should be tightly related to 1) responsivity, 2) detectivity and 3) speed. Especially, an explanation to the considerable D^* should be provided. A possible reason comes from the asymmetric output characteristics. The less related contents (e.g. calculation method of gain, transit time, EQE) should be moved to the supplementary materials.
4. I suggest providing a simulation to the working mechanism.

Reviewer #2 (Remarks to the Author):

The authors have taken my comments in serious consideration and have made important changes and clarifications in the manuscript.

First re: my comments about the lack of evidence of photoswitching at 0.25 pW, that was my mistake from the misinterpretation of their original Fig.6. Now it is clear but I would recommend that the label that states the power is moved to the top of the figure to be immediately obvious than both panels refer to measurements at 0.25 pW.

Then, although the authors have now taken measurements of noise, they still insist to call the D^* as shot noise and flicker noise, whereas in reality the former is the theoretically maximum achievable D^* whereas the latter is the actual D^* of the devices that the authors report...so I would ask to make this obvious to the labeling and the description. Related to this the authors state that the dominant noise is the shot noise, but I don't agree as their actual D^* is almost 2 orders of magnitude lower than what would be achievable only with shot noise...

Can the authors plot in Fig.6 b where they show the measured noise PSD also the theoretical noise PSDs from the shot noise, for the different values of V_{gm} and V_{gs} ? Then it will become obvious to what extent the device noise is governed by shot noise or flicker.

A last point is about the time response, so from Fig.6 it seems that the device is not as fast as the authors state in other figures. Obviously the device gets slower with lower incident power, which is expected, so it is questionable to what extent this work achieved simultaneously fast response and very high sensitivity...I think this topic should be discussed and elaborated further to avoid overselling, unless I am wrong.

Other than that I think the work is fine to publish.

Response to reviewers' comments including revisions in the main manuscript and supporting information for the manuscript titled "Enhanced responsivity and detectivity of fast WSe₂ phototransistor enabled by electrostatically tunable in-plane P-N homojunction"

Based on the reviewer's comments the revised manuscript and supplementary information now addresses all the concerns raised by the reviewers and incorporates their suggestions. We have made the following key changes in this submission:

- a. Added a sentence and references to justify electrostatic doping (in Introduction). (reviewer 1)
- b. Changed title of the paper to bring out the device architecture and physics more precisely. (reviewer 1)
- c. Modified device operation part in the main manuscript to make it more concise. Also, moved calculation methodologies for gain, lifetime and EQE to supplementary information (section 10). (reviewer 1)
- d. Added a TCAD simulated three gate structure, simulated band diagram and hole density profiles (dark and illumination) in supplementary section 2 to strengthen our claims on device operation. (reviewers 1)
- e. Modified sentences regarding D^* and clearly demarked shot noise and flicker noise D^* in Fig. 6e to eliminate any confusion regarding D^* . (reviewer 2)

In the revised manuscript and supporting information, the text changes have been shown in blue and deleted text has been shown in red strikethrough. Further, modified figures have also been marked with a red bounding box. In response letter the responses have been shown in red and the changes in main manuscript and Supplementary Information have been shown in blue.

Reviewer #1 (Remarks to the Author):

Comments

I think the present version look much better. However, the illustrations and writing still needs to be improved. For example,

1. Fig. 4b and e should be enlarged and with less but more eye-catching contents.

The authors thank the reviewer for the suggestion.

We have now modified Fig. 4b and moved Fig. 4e to the supplementary section 8 (Supplementary Information sections is now rearranged from the previous version.)

Original Fig.4 in the main manuscript:

Modified Fig. 4:

Along with this we have changed the relevant text in the main manuscript page 10.

Original sentences:

These trends, consistent with published reports,^{8,9,14} are explained through Fig. 4e depicting V_{MG} at (1) -0.5 V, and, (2) 0.4 V. For $V_{MG} = 0.4$ V, the channel is n-type, mostly filled with electrons with holes acting as minority carriers. These electrons fill up most of the available long-lived electron trap states near the conduction band edge in dark condition. Hence, under illumination, very few photogenerated electrons will get a chance to occupy the low number of remaining unfilled electron trap states. Further, we know that, 1) under photogeneration same number of electron and holes are generated, and, 2) concentrations of photogenerated electrons and holes dominate over dark state concentrations. Thus, under illumination, there will be negligible imbalance in photogenerated electron and hole concentrations. Similar results have been reported for MoSe₂ phototransistors in reference 14. When the laser is turned off, the photogenerated electrons and holes recombine fast, leading to low τ_f (high speed). For $V_{MG} = -0.5$ V, the S/D channel has high hole concentration and low electron density (minority carriers). Hence most trap states near the conduction band are empty and available for photogenerated electrons to occupy. This leads to a photogating effect that increases τ_{life} for photogenerated holes resulting in larger G and R compared to $V_{MG} = 0.4$ V. When the laser is turned off, the trapped electrons slowly get de-trapped from the long-lived trap states to recombine with the photogenerated holes, which leads to large τ_f (slow speed).

Modified:

These trends, consistent with published reports,^{8,9,14} are explained in detail in Supplementary Information section 8, and band diagram schematics under light for V_{MG} at (1) -0.5 V, and, (2) 0.4 V are shown in Supplementary Information Fig. 10.

2. “Patterned Fermi level modulation” could be better than “In-plane p-n homojunction”. There isn’t sufficient introduction to “electrostatic doping” and “in-plane homojunction”. Therefore several related refs should be included, e.g., *Nanoscale*, 2019, 11, 22531-22538; *ACS Photonics*, 2017, 4, 823–829; *J. Mater. Chem. C*, 2019,7, 1182-1187.

The authors thank the reviewer for the suggestion.

- a) **Patterned Fermi level modulation” could be better than “In-plane p-n homojunction:** We believe that the reviewer’s suggestion is more generic since “Patterned Fermi level modulation” can also be obtained with gated heterostructures and non in-plane geometries (like a lateral stacked heterostructure) and Fermi-level modulation may need not necessarily lead to a p-n junction configuration. We believe that the p-n junction physics is key to this study. Therefore, to capture the essence of our study more precisely which is based on the device architecture and p-n junction physics, we have replaced “In-plane p-n homojunction” with “in-plane **lateral** p-n homojunction” in the title. Therefore, the title has been slightly modified to “**Enhanced responsivity and detectivity of fast WSe₂ phototransistor using electrostatically tunable lateral in-plane p-n homojunction**”. Also, we have replaced “enabled by” by “using”.
- b) **Related references for electrostatic doping are included:** We have included the following references in the introduction of the main manuscript, to introduce electrostatic doping.

Changes to the main manuscript page 2:

Original sentence where references are included:

This work relies on electrostatic doping due to its ease of tunability, reversibility and area selectivity.⁴²⁻⁴⁴

References (16-18):

1. Li, J., Chen, X., Xiao, Y., Li, S., Zhang, G., Diao, X., Yan, H. & Zhang, Y. A tunable floating-base bipolar transistor based on a 2D material homojunction realized using a solid ionic dielectric material. *Nanoscale*, vol. 11, pp. 22531-22538, (2019).

2. Bie, YQ., Grosso, G., Heuck, M., Furchi, M.M., Cao, Y., Zheng, J., Bunandar, D., Navarro-Moratalla, E., Zhou, L., K. Efetov, D., Taniguchi, T., Watanabe, K., Kong, J., Englund, D. & Jarillo-Herrero, P. A MoTe₂-based light-emitting diode and photodetector for silicon photonic integrated circuits. *Nature Nanotech* vol. 12, pp. 1124–1129, (2017).
3. Frisenda, R., J. Molina-Mendoza, A., Mueller, T., Castellanos-Gomez, A. & S. J. van der Zant, H. Atomically thin p–n junctions based on two-dimensional materials, *Chem., Soc., Rev.* vol. 47, pp. 3339-3358, (2018).

3. Explanations to the working mechanism is too lengthy. Every word here should be tightly related to 1) responsivity, 2) detectivity and 3) speed. Especially, an explanation to the considerable D^* should be provided. A possible reason comes from the asymmetric output characteristics. The less related contents (e.g. calculation method of gain, transit time, EQE) should be moved to the supplementary materials.

We have addressed the reviewer's suggestions below in three separate parts (concise working mechanism, explanation of considerable D^* and less related contents moved to SI).

- i) **Tight explanation of the working mechanism:** In the modified manuscript, we have explained the working mechanism more concisely, as follows.

Original text in the manuscript page no 7:

- a) From the section “**Phototransistor operation: (i) Dark characteristics and Photoresponsivity**”: Fig. 2a shows p-type WSe₂ FET transfer characteristics ($I_D V_{MG}$) at $V_D = 1$ V where the source terminal is grounded, under dark, when V_{SG} is not applied. An on/off current ratio of $\sim 10^5$ and hole mobility of $5 \text{ cm}^2/Vs$ indicate good MG control and hole transport respectively. The output characteristics ($I_D V_D$) in the inset indicate Schottky barrier-dominated transport across the S/D contacts to the WSe₂ channel over the MG. High drain on-current (in nA) for positive voltage bias at the metal source contact to the WSe₂ channel, and low off-current ($\sim pA$) confirm the formation of a p-type Schottky contact between Pt and WSe₂ that is favourable for hole injection. Dark $I_D V_{MG}$ characteristics in Fig. 2b show a monotonous decrease in drain current as V_{SG} changes from -2 V to 2 V. This is due to increasing depletion of the hole concentration, resulting in the formation of a lateral p_M-n_S junction across the MG and SG WSe₂ regions, with increasing (more positive) V_{SG} . Hence the width of the conducting S/D channel decreases, with increasing V_{SG} leading to a reduction in current. For $V_{MG} = 0$ V, when V_{SG} is not applied, a $p_S-p_M-p_S$ configuration is formed in WSe₂ along the lateral SG-MG-SG direction, as shown in Fig. 2c. Under this condition, since

the S/D contacts are patterned with an overlap with only the MG, the S/D conduction channel width is defined by the physical width of the MG (2 μm).

The band diagram shown in Fig. 2c is considered to be flat under No- V_{SG} condition, for a simple understanding of the device operation. On the other hand, when $V_{\text{SG}} = 2 \text{ V}$ is applied for $V_{\text{MG}} = 0 \text{ V}$, the band configuration changes from $p_{\text{S}}-p_{\text{M}}-p_{\text{S}}$ to $n_{\text{S}}-p_{\text{M}}-n_{\text{S}}$, as $V_{\text{SG}} = 2 \text{ V}$ induces electrons in both SG WSe_2 regions making them n-type. This $p_{\text{M}}-n_{\text{S}}$ configuration between the MG and both SGs, leads to the formation of a depletion region parallel to the length of the S/D channel. As the space charge region (depletion) encroaches the MG area, it effectively decreases the conduction channel width to $< 2 \mu\text{m}$. Hence the absolute S/D current (I_{dark}) decreases in the dark state with positive V_{SG} . Based on this reasoning, band alignment for different V_{SG} conditions for a fixed $V_{\text{MG}} = 0 \text{ V}$ is depicted in Fig. 2c to further understand the trend in I_{dark} when V_{SG} is tuned from -2 V to no- V_{SG} to 2 V. Fig 2d shows the relative change in dark current with respect to the channel carrier concentration (p_{WSe_2}). It can be seen that SG modulation of the channel current is higher for lower p_{WSe_2} in the range of 10^{13} to 10^{15} cm^{-3} (sub-threshold region), as it is easier to deplete at lower S/D channel doping density. The calculation of p_{WSe_2} is shown in Supplementary Information section 3.

Photoresponse of the WSe_2 transistor was obtained under 532 nm laser illumination. Light was incident on the entire device area to ensure photogeneration of electron-hole (e-h) pairs in WSe_2 over the MG as well as the adjacent SG regions. Generation of e-h pairs gives rise to an additional current and the total current under illumination is referred to as the light current (I_{light}). Photocurrent ($I_{\text{ph}} = I_{\text{light}} - I_{\text{dark}}$) is the additional current collected by the contacts due to photogeneration of e-h pairs. The I_{ph} vs V_{MG} plot in Fig. 3a shows a monotonous increase in I_{ph} with V_{SG} increasing from -2 V to 2 V in the ON state. This trend in I_{ph} with V_{SG} is explained through band diagrams in Fig. 3b. As discussed earlier, when $V_{\text{MG}} = 0 \text{ V}$ and $V_{\text{SG}} = 2 \text{ V}$ ($n_{\text{S}}-p_{\text{M}}-n_{\text{S}}$ configuration) lateral internal electric fields are generated in WSe_2 due to formation of the two depletion regions. The electric fields (from SG to MG) in the lateral depletion regions drive electrons photogenerated near and in the MG depletion region of WSe_2 out towards the SG regions. Similarly, holes photogenerated near and in the SG depletion regions of WSe_2 are driven towards the MG channel. Hence, the S/D channel now carries the original as well as additional photogenerated holes supplied from the SG regions to the channel. These additional holes are also collected by the S/D contacts due to the presence of a source-to-drain electric field, thereby providing additional photocurrent in the $n_{\text{S}}-p_{\text{M}}-n_{\text{S}}$ configuration. Even a single SG will show a similar but smaller increase in photocurrent through the $n_{\text{S}}-p_{\text{M}}$ configuration under ON state.

- b) From the section “**Phototransistor operation: (ii) Temporal photoresponsivity and detectivity**”: These trends, consistent with published reports,^{8,9,14} are explained through Fig. 4e depicting V_{MG} at (1) -0.5 V, and, (2) 0.4 V. For $V_{MG} = 0.4$ V, the channel is n-type, mostly filled with electrons with holes acting as minority carriers. These electrons fill up most of the available long-lived electron trap states near the conduction band edge in dark condition. Hence, under illumination, very few photogenerated electrons will get a chance to occupy the low number of remaining unfilled electron trap states. Further, we know that, 1) under photogeneration same number of electron and holes are generated, and, 2) concentrations of photogenerated electrons and holes dominate over dark state concentrations. Thus, under illumination, there will be negligible imbalance in photogenerated electron and hole concentrations. Similar results have been reported for MoSe₂ phototransistors in reference 14. When the laser is turned off, the photogenerated electrons and holes recombine fast, leading to low τ_f (high speed). For $V_{MG} = -0.5$ V, the S/D channel has high hole concentration and low electron density (minority carriers). Hence most trap states near the conduction band are empty and available for photogenerated electrons to occupy. This leads to a photogating effect that increases τ_{life} for photogenerated holes resulting in larger G and R compared to $V_{MG} = 0.4$ V. When the laser is turned off, the trapped electrons slowly get de-trapped from the long-lived trap states to recombine with the photogenerated holes, which lead to large τ_f (slow speed). Fig. 4e plots the R vs τ_f trade-off indicating a slope of 4.2 A/Ws.

Modified text in the main manuscript:

- a) From the section “**Phototransistor operation: (i) Dark characteristics and Photoresponsivity**”: Fig. 2a shows p-type WSe₂ FET transfer characteristics ($I_D V_{MG}$) at $V_D = 1$ V where the source terminal is grounded, under dark, when V_{SG} is not applied. An on/off current ratio of $\sim 10^5$ and hole mobility of 5 cm²/Vs indicate good MG control and hole transport respectively. The output characteristics ($I_D V_D$) in the inset indicate Schottky barrier-dominated transport across the S/D contacts to the WSe₂ channel over the MG. High drain on-current (in nA) for positive voltage bias at the metal source contact to the WSe₂ channel, and low off-current (\sim pA) confirm the formation of a p-type Schottky contact between Pt and WSe₂ that is favourable for hole injection. Dark $I_D V_{MG}$ characteristics in Fig. 2b show a monotonous decrease in drain current as V_{SG} changes from -2 V to 2 V. This is due to increasing depletion of the hole concentration, resulting in the formation of a lateral p_M-n_S junction across the MG and SG WSe₂ regions, with increasing (more positive) V_{SG} . Hence the width of the conducting S/D channel decreases, leading to a reduction in current. For $V_{MG} = 0$ V, when V_{SG} is not applied, a p_S-p_M-p_S configuration is formed in WSe₂ along the lateral SG-MG-SG direction, as shown in Fig. 2c. Under this condition, since the S/D contacts are

patterned with an overlap with only the MG, the S/D conduction channel width is defined by the physical width of the MG (2 μm).

The band diagram shown in Fig. 2c is considered to be flat under No- V_{SG} condition, for a simple understanding of the device operation. On the other hand, when $V_{\text{SG}} = 2 \text{ V}$ is applied for $V_{\text{MG}} = 0 \text{ V}$, the band configuration changes from $p_{\text{S}}-p_{\text{M}}-p_{\text{S}}$ to $n_{\text{S}}-p_{\text{M}}-n_{\text{S}}$, as $V_{\text{SG}} = 2 \text{ V}$ induces electrons in both SG WSe_2 regions making them n-type. Therefore, with $p_{\text{M}}-n_{\text{S}}$ configuration between the MG and both SGs, as the space charge region (depletion) encroaches the MG area, it effectively decreases the conduction channel width to $< 2 \mu\text{m}$. Hence the absolute S/D current (I_{dark}) decreases in the dark state with positive V_{SG} . Based on this reasoning, band alignment for different V_{SG} conditions for a fixed $V_{\text{MG}} = 0 \text{ V}$ is depicted in Fig. 2c to further understand the trend in I_{dark} when V_{SG} is tuned from -2 V to no- V_{SG} to 2 V . Fig 2d shows the relative change in dark current with respect to the channel carrier concentration (p_{WSe_2}). It can be seen that SG modulation of the channel current is higher for lower p_{WSe_2} in the range of 10^{13} to 10^{15} cm^{-3} (sub-threshold region), as it is easier to deplete at lower S/D channel doping density. The calculation of p_{WSe_2} is shown in Supplementary Information section 3.

Photoresponse of the WSe_2 transistor was obtained under 532 nm laser illumination. Light was incident on the entire device area to ensure photogeneration of electron-hole (e-h) pairs in WSe_2 over the MG as well as the adjacent SG regions. From the total current upon illumination (I_{light}), the photocurrent is obtained as $I_{\text{ph}} = I_{\text{light}} - I_{\text{dark}}$. The I_{ph} vs V_{MG} plot in Fig. 3a shows a monotonous increase in I_{ph} with V_{SG} increasing from -2 V to 2 V in the ON state. This trend in I_{ph} with V_{SG} is explained through band diagrams in Fig. 3b. When $V_{\text{MG}} = 0 \text{ V}$ and $V_{\text{SG}} = 2 \text{ V}$ ($n_{\text{S}}-p_{\text{M}}-n_{\text{S}}$ configuration) electric fields (from SG to MG) in the lateral depletion regions drive electrons photogenerated near and in the MG depletion region of WSe_2 out towards the SG regions. Similarly, holes photogenerated near and in the SG depletion regions of WSe_2 are driven towards the MG channel. These additional holes provide additional photocurrent in the $n_{\text{S}}-p_{\text{M}}-n_{\text{S}}$ configuration. Even a single SG will show a similar but smaller increase in photocurrent through the $n_{\text{S}}-p_{\text{M}}$ configuration under ON state.

- b) From the section “**Phototransistor operation: (ii) Temporal photoresponsivity and detectivity**”: These trends, consistent with published reports,^{8,9,14} are explained in detail in Supplementary Information section 8, and band diagram schematics under light for V_{MG} at (1) -0.5

V, and, (2) 0.4 V are shown in Supplementary Information Fig. 10. Fig. 4e plots the R vs τ_f trade-off indicating a slope of 4.2 A/Ws.

- ii) **Explanation for the considerable D***: Regarding D*, firstly, we achieve 25x enhancement in D* under $V_{MG} = 0.4$ V, $V_{SG} = 2$ V compared to the no- V_{SG} condition, where the S/D voltage condition was fixed. Notably, fixed S/D voltage for both side gate conditions keeps a constant Schottky barrier for carrier injection for both V_{SG} conditions. Only positive V_{SG} modulates the channel width by extending the depletion inside the transport channel which causes lowering in transistor dark current, hence S (noise PSD). Additionally, under illumination R increases due to more efficient carrier separation. The combination of the above two factors helps to achieve considerable D* under n-p-n configuration.

We have changed the sentences in the following way to better explain the considerable D*.

Original sentence in the main manuscript page 12:

A 25x enhancement in D* is obtained for $V_{SG} = 2$ V, at $V_{MG} = 0.4$ V. This can be attributed to the decrease in S by an order of magnitude at 1 Hz for $V_{SG} = 2$ V when compared to No- V_{SG} . This further minimizes the NEP and hence increases D*.

Modified sentence in main manuscript page 11:

A 25x enhancement in D* is obtained for $V_{SG} = 2$ V, at $V_{MG} = 0.4$ V. This can be attributed to a substantial reduction in NEP due to i) enhanced R (~ 1.45x) resulting from efficient photogenerated hole separation, as well as, ii) decrease in I_{dark} (greater than 10x, see Supplementary Information section 14) resulting in a decrease in S by nearly an order of magnitude^{12,13} at 1 Hz for $V_{SG} = 2$ V when compared to No- V_{SG} . The reduction in NEP due to both these factors increases D*. We have added the calculation related to gain, transit time and EQE in the supplementary section 12.

- iii) **The less related contents (e.g. calculation method of gain, transit time, EQE) moved to the supplementary materials**: We have now moved the above mentioned sections to the supplementary information. Changes are described below.

Original section removed from the main manuscript to supplementary:

- a) τ_{life} values were extracted from exponential fits of the temporal decay characteristics. $\tau_{transit}$ is the time taken for a hole to travel from the source to the drain contact along the S/D channel. $\tau_{transit}$ was obtained using,

$$\tau_{\text{transit}} = \frac{L^2}{\mu V_D} \dots\dots\dots (2)$$

where ‘L’ is the S/D channel length and μ is channel carrier mobility. Mobility was calculated from the $I_D V_G$ characteristics for no- V_{SG} and positive V_{SG} conditions (Fig. 2a, b) using,

$$\mu = \frac{L}{W \times V_D (\epsilon_0 \epsilon_r / d)} \times \frac{dI_D}{dV_G} \dots\dots\dots (3)$$

Here W and d are the channel width and dielectric (hBN) thickness, respectively. ϵ_0 and ϵ_r are the dielectric constants for vacuum and hBN.

Fundamentally, hole mobility in the WSe₂ channel should not change with V_{SG} ; however, the mobility values calculated from the on-currents show some variation with V_{SG} due to the varying channel width. As a result, τ_{transit} also varies with V_{SG} .

b) EQE is calculated using,

$$\text{EQE} = \frac{R}{G} \times \frac{hc}{\lambda q} \dots\dots\dots (4)$$

where, h is Planck’s constant, c is the speed of light, λ is the wavelength of incident light and q is electronic charge.

Changes to the main manuscript:

Supplementary Information section 10 details the methodology for calculating carrier lifetimes (τ_{life}), transit times (τ_{transit}), photoconductive gain (G) and external quantum efficiency (EQE).

Newly added Supplementary section 10:

This section describes the methodology to calculate carrier lifetime (τ_{life}), transit time (τ_{transit}), photoconductive gain (G) and external quantum efficiency (EQE).

τ_{life} values were extracted from exponential fits of the temporal decay characteristics, as shown in section 4 of this Supplementary Information. τ_{transit} is the time taken for a hole to travel from the source to the drain contact along the S/D channel. τ_{transit} was obtained using,

$$\tau_{\text{transit}} = \frac{L^2}{\mu V_D} \dots\dots\dots (1)$$

where ‘L’ is the S/D channel length and μ is channel carrier mobility. Mobility was calculated from the $I_D V_G$ characteristics for no- V_{SG} and positive V_{SG} conditions (Fig. 2a, b) using,

$$\mu = \frac{L}{W \times V_D (\epsilon_0 \epsilon_r / d)} \times \frac{dI_D}{dV_G} \dots\dots\dots (2)$$

Here W and d are the channel width and dielectric (hBN) thickness, respectively. ϵ_0 and ϵ_r are the dielectric constants for vacuum and hBN.

Fundamentally, hole mobility in the WSe₂ channel should not change with V_{SG} ; however, the mobility values calculated from the on-currents show some variation with V_{SG} due to the varying channel width. As a result, $\tau_{transit}$ also varies with V_{SG} .

EQE is calculated using,

$$EQE = \frac{R}{G} \times \frac{hc}{\lambda q} \dots\dots\dots(3)$$

where, h is Planck's constant, c is the speed of light, λ is the wavelength of incident light and q is electronic charge.

4. I suggest providing a simulation to the working mechanism.

The authors thank the reviewer for the suggestion. We have now included a section in newly added Supplementary section 2, which shows the simulated three gate structure (without S/D channel) using TCAD (Sentaurus from Synopsys) and the effect of electrostatic doping under dark and illuminated conditions with fixed V_{MG} and different V_{SG} conditions. The newly included Supplementary section is presented below.

Addition in Supplementary section 2:

Supplementary Figure 2. TCAD simulations to illustrate device operation. **a** A representative schematic of the simulated device structure. **b** Band diagrams under fixed $V_{MG} = -2$ V and $V_{SG} = 0$ V and 1.5 V showing modulation of the energy bands to a stronger p-n configuration with increasing V_{SG} . **c** Hole density profile shows encroachment of MG channel by 50 nm from one side (100 nm from both sides) for a V_{SG} of 1.5 V. **d** A monotonic increase in photogenerated hole carrier density with increase in V_{SG} demonstrates the benefit of side gates.

In this section we have presented a simulation study of side gate architecture with an Si/SiO₂ system, as shown in Figure 2a. This device structure mimics the actual WSe₂/hBN device presented in this study. Si/SiO₂ was chosen since the material and physics models are well calibrated in TCAD software for this system. Although the dielectric constants for the materials are different, this mainly affects the quantitative extent of electrostatic doping keeping the qualitative trends the same, helping to realize the effect of side gates on phototransistor operation. In this structure we have used 20 nm thick SiO₂ as the gate dielectric and low doped p-type silicon of 10 nm thickness as the channel. Two metal side gates and a middle gate sandwiched between the two side gates were placed over the SiO₂. A metal substrate contact was added to the silicon channel to complete the electrical connections. Figure 2b shows the modulation of the silicon channel band diagram along the SG-MG-SG direction. Under fixed $V_{MG} = -2$ V, which keeps the MG channel region p-type, the side gate regions become more n-type as V_{SG} moves from 0 V to 1.5 V. This helps in efficient carrier separation with positive V_{SG} . Additionally, the hole density profile in Figure 2c

shows a near 5% encroachment (100 nm from both sides for a 2 μm MG width) of the MG channel due to applied V_{SG} of 1.5 V. The encroachment may be higher in case of a WSe_2 channel that was used in our device. This is likely due to the lower dielectric constant of WSe_2 as compared to Si. Hence, the lateral p-n junction electric field can penetrate further inside the middle gate channel of WSe_2 when compared to silicon, from the SG_MG direction. This reduction in MG channel width with positive V_{SG} reduces the dark current and enhances the detectivity as seen in the experimental data. Finally, Figure 2d shows the normalized photogenerated hole concentration (normalized against the photogenerated hole concentration at $V_{SG} = 0$ V) vs V_{SG} at a fixed $V_{MG} = -2$ V and for a fixed optical carrier generation rate that resembles the laser power in our study. A monotonic increase in photogenerated hole concentration with increase in V_{SG} demonstrates the benefit of side gates in enhancing photocarrier population which increases the I_{ph} and responsivity of the device, as seen in the experimental data.

Reviewer #2 (Remarks to the Author):

The authors have taken my comments in serious consideration and have made important changes and clarifications in the manuscript.

1. First re: my comments about the lack of evidence of photoswitching at 0.25 pW, that was my mistake from the misinterpretation of their original Fig.6. Now it is clear but I would recommend that the label that states the power is moved to the top of the figure to be immediately obvious than both panels refer to measurements at 0.25 pW.

The authors thank the reviewer for the suggestion. We have now moved the input power condition (0.25 pW) to the top of Fig. 6a.

2. Then, although the authors have now taken measurements of noise, they still insist to call the D^* as shot noise and flicker noise, whereas in reality the former is the theoretically maximum achievable D^* whereas the latter is the actual D^* of the devices that the authors report...so I would ask to make this obvious to the labeling and the description. Related to this the authors state that the dominant noise is the shot noise, but I don't agree as their actual D^* is almost 2 orders of magnitude lower than what would be achievable only with shot noise... the authors plot in Fig.6 b where they show the measured noise PSD also the theoretical noise PSDs from the shot noise, for the different values of V_{gm} and V_{gs} ? Then it will become obvious to what extent the device noise is governed by shot noise or flicker.

The authors thank the reviewer for the inputs regarding the explanation of D^* . In literature, a large number of studies on photodetection using van-der Waals materials report shot noise-limited detectivity. However, we had previously agreed with the reviewer that **D^* is dominated by flicker noise instead of shot noise.**

The reviewer has requested that we add the theoretical shot noise PSDs in Figure 6b for varying V_{MG} and V_{SG} . However, we have already shown this data in a comparison of flicker and shot noise governed D^* in SI section 11 (now it is SI section 13 in rearranged SI), as depicted below.

Figure 1. A comparison of flicker noise- and shot noise-limited specific detectivity (D^*) under No- V_{SG} and $V_{SG} = 2$ V, for $V_{MG} = -0.4$ V and 0.4 V

We have made two additional changes to bring in more clarity as per the suggestions.

(i) **We have changed the sentence in the introduction part**, by removing the reference to shot noise, with the following sentence, which we believe was creating the confusion regarding this work's claims to, "Typically, D^* depends on the dark current of the detector. Low dark current is essential for high D^* ".^{14,45}

(ii) **Labeling of D^* plot:** We have also changed the labeling in Fig. 6e for flicker noise as 'Measured' and shot noise as 'Theoretical limit' to segregate the two clearly, as shown in Figure 2, as per the reviewer's suggestion.

Figure 2: D^* vs τ_f plot, shows proper labeling of flicker and shot noise.

Original sentence in the main manuscript page 2:

Typically, D^* is extracted from the shot noise¹¹, which depends on the dark current of the detector. Low dark current is essential for high D^* .

Modified text:

Typically, D^* depends on the dark current of the detector. Low dark current is essential for high D^* .^{12,13}

Modified Figure 6e labeling:

3. A last point is about the time response, so from Fig.6 it seems that the device is not as fast as the authors state in other figures. Obviously the device gets slower with lower incident power, which is expected, so it is questionable to what extent this work achieved simultaneously fast response and very high sensitivity...I think this topic should be discussed and elaborated further to avoid overselling, unless I am wrong.

We elaborate on the reviewer's queries below:

The reviewer had rightly stated that devices could get slower at lower incident power. However, our study demonstrates two distinct aspects.

- a) A significant improvement in D^* is seen due to the side gates (Fig. 6c in main manuscript).
- b) **Significant reduction of the responsivity-speed trade off which is the major highlight of our work:** At a constant optical input power, with the side gate structure it is possible to achieve high responsivity without losing speed significantly. On the other hand, in conventional cases it is usually seen that, in order to achieve a highly responsive phototransistor one needs to compromise with the device speed at a fixed incident power.⁴

Additionally, for the same middle gate bias, Fig 6a (0.25 pW incident power) shows that the side gates improve the detection limit of our phototransistor (upper panel) in comparison to the No- V_{SG} case (lower panel) which shows no clear photoswitching. This data again emphasizes the advantage of SG on the detectability of weaker light signals, over No- V_{SG} condition (conventional phototransistor operation). However, it should be noted that when these devices

are studied at very low current values (~ few pA) which limit the speed of measurement, it could be beyond the measurement setup to capture fast response speeds at such low current values.

Hence, we believe, that the main claims of improved responsivity-speed trade-off and enhanced detectivity still hold. However, to bring in more clarity as per the reviewer's suggestion, we have now decoupled the responsivity-speed claim, from D^* in the main manuscript text to avoid any confusion. The following changes in the main manuscript have been done.

Original sentence in the main manuscript abstract:

A combined effect of increased illumination current and reduced dark current with the lateral p-n junction helps achieve responsivity and specific detectivity (D^*) enhancement upto 2.4x and 25x respectively at nearly the same switching speed (14-16 μ s) over a wide range of laser power (160 pW- 33 nW). High responsivity of 170 A/W at 300 pW laser power and a maximum measured flicker noise-limited D^* of 1.1×10^{12} Jones along with the ability to detect sub-1 pW laser switching are demonstrated.

Modified manuscript:

Increased illumination current with the lateral p-n junction helps achieve responsivity enhancement upto 2.4x at nearly the same switching speed (14-16 μ s) over a wide range of laser power (300 pW- 33 nW). The added benefit of reduced dark current enhances specific detectivity (D^*) by nearly 25x to yield a maximum measured flicker noise-limited D^* of 1.1×10^{12} Jones. High responsivity of 170 A/W at 300 pW laser power along with the ability to detect sub-1 pW laser switching are demonstrated.

Original sentence in main manuscript page 3:

As a result, the lateral p-n diode action enables responsivity and D^* enhancement by 1.1x-2.4x and 2x-5.3x respectively at nearly the same switching speed (14-16 μ s) over a wide range of laser power (300 pW- 33 nW). High responsivity (94 A/W), and speed (14 μ s) are demonstrated for 1 nW incident power, reaching peak values of 170 A/W at 300 pW. Along with this a flicker noise-limited measured D^* value of 1.1×10^{12} Jones, makes this one of the fastest high responsivity and high detectivity WSe₂ phototransistors till date.

Modified sentence:

As a result, the lateral p-n diode action enables responsivity enhancement by 1.1x-2.4x at nearly the same switching speed (14-16 μ s) over a wide range of laser power (300 pW- 33 nW). High responsivity (94 A/W), and speed (14 μ s) are demonstrated for 1 nW incident power, reaching peak values of 170 A/W at

300 pW. Along with this a flicker noise-limited maximum measured D^* value of 1.1×10^{12} Jones, enhanced by 4.2x-25x due to dark current reduction by the lateral p-n junction, makes this one of the fastest high responsivity and high detectivity WSe₂ phototransistors till date.

Reference:

4. Thakar, K., Mukherjee, B., Grover, S., Kaushik, N., Deshmukh, M., & Lodha, S. Multilayer ReS₂ Photodetectors with Gate Tunability for High Responsivity and High-Speed Applications. *ACS Appl. Mater. Interfaces* vol. 10, no. 42, pp. 36512–36522, (2018).